# AI Cartography: Mapping the Latent Landscape of AI Benchmark Ecosystems

Michael Hardy [* 1]   Anka Reuel [1]   Lijin Zhang [1]   Jodi M. Casabianca [2]   Sang Truong [1]   Yash Dave [1]   Hansol Lee [1]   Benjamin Domingue [1]   Sanmi Koyejo [1]

## Abstract

While aggregate leaderboard scores drive AI development, they contain substantial measurement noise whose sources and magnitudes remain unquantified, making it unclear when rankings reflect genuine capability differences versus evaluation artifacts. We introduce a framework for measuring the latent landscape in AI benchmark ecosystems. Applying Confirmatory Factor Analysis (CFA) and Generalizability Theory to 4,000+ models from the Open LLM Leaderboard, we decompose sources of ranking variance and establish: (1) structures assumed in current reporting practice underestimate the strength of relationships between benchmarks; (2) evidence of local dependence among leaderboard items, undermining uses of benchmarks as measurement instruments under current scoring systems; (3) contributor metadata explains more rank-relevant variance ($\approx 9\%$) than architecture or deployment categories in this context; (4) a manifest-score "scaling law" slope has low reliability ($\mathcal{R}_\beta = 0.53$); by contrast, the latent general-factor size slope is highly stable across ecosystem controls ($\mathcal{R}_g = 0.97$). We are able to provide unique insights into benchmark dynamics, such as which benchmarks are a function of LLM size and which can be oppositely impacted by post-training practices. We provide actionable diagnostics to determine how benchmark rankings can be trusted and how benchmark design can be improved.

## 1. Introduction

AI benchmark ecosystems have become the measurement instruments of modern machine learning. Open leaderboards such as HELM (Liang et al., 2023) and the Open

[1]Stanford University, Stanford, CA, USA [2]BroadMetrics, Edison, NJ, USA. Correspondence to: Michael Hardy <hardym[$\alpha\tau$]stanford[○]edu>.

*Proceedings of the 43rd International Conference on Machine Learning*, Seoul, South Korea. PMLR 306, 2026. Copyright 2026 by the author(s).

LLM Leaderboard (Fourrier et al., 2024) summarize model behavior on diverse tasks (e.g., MMLU, Hendrycks et al. 2020; BBH, Suzgun et al. 2022) into a small set of per-benchmark scores and then a single composite. This compression is operationally convenient, but it hides strong and rarely validated assumptions about what the scores *mean* (Salaudeen et al., 2025; Reuel et al., 2024; Truong et al., 2025; Casabianca, 2025). A benchmark score is typically treated as a scalar measure of a *latent* intended construct (e.g., "reasoning"). Formally, this assumes a latent variable exists and that aggregation is valid: item responses reflect a low-dimensional capability vector plus measurement error. When these assumptions fail, aggregation can produce *structural confounding*: the same total score may combine general capability, benchmark-specific residual structure, item redundancy, and evaluation artifacts in unknown proportions. (see § A). This matters because the ecosystem is central in AI development, referenced for funding, and used to infer scaling laws and compare training interventions. We seek to improve practitioner decisions by disentangling the various signals within benchmark scores.

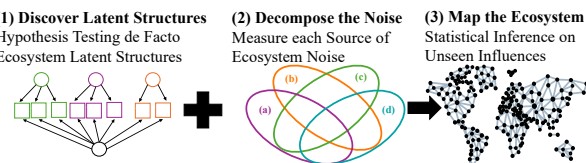

*Figure 1.* AI Latent Landscape Cartography: Study Overview

**Estimating instead of assuming.** A leaderboard ranking is implicitly a measurement-theoretic claim: that a scalar composite of item-level responses reflects one or more latent capabilities of interest, and that variation in the composite score corresponds to variation in the latent capability under measurement. Making this claim explicit requires:

1. **Exploring and confirming internal structure.** Which latent structure best explains the observed dependence among benchmark items? Current reporting assumes each benchmark measures a single independent construct; violating this assumption causes structural confounding (§2.1, Fig. 2).

2. **Ecosystem noise decomposition.** How much of the

observed score variance is attributable to construct irrelevant ecosystem features–benchmark, contributor practices, architecture, deployment? Without quantifying these, rankings can conflate signal with artifact (§2.2).

3. **Covariate-controlled inference.** Can model size be disentangled from training innovation once latent structure and ecosystem variance are modeled jointly? Naïve regressions on manifest scores confound all three (§2.3).

In response to those needs, we present a framework for measuring AI capabilities and scaling laws controlling for many sources of variance in AI benchmark ecosystems. We measure the score variation found within each LLM (§ 2.1) and contributed by their ecosystem (§ 2.2) before combining these tools to map de-noised scaling laws (§ 2.3). In brief, we measure the leaderboard ecosystem as a noisy "landscape" of structural, ecosystem, and statistical variation. This framework has three progressive methods (Fig. 1):

1. **Method 1: Structural discovery (CFA/structural equation modeling [SEM]):** we statistically test hypothetical latent internal structures across bootstrapped sets of items (Bollen, 1989; Browne & Cudeck, 1992; Brown, 2015; Muthén, 1993; Saris et al., 2009). Because the variables of interest are not observed, we take rigorous approaches to determine the latent structure that best explains AI capabilities, avoiding overfit propensities (§ 2.1, C-D). **Preview**: *regardless of the latent structure tested, significant nonrandom error persists* (unrelated to the AI capability of interest) that could threaten the validity of leaderboard score comparisons.

2. **Method 2: Ecosystem variance decomposition (G-theory):** we use a fully crossed mixed-effects variance decomposition model (architecture, benchmark, contributor/provenance, deployment type, § 2.2, G) to quantify sources of leaderboard variability and ranking reliability (Brennan, 2001; Cronbach & Shavelson, 2004; Hardy, 2025a). **Preview**: *we can more reliably rank-order AI performance by contributor metadata than we can based on architecture or deployment type*.

3. **Method 3: Mixed-effects latent regression:** we regress AI behaviors on the bifactor structure from Method 1 above using noise structures from Method 2 in a latent regression to estimate scaling laws specific to observable practices in ML (De Boeck, 2011; Chalmers, 2015). Using latent regression (§ 2.3, H), we estimate ecosystem effects on specific AI benchmark behaviors so we can infer what tuning practices help underlying capabilities as they scale. **Preview**: *while scaling improves performance, this is not uniformly true*, as scaling progress in one benchmark can be negatively associated with performance in another.

**How the three methods interlock** The framework's sequential logic is deliberate: Method 1 establishes *what* is being measured, Method 2 quantifies *how noisy* measurements are, and Method 3 estimates *what drives* variation in the latent quantities after controlling for both structure and noise. CFA identifies a bifactor latent structure, motivating an investigation into general scaling laws and reveals persistent residual dependencies, motivating generalizability studies of ecosystem noise sources that latent structures alone cannot absorb. Then, variance decomposition measures the facets of the noise landscape, such as contributor variance ($\approx$9%, Table 2) and model size-linked variance ($\approx$40%, § G.8), but this estimation cannot disentangle these sources across *latent* levels. This motivates latent regression, which regresses capacities directly on size and post-training while controlling for ecosystem variance, completing the de-contaminated picture of how model capabilities are shaped.

**Contributions.** To the best of our knowledge, this is the first study to use these advanced methods in AI benchmarking and to measure and control for ecosystem factors. By so doing, we show how current reporting misrepresents the underlying constructs measured (§ 4.1). We provide de-noised scaling laws for general AI capability and for benchmark-specific constructs (Figs. 3-4, § 4.3). We disentangle post-training practices from model size effects (Table 3, § 4.3), and measure the share of AI score variation attributable to each source of variation (§ 4.2, Table 2). We demonstrate how, after noise controls, some benchmarks have markedly less usable signal (§ 4.4). We offer novel methodological solutions that make advanced measurement theoretic tools usable in benchmarking ecosystems (§ 2). By offering clarity around ecosystem and AI factors influencing benchmark rankings (§ 4.5), we provide a map of the latent landscape for a popular AI benchmark suite.

## 2. Methods: from Benchmarking to Measuring

The central challenge in AI evaluation is separating signal (genuine capability) from noise (measurement artifacts, see Appendix § A). Leaderboards compress a high-dimensional response profile into per-benchmark accuracies and then a single composite. This compression is convenient but implicitly assumes that (i) each benchmark is an approximately unidimensional scale, (ii) benchmark totals are commensurate indicators of a smaller set of underlying capabilities, and (iii) local independence–items are no longer correlated after controlling for the latent construct. When these assumptions fail, aggregation makes scores *structurally uninterpretable* and confounds signal with multiple forms of ecosystem noise (Salaudeen et al. 2025; Casabianca 2025; § A).

**Pseudo-replication.** In particular, leaderboard submissions are not i.i.d. draws from a population of independent models. Many entries are fine-tunes, merges, or quantizations of a small number of base architectures, creating

hierarchical dependence that inflates precision and biases structure toward dominant families. Methods 2 and 3 control for such contribution phenomena explicitly.

**High-dimensional item pools.** Each benchmark contains hundreds to thousands of items (Table 4), so fitting a CFA (Method 1, § 2.1) or IRT (Method 3, § 2.3) model to the full item set is computationally infeasible and invites overfitting to item-specific artifacts. We address this via *meta-analytic item set bootstrapping*: for each of $B$ replications, we sample a subset of $r_k$ items per benchmark $k$, fit the target model, and aggregate results across replications (see E). The results are *population quantities*, the expectations under the ecosystem-induced distribution. When meta-aggregating model statistics, Method 1 uses mixed-effects meta-regression for reported fit statistics (see §E.3) and Method 3 uses inverse-error weighted methods drawn from the posterior (see §E.4).

## 2.1. Method 1: CFA for latent structure discovery

CFA can statistically test which latent factor structure best predicts the observed dependence (tetrachoric/robust correlations) among benchmark items and LLM responses. Because the variables of interest–latent AI capabilities–are unobserved, we compare six competing structural hypotheses (Table 1, Fig. 2) and let the data adjudicate among them. By so doing, we create a coarse map of LLMs' "vectors of mind" (Thurstone, 1935) statistically testing each as a "nomological network" (Cronbach & Meehl, 1955).

### 2.1.1. THE GENERAL CFA MEASUREMENT MODEL

For LLM $i$ and item $j$ ($j$=1, ...$p$),[1] a general[2] CFA model is

$$\mathbf{y}_i = \boldsymbol{\nu} + \boldsymbol{\Lambda}\boldsymbol{\eta}_i + \boldsymbol{\epsilon}_i, \quad \boldsymbol{\eta}_i \sim \mathcal{N}(\mathbf{0}, \boldsymbol{\Phi}), \ \boldsymbol{\epsilon}_i \sim \mathcal{N}(\mathbf{0}, \boldsymbol{E}), \ (1)$$

where $\boldsymbol{\Lambda}$ is a $p \times m$ matrix of factor loadings encoding which items load on which latent factors (as in Fig. 2), $\boldsymbol{\Phi}$ is the $m \times m$ factor covariance, and $\boldsymbol{E}$ is the (typically diagonal) residual covariance.[3] The model-implied covariance is $\boldsymbol{\Sigma}(\boldsymbol{\theta}) = \boldsymbol{\Lambda}\boldsymbol{\Phi}\boldsymbol{\Lambda}^\top + \boldsymbol{E}$. Each candidate structure $s \in \mathcal{S}$ specifies a distinct set of constraints on $(\boldsymbol{\Lambda}, \boldsymbol{\Phi}, \boldsymbol{E})$, and estimation proceeds by minimizing a discrepancy $F(\boldsymbol{\Sigma}(\boldsymbol{\theta}), \mathbf{S})$ between the model-implied and observed sample covariance $\mathbf{S}$. CFA primarily predicts the implied *dependence structure* (correlations), rather than labels; this enables hypothesis testing by providing evidence confirming the internal struc-

ture of the leaderboard (Casabianca, 2025).

### 2.1.2. CFA METHODOLOGICAL SOLUTIONS

**Addressing overfitting tendencies of flexible latent structures.** Higher-capacity models (e.g., correlated bifactor) can absorb covariance artifacts unrelated to true latent organization, inflating apparent fit. **Solution 1: Non-traditional out-of-sample prediction.** For each bootstrap, we compute held-out AUC and MAE on within-sample latent factor scores, providing a direct overfitting diagnostic beyond traditional indices (Table 5, § C.3). **Solution 2: Within-replication permutation controls.** For each bootstrap $b$, we additionally construct a randomized item-to-benchmark mapping $\pi^{(b)}$ preserving per-benchmark counts $\{r_k\}$ and refit. The contrast $\Delta T_s^{(b)} = T(s \mid \pi^{(b)}) - T(s \mid \mathrm{id})$ quantifies how much of $s$'s advantage depends on true benchmark grouping rather than model flexibility (§ C.3, Table 9). **Solution 3: Multiple estimation methods.** We fit with both Diagonally Weighted Least Squares (DWLS) and Metropolis-Hastings Robbins-Monro (MH-RM) estimations.

**Identifying residual misfit in each structure by aggregating score-test trends.** Global fit indices (e.g., RMSEA, CFI) do not localize *where* a latent structure fails. We compute the locally most powerful Lagrange-multiplier or modification-index (MI) statistics,[4] $\mathrm{MI}(\psi)$, for every candidate residual covariance and cross-loading $\psi = \Theta_{ab}$, then aggregate Standardized Expected Parameter Changes (SEPCs) across bootstraps and structures. This maps item pairs whose residual associations exceed what the latent structure can explain, and whether such violations are stable structural properties or item-idiosyncratic (§ D; Figs. 6, 7).

## 2.2. Method 2: Fully crossed generalizability study

Variance decomposition can quantify how much observed score variation is attributable to each ecosystem facet and interaction and assess the stability of observed rankings.

---

[1] A comprehensive notation guide can be found in § K

[2] The general form of the present study uses binary response data. Let $y_{ij} \in \{0, 1\}$ be LLM submission $i$'s response to item $j$, and let $\boldsymbol{\eta}_i \in \mathbb{R}^m$ be latent abilities. Under a general latent-response formulation for *binary* items:
$$y_{ij}^\star = \nu_j + \boldsymbol{\lambda}_j^\top \boldsymbol{\eta}_i + \epsilon_{ij}, \ \epsilon_{ij} \sim \mathcal{N}(0, \varepsilon_j), \ y_{ij} = \mathbb{1}[y_{ij}^\star > \tau_j].$$
[3] Traditionally, $\boldsymbol{\Theta}$ is used for $\mathrm{Cov}(\boldsymbol{\epsilon})$; we use $\boldsymbol{E}$ here to distinguish from $\boldsymbol{\Theta}$ in Method 3.

[4] Typically used to identify item redundancy, bootstrapped 1-df score tests can quantify residual covariances for each structure and identify violations of local dependence that threaten the use of benchmarks as a measurement tool:

**Proposition 2.1** (Modification indices as Lagrange-multiplier tests)**.** *Let $\ell(\tilde{\boldsymbol{\theta}})$ be twice continuously differentiable at the constrained optimum $\hat{\boldsymbol{\theta}}$, with $\psi$ constrained to zero. Under standard regularity conditions, the score statistic $\mathrm{MI}(\psi) = U_\psi(\hat{\boldsymbol{\theta}})^\top \mathcal{I}_{\psi\psi\cdot\theta}(\hat{\boldsymbol{\theta}})^{-1} U_\psi(\hat{\boldsymbol{\theta}}) \xrightarrow{d} \chi_1^2$, where $U_\psi(\hat{\boldsymbol{\theta}}) = \partial\ell/\partial\psi\big|_{\psi=0}$ is the score and $\mathcal{I}_{\psi\psi\cdot\theta}$ is the Schur complement of the partitioned Fisher information. The SEPC $\widehat{\Delta\psi} \approx \mathcal{I}_{\psi\psi\cdot\theta}^{-1} U_\psi(\hat{\boldsymbol{\theta}})$ estimates the magnitude and sign of the residual association on a correlation scale.* (Proof: § D.1.2, see Appendix § D)

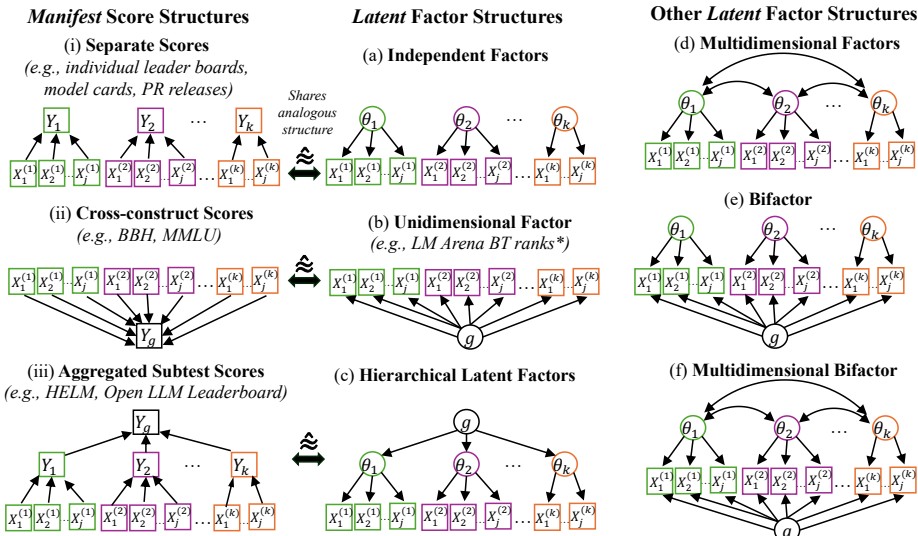

*Figure 2.* **Possible Manifest and Latent Scoring Structures** used or assumed in AI Benchmark Ecosystem. **Left (i, ii, iii)**: common manifest scores structures. **Middle (a, b, c)**: analogous latent structures, where pairs (a,i), (b,ii), and (c,iii) differ in the causal direction implied by their estimations. **Right (d, e, f)**: other potential latent structures which may be explanatory of observed scores. *Structure for a LMArena leaderboard (e.g., Text) under Bradley-Terry-style pairwise comparison unidimensionality assumptions (Chiang et al., 2024).

### 2.2.1. The Generalizability study model

We use leaderboard metadata to identify characteristics belonging to a set of benchmark scores: (a) architecture, (b) benchmark uniqueness, (c) contributors[5] to the leaderboard, and (d) deployment, post-training, and tuning decisions. (More information in § G). Let $s_i$ be the score of submission $i$ on benchmark $b_i$ with architecture $a_i$, contributor $c_i$, and deployment $d_i$. This crossed random-effects model is

$$s_i = \mu + \sum_{S \in \mathcal{S}:\mathcal{P}^*(\{a,b,c,d\})} u_{S(i)} + \varepsilon_i, \quad (2)$$

where $\mathcal{P}^*$ denotes the non-empty Power Set defining the facets (i.e., all main effects and their interactions $A, B, AB, AC, \ldots, ABCD$), each $u_{S(i)} \sim \mathcal{N}(0, \sigma_S^2)$, and $\varepsilon_i$ absorbs the residual and highest-order interaction ($ABCD$). Variance shares $p_S = \sigma_S^2/(\sum_{S'} \sigma_{S'}^2 + \sigma_\varepsilon^2)$ quantify the noise sources, with additional details in § G.

### 2.2.2. G-study methodological solutions

**Controlling for influence of model size with random slopes.** Without controlling for size, variance attributable to each facet may simply reflect scale differences rather than genuine ecosystem effects. Extend (2) with the log-size

covariate $x_i = \log_{10}(\#\mathrm{params}_i(\mathrm{B}))$ and random slopes:

$$s_i = \mu + \beta x_i + \sum_{S \in \mathcal{S}} \left(u_{g_S}^S + v_{g_S}^S x_i\right) + \varepsilon_i, \quad (3)$$

where $u_{g_S}^S \sim \mathcal{N}(0, \sigma_{S,0}^2)$ are random intercepts and $v_{g_S}^S \sim \mathcal{N}(0, \sigma_{S,1}^2)$ are random slopes, constrained uncorrelated within each term. This decomposes total variance into intercept components (baseline performance) and slope components (scaling-relationship heterogeneity), enabling the novel reliability diagnostics $\mathrm{SNR}_\beta$ and $\mathrm{PSI}_S$ defined below.

**Robustness checks** Because the four-facet crossing is sparse, we interpret higher-order components descriptively and emphasize robustness of broad variance-ordering patterns rather than individual high-order interaction estimates. We perform robustness checks on decomposition sensitivity, including adjusting category granularity and Bayesian estimation. We verify that the observed order of $B \gg C > A > D$ in our main finding is stable.[6]

**Definition 2.2** (Slope Signal-to-Noise Ratio and Generalizability)**.** The SNR of the fixed-effect slope $\beta$ is the ratio of its square to the total variance of the random slopes: $\mathrm{SNR}_\beta \equiv \frac{\beta^2}{\sum_S \sigma_{S,1}^2}$ and $\mathcal{R}_\beta \equiv \frac{\beta^2}{\beta^2 + \sum_S \sigma_{S,1}^2}$, where $\mathcal{R}_\beta$ is the reliability of a general scaling law. $\mathrm{SNR}_\beta \gg 1$ implies a robust, universal scaling law; $\mathrm{SNR}_\beta \approx 1$ or less

---

[5]Contributors represent provenance: the facet is an operational metadata grouping. It should not be interpreted as a person-level causal effect; it aggregates submission source, availability status, documentation practices, and unobserved training/provenance differences. This includes variation due to missing model card and subsequent removal (Clémentine Fourrier [@clefourrier], 2024).

[6]We also estimate (3)*, with correlations are estimated freely. (3)* resulted in a larger contributor-linked variance share, $p_C = 0.12$. The fit was singular, so we only report it in Table 14. § G.3 has details.

*Table 1.* Latent Structural Factor Models for AI Benchmark Ecosystem Analysis and Basic Results

| Structure | Formulae: $y_{ikj} =$ | Covariance | Key Idea | RMSEA | AUC | MAE |
|---|---|---|---|---|---|---|
| Independent Factors (Fig 2.a, § C.2.1) | $\lambda_{kj} f_{ik} + \epsilon_{ikj}$ $\mathbf{\Lambda\Lambda}^\top + \boldsymbol{E}$ | Block-diagonal across benchmarks (no inter-benchmark covariance) $\mathrm{Cov}(f_{ik}, f_{ik'}) = 0$ | No meaningful cross-domain transfer. One factor per benchmark. (6 factors) | 0.09 | **0.95** | 0.22 |
| One General Factor/ Unidimensional (Fig 2.b, § C.2.2) | $\lambda_{kj} g_i + \epsilon_{ikj}$ $\mathbf{\Lambda\Lambda}^\top + \boldsymbol{E}$ | All covariance explained by one source (+ diagonal residuals): $\mathrm{Var}(g_i) = 1$ | Single, general factor ($g$) explaining all shared variance. (1 factor) | 0.06 | 0.90 | 0.23 |
| Hierarchical Structure/2nd-Order (Fig 2.c, § C.2.3) | $\lambda_{kj} f_{ik} + \epsilon_{ikj},$ $f_{ik} = \gamma_k g_i + \delta_{ik}$ $\gamma\gamma^\top + \mathbf{\Delta}$ | Inter-benchmark covariance implied via $g$ through $f_k$ (hierarchically structured) $\mathrm{Cov}(f_{ik}, f_{ik'}) = \gamma_k \gamma_{k'}$ | $g$ acts as the cause of correlations among the first-order factors (7 factors) | 0.05 | 0.94 | 0.24 |
| Correlated Factors/ Multidimensional (Fig 2.d, § C.2.4) | $\lambda_{kj} f_{ik} + \epsilon_{ikj}$ $\mathbf{\Lambda\Psi\Lambda}^\top + \boldsymbol{E}$ | Cross-benchmark covariance via $\mathbf{\Psi}$ (dense factor correlation matrix) $\mathrm{Cov}(\mathbf{f}_i) = \mathbf{\Psi}$ | Benchmark abilities are distinct but interrelated (6 factors) | 0.05 | 0.73 | 0.35 |
| Bifactor/ Orthogonal General Factor) (Fig 2.e, § C.2.5, I) | $\lambda_{kj}^{(g)} g_i + \lambda_{kj}^{(f)} f_{ik} + \epsilon_{ikj}$ $\mathbf{\Lambda}_g \mathbf{\Lambda}_g^\top + \mathbf{\Lambda}_f \mathbf{\Lambda}_f^\top + \boldsymbol{E}$ | Covariance decomposes into general + benchmark-specific; no correlations among factors: $g_i \perp f_{ik}, f_{ik} \perp f_{ik'}$ | Assumes benchmark-specific ($f_k$) variance is orthogonal to general ($g$) sources. (7 factors) | 0.05 | **0.95** | **0.21** |
| Correlated Bifactor (Fig. 2.f, § C.2.6) | $\lambda_{kj}^{(g)} g_i + \lambda_{kj}^{(f)} f_{ik} + \epsilon_{ikj}$ $\mathbf{\Lambda}_g \mathbf{\Lambda}_g^\top + \mathbf{\Lambda}_f \mathbf{\Psi\Lambda}_f^\top + \boldsymbol{E}$ | Residual cross-benchmark covariance after removing $g$ captured by $\mathbf{\Psi}$ over specifics: $g_i \perp \mathbf{f}_i, \ \mathrm{Cov}(\mathbf{f}_i) = \mathbf{\Psi}$ | Tests for residual latent covariance after partialing $g$ (7 factors) | **0.04** | 0.77 | 0.32 |

Note: $y_{ikj}$ = response of LLM $i$ to item $j$ in benchmark $k$; $g_i$ = general factor; $f_{ik}$ = benchmark-specific factors; $\lambda$ = loadings; $\epsilon_{ikj} \sim \mathcal{N}(0, \theta_{kj})$ = residuals; $\mathbf{\Psi}$ = correlation matrix; $\boldsymbol{E}$ = diagonal residual covariance matrix. Model details are in C and Tables 7–9. Study-wide notation is in § K and Bifactor code is found in I (all code and model specifics are in the online repository).
The final columns report the aggregated sample-weighted mean AUC and MAE across bootstraps for held out observations. RMSEA is a traditional CFA metric aggregated across estimations. RMSEA values reported in an earlier version were the result of a transcription error.

indicates that the average scaling law is misleading because context-specific variation rivals the fixed effect (see § 4.3).

**Definition 2.3** (Proportion of Slope Instability). For each facet or interaction term $S$: $\mathrm{PSI}_S \equiv \frac{\sigma_{S,1}^2}{\sum_{S'} \sigma_{S',1}^2}$. $\mathrm{PSI}_S$ attributes scaling-law instability to its ecosystem sources: a large PSI implies that different sources of variation respond to size in fundamentally different ways (see Table 2).

We use the *bifactor* downstream because it balances covariance fit, predictive performance, parsimony, and interpretability. The *correlated bifactor* acts as an upper-bound flexibility check rather than the preferred scientific structure.

**2.3. Method 3: Mixed-effects latent regression**

To estimate *ecosystem-controlled scaling laws* on *latent abilities*, both general and benchmark-specific, Method 3 operates directly in the latent space, unifying measurement and regression into a single coherent model.

2.3.1. HIERARCHICAL TWO-COMPONENT MODEL

The hierarchical mixed-effects latent regression model comprises a **measurement layer** (items → latent traits) and a **structural layer** (latent traits → covariates).

**Measurement layer (bifactor IRT).** Let $\boldsymbol{\theta}_i = (\theta_{i0}, \theta_{i1}, \ldots, \theta_{iK})^\top \in \mathbb{R}^{K+1}$ be the latent ability vector for LLM $i$, with general factor $g = \theta_{i0}$ and benchmark-specific factors $\theta_{ik}$. For item $j$ in benchmark $k(j)$:

$$\Pr(y_{ij} = 1 \mid \boldsymbol{\theta}_i) = \sigma\big(a_{j0}\theta_{i0} + a_{j,k(j)}\theta_{i,k(j)} - b_j\big), \quad (4)$$

where $\sigma(t) = (1 + e^{-t})^{-1}$, $a_{j0}$ and $a_{j,k(j)}$ are discrimination parameters for the general and specific factors (compare with $\lambda$ in Table 1, bifactor), and $b_j$ is item difficulty. Each item loads on the general factor *and* one specific factor.

**Structural layer (mixed-effects regression on latent traits).** The matrix of latent traits $\mathbf{\Theta} \in \mathbb{R}^{N \times (K+1)}$ is decomposed into fixed and random components:

$$\mathbf{\Theta} = \mathbf{V\Gamma} + \mathbf{W}\boldsymbol{\zeta} + \mathbf{E}, \quad (5)$$

where $\mathbf{V}$ is the $N \times F$ fixed-effect design matrix (intercept, $\log_{10}$(#params), deployment indicators), $\mathbf{\Gamma}$ is the $F \times (K+1)$ fixed-effect coefficients (each column $\boldsymbol{\gamma}_k$ has covariate effects on latent dimension $k$), $\mathbf{W}$ is the random-effect design matrix for contributor group, $\boldsymbol{\zeta} \sim \mathcal{N}(\mathbf{0}, \mathbf{\Sigma}_\zeta)$ captures contributor-level heterogeneity, and $\mathbf{E}$ is residual latent variation with rows $\mathbf{e}_i \sim \mathcal{N}(\mathbf{0}, \mathbf{\Sigma}_E)$, accounting for pseudo-replication. Because regressing manifest benchmark scores on size conflates $g$, benchmark specifics, and benchmark-dependent measurement error, a key output is the *scaling vector* $\boldsymbol{\beta} = (\beta_0, \beta_1, \ldots, \beta_K)$, where $\beta_k = \Gamma_{x,k}$ is the effect of log-size on latent ability $k$. This replaces the scalar "scaling law" with a dimension-specific profile.

**Proposition 2.4** (Latent regression corrects size effects confounded by measurement error). *Assume the observed benchmark score $\bar{y}_{ik}$ is a noisy proxy for $\theta_{ik}$ with benchmark-dependent attenuation: $\bar{y}_{ik} = \lambda_k \theta_{ik} + e_{ik}$, $\mathbb{E}[e_{ik} \mid x_i] = 0$, $\mathrm{Var}(e_{ik})$ varies with $k$. Then OLS regression of $\bar{y}_{ik}$ on $x_i$ yields benchmark-dependent slope attenuation proportional to $\lambda_k$, while regression of $\theta_{ik}$ on $x_i$*

*(as in 5) recovers the latent slope $\beta_k$ up to the mixed-model dependence structure.*

*Proof.* By linearity, $\mathbb{E}[\bar{y}_{ik} \mid x_i] = \lambda_k \mathbb{E}[\theta_{ik} \mid x_i]$ and the population regression coefficient is $\lambda_k \beta_k$ when $\theta_{ik} = \alpha_k + \beta_k x_i +$ noise, whereas regressing $\theta_{ik}$ yields $\beta_k$. Mixed effects preserve unbiasedness under correct specification by accounting for cluster dependence in the likelihood. □

### 2.3.2. LATENT REGRESSION METHOD ESTIMATION

The term $\mathbf{W}\boldsymbol{\zeta}$ in (5) models contributor-level random effects on each latent dimension, absorbing the cluster structure induced by pseudo-replication and preventing inflated precision. The bifactor measurement structure coupled with fixed and random effects on $K + 1$ latent dimensions creates a high-dimensional, multimodal likelihood surface. Thus, we fit (4)–(5) with Metropolis-Hastings Robbins-Monro (MH-RM) (Cai, 2010b; Chalmers, 2015).[7] Similar to Method 1 (§ 2.1.2), item set bootstrapping is used.

### 2.3.3. SLOPE RELIABILITY IN LATENT SPACE

For each latent dimension $d \in \{0, 1, \ldots, K\}$, we define the *slope reliability index*: $\mathcal{R}_d \equiv \frac{\beta_d^2}{\beta_d^2 + \tau_d^2}$, where $\tau_d^2 = \mathrm{Var}(v_{\cdot,d})$ is the random-slope variance for dimension $d$ across bootstraps (see § E). High $\mathcal{R}_d$ indicates that the scaling law for latent ability $d$ is stable across contributors; low $\mathcal{R}_d$ signals context-dependent scaling (Table 3).

## 3. Experiment and Data

We analyze public submissions to the Hugging Face Open LLM Leaderboard (Fourrier et al., 2024),[8] a comprehensive and diverse collection of AI benchmark data, encompassing over 4,000 unique Large Language Model (LLM) instances. This large-scale dataset provides an ideal testbed for noise decomposition: models vary across architectures, contributors, deployment types, and scales, enabling systematic quantification of noise sources. The public dataset contains model metadata, benchmark scores, and availability of item-level responses. The Open LLM Leaderboard is a large-scale initiative to transparently track, rank, and evaluate open-source LLMs. The analyzed benchmarks are six widely used tasks with relatively objective metrics: IFEval (Zhou et al., 2023), BBH (Suzgun et al., 2022), MATH Level 5 (Hendrycks et al., 2021), GPQA (Rein et al., 2023),

---

[7]MH-RM is a strong choice in high-dimensional IRT as it alternates between (i) a MH step to sample from the conditional posterior of latent traits given current parameters, and (ii) a RM step to update parameters along the stochastic gradient of the marginal log-likelihood.

[8]https://huggingface.co/spaces/open-llm-leaderboard/open_llm_leaderboard

*Table 2.* Method 2 Variance Component Estimates for (2)-(3)

| Variation Source | $p_S$ (2) | $p_S$ (3) | $\mathbf{PSI}_S$ |
|---|---|---|---|
| A (Architecture) | 0.042 | 0.063 | 0.13 |
| B (Benchmark) | 0.431 | 0.461 | 0.30 |
| C (Contributor) | 0.086 | 0.078 | 0.17 |
| D (Deployment) | 0.019 | 0.011 | 0.03 |
| A×B | 0.029 | 0.037 | 0.05 |
| A×C | 0.035 | 0.041 | 0.06 |
| A×D | 0.002 | 0.003 | 0.00 |
| B×C | 0.037 | 0.027 | 0.03 |
| B×D | 0.049 | 0.046 | 0.02 |
| C×D | 0.008 | 0.011 | 0.01 |
| A×B×C | 0.023 | 0.038 | 0.07 |
| A×B×D | 0.004 | 0.006 | 0.01 |
| A×C×D | 0.080 | 0.047 | 0.07 |
| B×C×D | 0.006 | 0.018 | 0.05 |
| $\epsilon_{\mathrm{resid}}$ (+ A×B×C×D) | 0.150 | 0.113 | |

Proportion of variance explained $p_S$ represents the total variance associated with (both slope and) intercept. The raw variances for the base (2) and slope (3) models can be found in Table 12 and 11, respectively. Robustness checks and Bayesian posterior draws of $p_S$ (2) are in Tables 10 and 13, respectively.

MuSR (Sprague et al., 2024), and MMLU-Pro (Wang et al., 2024). Appendix § B has additional data details.

The CFA and latent regression analyses use item-level responses, while the variance decomposition analyses focus on a *model–benchmark* score: each observation corresponds to one evaluated model (as represented by a specific leaderboard submission) on one benchmark, which is essential for identifying how reliably one can generalize across benchmarks rather than merely fit the leaderboard's composite.

## 4. Results and Discussion

Benchmark ecosystems are now treated as scientific instruments, used to rank models, infer scaling laws, and justify training interventions. Our results show that, without explicit measurement modeling, these instruments mix signal with multiple layers of noise. We summarize the implications as a *Latent Cartography*: we map leaderboard topographies defined by latent ability structure, ecosystem facets of variation, LLM covariates, and residual item-level artifacts.

### 4.1. What the ecosystem is actually measuring

**A size-related general factor dominates; benchmark factors are largely residual.** Across item-resampled CFA comparisons, bifactor-family models best predict the dependence structure (see § F), implying that much of the cross-benchmark covariance is captured by a single general factor $g$ with benchmark-specific residual structure.

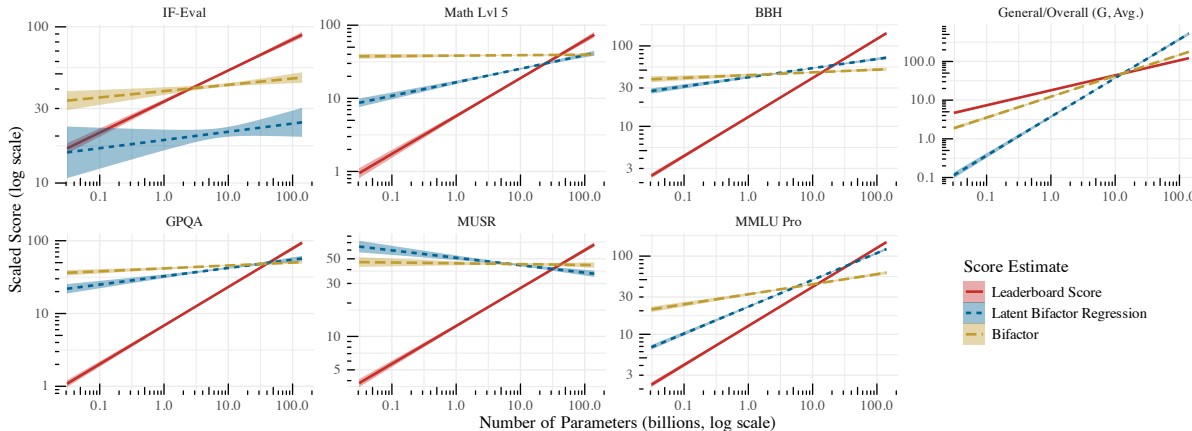

*Figure 3.* Noise-controlled Scaling Laws: Log-log of Performance by Number of Parameters. Shaded regions indicate the 95% confidence bands for the linear relationship estimated by robust regression using an M estimator with Tukey's biweight (Venables & Ripley, 2002). Prior to log transformation of the y-axis, each estimated latent score $\Theta_k$ is converted from its assumed distribution to positive values via the cumulative normal distribution function, $S_k = \Phi(\Theta_k)$ before all scores are min-max scaled to the range $s \in S_k : [1 \dots 100]$.

*Table 3.* Reliabilities of Slopes in Latent Regression

| Coefficient | stat | g | bbh | gpqa | ifeval | mmlu | musr | math5 |
|---|---|---|---|---|---|---|---|---|
| $\log_{10}$(#params) | $\beta$ | 1.05 | -0.05 | 0.07 | -0.08 | 0.52 | 0.04 | 0.13 |
| | $\mathcal{R}_d$ | 0.97 | 0.05 | 0.20 | 0.39 | 0.57 | 0.07 | 0.18 |
| 'Chat template'/IFT | $\gamma$ | -0.22 | -0.03 | -0.66 | 1.82 | 0.08 | -1.03 | 0.29 |
| | $\mathcal{R}_d$ | 0.86 | 0.05 | 0.44 | 0.99 | 0.19 | 0.47 | 0.86 |
| Chat (RLHF, DPO, etc) | $\gamma$ | -1.05 | 0.02 | 0.42 | -0.84 | -0.31 | 0.52 | -0.92 |
| | $\mathcal{R}_d$ | 0.64 | 0.01 | 0.37 | 0.72 | 0.14 | 0.36 | 0.79 |
| Domain-specific tune | $\gamma$ | -1.16 | -0.07 | 0.20 | -0.80 | -0.46 | 0.39 | -0.43 |
| | $\mathcal{R}_d$ | 0.98 | 0.18 | 0.31 | 0.95 | 0.56 | 0.45 | 0.86 |
| Merges & MoErges | $\gamma$ | -0.65 | 0.09 | 0.24 | -0.79 | -0.64 | 0.41 | -0.58 |
| | $\mathcal{R}_d$ | 0.89 | 0.05 | 0.35 | 0.95 | 0.65 | 0.31 | 0.80 |
| Pretrained only | $\gamma$ | -1.19 | 0.01 | 0.14 | -0.86 | -0.40 | 0.70 | -0.61 |
| | $\mathcal{R}_d$ | 0.76 | 0.00 | 0.21 | 0.91 | 0.35 | 0.80 | 0.87 |
| Continuous pretrain | $\gamma$ | -1.51 | -0.17 | 0.06 | -1.05 | -0.34 | 0.72 | -0.74 |
| | $\mathcal{R}_d$ | 0.99 | 0.32 | 0.07 | 0.86 | 0.26 | 0.77 | 0.79 |
| Mixture of Experts | $\gamma$ | -0.84 | -0.04 | 0.11 | 0.01 | -0.29 | -0.08 | 0.00 |
| | $\mathcal{R}_d$ | 0.97 | 0.10 | 0.19 | 0.00 | 0.35 | 0.27 | 0.00 |

Results from bootstrapped latent regression analysis. Variables are taken directly from the Hugging Face Open LLM Leaderboard. Only $\log_{10}$(#params) is continuous; all others are binary. $\beta$ and $\gamma$ represent the per-factor estimates associated with that coefficient and $\mathcal{R}_d$ represents the reliability of the slope defined by that estimate. Chat and 'Chat template' are not mutually exclusive.

This supports a *two-level* decomposition: performance $\approx g + \{s_k\}_{k=1}^K +$ item noise, where $s_k$ captures what remains common within benchmark $k$ after partialing out $g$. $g$ is strongly correlated with number of parameters (Fig. 3).

**Proposition 4.1** (Benchmark scores are not construct-preserving summaries)**.** *If the ecosystem is better approximated by a bifactor model than by per-benchmark unidimensional models, then benchmark total scores are not sufficient statistics for latent capability profiles: there exist submissions $i \neq i'$ such that* $\text{Score}_{ik} = \text{Score}_{i'k} \ \forall k$ *but* $(g_i, s_{i1}, \dots, s_{iK}) \neq (g_{i'}, s_{i'1}, \dots, s_{i'K})$.

**Benchmark labels do not fully organize residual dependence.** Permutation controls show that flexible SEMs can

retain strong fit even under randomized item-to-benchmark assignment. The implication is that benchmark membership is an incomplete index of the residual covariance structure once $g$ is removed. Shared evaluation artifacts, and latent overlaps cut across benchmark labels. Consequently, benchmark ecosystems should be evaluated as *measurement systems*, not as a set of independent test suites. Importantly, all latent structures tested still underestimate the strength of the interbenchmark item relationships (see also § D, § F and Fig. 7). Taken together, the meta-analytic bootstrap, permutation control, and MI-based local diagnostics provide convergent evidence that (i) unidimensional and independent-benchmark assumptions are untenable, (ii) a strong general factor is real and stable across item perturbations, and (iii) benchmark labels only partially explain residual structure.

### 4.2. Actionable cartographic layers

**Predictable benchmark difficulty (a controllable nuisance)** Variance decomposition shows that a large share of leaderboard score variance is attributable to the benchmark main effect (difficulty). This is "nuisance" variation in the sense that it is predictable from benchmark identity and can be controlled by (i) stratified reporting, (ii) difficulty-normalized scoring, or (iii) explicit modeling (our G-study).

**Contributor variance (signal about training; noise for architecture)** The contributor/provenance facet is consistently larger than the combined architecture and deployment main effects plus their interaction, across estimations and robustness checks. In this case, contributor variance is *not* "random noise"—it is structured variation associated with unobserved implementation factors (data mixtures, compute, tuning recipes). It is noise only relative to the question "what does architecture explain?" This motivates two dis-

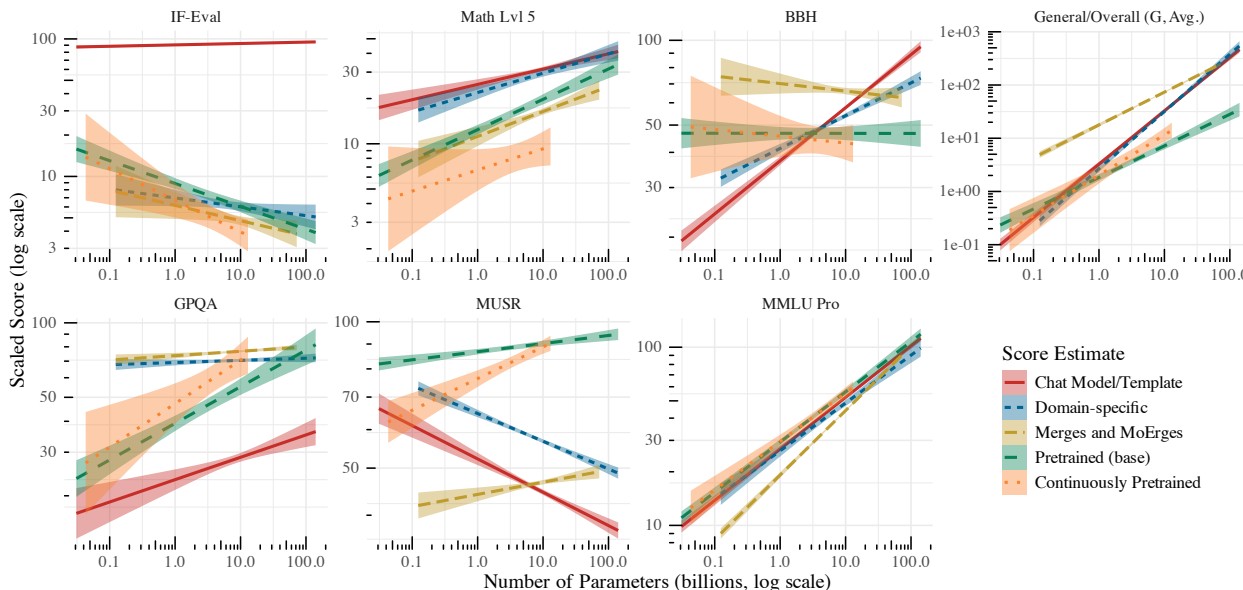

*Figure 4.* Noise-controlled Scaling Laws by Deployment Type: Log-log of Performance by Number of Parameters. Shaded regions indicate the 95% confidence bands for the linear relationship between x and y estimated by robust regression using an M estimator with Tukey's biweight (Venables & Ripley, 2002). In this plot, the HF metadata categorizations of "chat model" and "chat template" are combined ($cm \vee ct$) to better highlight the effects of common SFT practices. Categories are mutually exclusive. Prior to log transformation of the y-axis, each estimated latent score $\Theta_k$ is converted from its assumed distribution to positive values via the cumulative normal distribution function, $S_k = \Phi(\Theta_k)$ before all scores are min-max scaled to the range $s \in S_k : [1 \dots 100]$.

tinct evaluation modes: **Architecture-centric evaluation:** treat provenance as a nuisance facet and report provenance-adjusted uncertainty; and **Method-centric evaluation:** treat provenance as the object of measurement and analyze what practices drive it. Mixed latent regression makes this separation explicit by estimating provenance variance *in latent space*, after controlling for size and deployment.

**Violations of Local dependence** Bootstrapped SEPC maps reveal persistent residual couplings between item pairs even under the best-fitting structures (see § F): shared LLM behaviors unrelated to reported constructs persisted even after accounting for various latent structures, which can threaten the use of benchmarks as measurement instruments.

### 4.3. Scaling laws are a vector, not a scalar

One motivation for leaderboards is to infer scaling effects, and our results indicate that "scaling" is not a single "law" of the ecosystem but a *vector of slopes* over latent abilities. From the mixed-effects covariance decomposition (3), we calculate the signal-to-noise ratio of the effects of size: $\mathrm{SNR}_\beta = \frac{\beta^2}{\sum_S \sigma_{S,1}^2} = 1.12$ and $\mathcal{R}_\beta = 0.53$, suggesting that an average scaling law would be misleading and that scaling is context-dependent (see § G.6). Using (4)-(5), $g$ exhibits the strongest and most stable scaling with log-parameters ($\mathcal{R}_g = 0.97$, Table 3), consistent with the observation that

leaderboard trends track $g$.[9] In contrast, several benchmark-specific factors have negligible size-slopes with deployment and provenance controls. This resolves a common ambiguity in raw-score regressions: apparent scaling of a benchmark can be mediated primarily through $g$ rather than through the specific skill the benchmark purports to isolate. Without size control, signal is buried in confounded noise (see § J.1.1).

**Disentangling Scaling Laws from Ecosystem Noise** By regressing latent abilities on model size, we can isolate a "denoised" scaling relationship from confounding ecosystem factors. Figure 3 shows the estimated scaling laws for the general factor and specific abilities under three views: (i) the raw leaderboard scores, (ii) the basic bifactor latent scores, and (iii) the "denoised" latent scores from the mixed-effects regression. For the general factor, which closely tracks the overall leaderboard average, the bifactor $g$ reveals a scaling relationship that is as strong as the naïve leaderboard trend. After controlling for deployment choices and contributor effects, increasing model size robustly improves general cross-domain capability. The bifactor-only estimate, in contrast, shows a weaker relationship, demonstrating that failing to control for ecosystem noise attenuates the apparent

---

[9]The $\mathcal{R}_\beta$ from Method 2 and $\mathcal{R}_d$ from Method 3 are conceptually similar but not directly exchangeable: the former describes the stability of a manifest-score scaling slope across ecosystem facets and the latter describes the stability of latent-dimension slopes after noise controls.

effect of scale on latent ability.

## 4.4. Insights into specific capabilities and benchmarks

**Does Knowledge Scale and Reasoning Stagnate?** We gain insights into specific factors:

- **General Knowledge (MMLU-Pro):** The denoised scaling effect for the MMLU-specific factor remains strong, adding to the trend for the general factor. This provides rigorous, model-based evidence that suggests that gains on knowledge-intensive benchmarks are heavily driven by scale, even beyond the enhancement of general capability.
- **Complex Reasoning (GPQA and BBH):** In stark contrast, the denoised scaling laws for the reasoning-focused GPQA and BBH factors have trivial effect sizes: after controlling for general capability and ecosystem factors, simply increasing model size yields diminishing returns for these complex reasoning skills.

**Specific Benchmarks: Focus improves usable signal.** Our approach has insights for specific benchmarks.

- **Heterogeneous Reasoning (BBH):** BBH is composed of a diverse set of the 23 most difficult tasks from BIG-bench (Srivastava et al., 2023) and fails to show a relationship with any of the post-training methods captured. Improvements on BBH in this ecosystem are driven by $g$, not a unique benchmark signal, *challenging the practice of hill climbing on benchmarks primarily designed to be difficult*.
- **Clear Targets (IF-Eval and MATH)** By contrast, IF-Eval and Math Level 5 are benchmarks with more clearly defined tasks (verifiable instruction following and math problem solving, respectively), and their effects exhibit high reliability with deployment choices. Whether positive or negative, focused benchmarks give clearer signal.

**Instruction Following and "Soft Reasoning": Evidence of a Cognitive Trade-off?** Striking results emerge from IF-Eval (instruction following) and MuSR (soft reasoning).

- For the **IF-Eval-specific factor**, we find no significant scaling effect attributable to model size after controls. Unique gains on this benchmark appear to be driven almost entirely by specific tuning choices not raw scale.
- For the **MuSR-specific factor**, we observe a significant, albeit small, *negative* relationship with model size in the fully controlled model. Furthermore, several covariates associated with improved instruction-following show a negative effect on the MuSR factor. This suggests a potential "alignment tax" or cognitive trade-off: the very techniques used to make models better instruction-followers may subtly impair their capacity for the kind of flexible, "soft" reasoning required by MuSR tasks.

Interestingly, the most significant SEPC *overfit* across latent

structures occurred between MuSR–IF-Eval items, suggesting a possible inversion of effects of training practices. We interpret these findings as ecosystem-level evidence that current optimization practices (often correlated with scale and deployment) do not uniformly improve all latent dimensions, such as explicit "instruction-following" or implicit "soft reasoning". For benchmarking, the key point is methodological: latent mixed regression can identify where leaderboard gains are driven by $g$ versus where they reflect targeted improvements (or regressions) in specific abilities.

## 4.5. Ranking Changes under $g$ and Ceiling Sensitivity

We compare reported leaderboard percent ranks with percent ranks derived from the precision-weighted means for $g$ (4)-(5) of posterior draws, $\widetilde{\theta}_{i0}$ (see § E.4.5). Overall rankings (Spearman's $\rho = 0.86$) and their broader distribution (Distance Correlation $\mathrm{dCor} = 0.82$) are largely preserved after noise control, but with meaningful localized disruptions (Kendall's $\tau = 0.66$). In particular, only one model in the top 1% of the original overall leaderboard remains in the top 1% after conditioning on $g$, whereas approximately 90% of models remain within the top decile. This is contrasted with the bottom 1% of models: the majority (64%) stayed in the bottom 1%. These findings suggest that top leaderboard positions are substantially sensitive to ecosystem noise, even when the broader ordering of models remains comparatively stable. This directly supports the central claim of the paper: naïve leaderboard rankings conflate underlying capability with ecosystem artifacts, with the largest practical consequences occurring at the decision-relevant margins of the ranking distribution.

## 5. Conclusion

We introduced an advanced framework for AI benchmark ecosystems that systematically identifies, quantifies, and controls for measurement noise. Combining CFA, G-Theory, and latent regression, our analyses demonstrate that major sources of benchmark noise can be identified, quantified, and adjusted for. Collectively, the evidence points toward the strength of human decisions in impacting benchmark performance, whether good or bad: contributor information comprises a significant share of observed variation; instruction-tuning practices work but can also be negatively associated with "soft reasoning"; and benchmark construction approaches distinguish usable signal. By measuring the entire latent landscape, the research community can make benchmark rankings more trustworthy and evaluation claims more credible.

## Impact Statement

A potential positive impact is more transparent and responsible interpretation of model comparisons, scaling claims, and architectural improvements, which may reduce overconfident conclusions drawn from small or unstable leaderboard differences. By providing tools to report uncertainty and disentangle general from benchmark-specific gains, this framework could support better research prioritization and evaluation standards. However, increasing the precision of benchmark analysis may also reinforce leaderboard-centric development or enable more targeted optimization for dominant latent factors while neglecting under-measured abilities such as safety or real-world robustness. Additionally, latent factors are statistical abstractions and could be misinterpreted as definitive constructs of intelligence. Overall, the work seeks to strengthen evaluation rigor, but its benefits depend on careful use and continued attention to broader societal and safety considerations beyond benchmark scores.

**Limitations and scope** Our empirical results are based on a particular leaderboard snapshot and six benchmarks; variance proportions and factor strengths are ecosystem-dependent. The estimates are measured via available observational data and metadata and should not be interpreted causally. The metadata itself is of unknown quality, making contributor/deployment labels noisy proxies. Further, models submitted to Open LLM Leaderboard are not representative of all LLMs. Leaderboards themselves can exhibit temporal instability, as their ecosystems change with time. Latent structural equation model selection is not ontology: bifactor fit supports a parsimonious decomposition of covariance, but the semantics of $g$ and $s_k$ depend on the item pool and evaluation protocol. Bifactor is known to over-extract general factors and absorb local dependence on human data, but it has not been studied whether that is true for LLMs. We leave these open questions and lines of work for future studies.

## Acknowledgments

We thank Mark Wilson for his feedback and support. We are also grateful for the members of Berkeley's BEAR Center, Stanford STAIR Lab, Stanford Psychometrics Lab, and reviewers for constructive feedback.

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

# A. Background and Related Work

AI benchmark ecosystems now serve as de facto scientific instruments: they are used to compare models, to validate training interventions, and to infer scaling laws. Yet the dominant reporting primitive remains a *sum score* (accuracy or normalized accuracy) per benchmark and then an average across benchmarks. This practice implicitly assumes (i) that each benchmark is approximately unidimensional and internally coherent, and (ii) that benchmark totals are commensurate indicators of a smaller number of underlying capabilities. When these assumptions fail, ecosystem conclusions become unstable: model rankings conflate general capability with benchmark-specific artifacts; benchmark overlap inflates perceived evidence; and scaling relationships can be driven by item-selection effects rather than transferable competence.

## A.1. Measurement critiques of leaderboards and benchmark aggregation

Recent work has argued that leaderboard scores embed unvalidated measurement assumptions and can distort comparisons when measurement error and construct mismatch are ignored (Salaudeen et al., 2025; Reuel et al., 2024; Hardy & Kim, 2026). The unidimensionality assumption has been highlighted as particularly consequential: when violated, benchmark totals become confounded composites (Truong et al., 2025), allowing structural misspecification to masquerade as signal (Casabianca, 2025). Our contribution is to make these critiques testable at the ecosystem level by explicitly comparing CFA/SEM structures and quantifying residual dependence. In our setting, local independence corresponds to a diagonal $E$: conditional on $\eta_i$, item residuals should be independent.

**Proposition A.1** (Benchmark totals are not sufficient under multidimensional measurement). *If item responses are generated by* (1) *with* $m > 1$ *and nontrivial loadings on multiple dimensions, then there exist submissions* $i, i'$ *such that (i) they have identical benchmark total scores, yet (ii)* $\eta_i \neq \eta_{i'}$, *implying different capability profiles.*

*Proof.* Let two items load on different factors (non-collinear $\lambda_j$). Construct $\eta_i, \eta_{i'}$ that swap strength across dimensions while preserving marginal correctness probabilities on the benchmark's mixture of items; totals can match while latent vectors differ. Thus totals do not uniquely identify $\eta$. $\square$

## A.2. Psychometric methods for dataset and benchmark evaluation

IRT has been used to estimate item difficulty/discrimination, detect saturated datasets, and improve leaderboard rankings (Lalor et al., 2018; Vania et al., 2021; Rodriguez et al., 2021). Adaptive testing has been proposed to reduce evaluation cost (Maia Polo et al., 2024). These efforts demonstrate the value of item-level modeling but typically assume unidimensionality or focus on within-benchmark properties. Our focus is complementary: we test *ecosystem latent structure* (e.g., bifactor vs. correlated factors) and use SEM diagnostics to localize redundancy and method effects (See §D.1.3) (Muthén, 1993; Saris et al., 2009).

A key limitation of prior exploratory work (e.g., PCA on LLM response matrices) is that it identifies dominant directions of variance without specifying *which* constructs are being measured. In contrast, Confirmatory Factor Analysis (CFA) and Structural Equation Modeling (SEM) are hypothesis-driven: they predict the *dependence structure* among item responses and yield interpretable latent constructs, with explicit constraints corresponding to competing scientific theories of capability organization.

*Remark* A.2 (CFA is not PCA). Unlike PCA, CFA does not merely identify dominant directions of variance. It tests falsifiable hypotheses about dependence structure by encoding benchmark theories as constraints, modeling measurement error explicitly, and testing whether residual covariance remains after conditioning on the latent construct. This is precisely the distinction needed to evaluate whether benchmark composites are valid measures.

**Identification of latent constructs in LLM benchmarking** Several works attempt to infer latent constructs but can produce unstable or poorly aligned factors when i.i.d. assumptions and ecosystem structure are ignored (Kipnis et al., 2024; Burnell et al., 2023). Misalignment between inferred constructs and benchmark intent has also been observed, e.g., Polo et al. find that a construct of general knowledge is "needed" 8x more for Winogrande (commonsense pronoun resolution, (Sakaguchi et al., 2019)) than it is for MMLU-Pro (general knowledge). Our approach addresses these failure modes by (i) testing explicit SEM hypotheses with robustness controls, and (ii) embedding latent measurement within mixed-effects models to correct pseudo-replication.

### A.3. Noise sources and generalizability in benchmark ecosystems

Structured noise in benchmarks has been discussed, but the relative magnitudes of ecosystem noise sources are rarely quantified. Generalizability theory provides a principled decomposition of score variance into crossed facets and their interactions for AI evaluation (Brennan, 2001; Cronbach & Shavelson, 2004; Hardy, 2025a). We adapt this to benchmark ecosystems to estimate conditional ranking reliability under different generalization decisions (e.g., ranking architectures while treating contributors/deployments as nuisance).

### A.4. Scaling laws under confounding

Scaling laws are typically studied under controlled training runs (Kaplan et al., 2020; Hoffmann et al., 2022; Schaeffer et al., 2025; Isik et al., 2026; Sun & Wang, 2026). In open ecosystems, size is confounded with deployment and provenance (training recipes, data mixtures, compute budgets). Our contribution is methodological: we introduce reliability metrics for scaling slopes in a fully crossed mixed-effects setting and, crucially, estimate *capability-specific* scaling in latent space via mixed latent regression (De Boeck, 2011; Chalmers, 2015).

## B. Dataset information

The Hugging Face Open LLM Leaderboard (Fourrier et al., 2024) is a public, large-scale initiative to transparently track, rank, and evaluate open-source LLMs, with 87,939,588 unique response observations. The version we use includes LLM submissions recorded on or before 2025-03-13 ($N = 4{,}287$), according to their submission date. It employs the Eleuther AI Language Model Evaluation Harness to automate standardized scoring across a suite of benchmarks, providing a reproducible and continuously updated snapshot of the field. We employ all six prominent benchmarks in this leaderboard representing a range of cognitive tasks:

- **MMLU-Pro** (Wang et al., 2024) : A challenging extension of the Massive Multitask Language Understanding benchmark (Hendrycks et al., 2020), designed to measure general knowledge and problem-solving skills across a vast array of subjects.
- **IF-Eval** (Zhou et al., 2023): An instruction-following benchmark that assesses a model's ability to adhere to complex and nuanced user directives.
- **BBH** (Suzgun et al., 2022): The Big-Bench Hard benchmark, a collection of tasks identified as being particularly difficult for contemporary LLMs, focusing on multi-step and advanced reasoning.
- **MuSR** (Sprague et al., 2024) : The Multi-Step Soft Reasoning benchmark, which evaluates a model's capacity for reasoning that requires integrating multiple pieces of information without strict logical formality.
- **GPQA** (Rein et al., 2023): A graduate-level question-answering dataset composed of difficult questions written by domain experts, designed to probe deep, specialized knowledge.
- **Open LLM Math Level 5**/"MATH" (Hendrycks et al., 2021): A benchmark testing mathematical problem-solving ability, aligned with high school mathematics curricula (abbreviated in some figures and tables as 'openllm').

Counts of unique items/tasks and LLM models per benchmark are in Table 4.

**Leaderboard Scores** The Open LLM Leaderboard aggregates benchmark scores, used in Method 2, by normalizing raw scores relative to a random baseline and a maximum possible score, then averaging normalized scores. Scores are the leaderboard-reported task metrics (e.g., accuracy, normalized accuracy, exact match), expressed on the leaderboard scale.

**Data Exclusion** We exclude from our analyses in Methods 2 and 3 the six observations with no positive model size (# params$\leq 0$), leaving 4,281 total LLMs. For item set bootstraps in DWLS estimation method of Method 1, items with missingness are not included in the item set. For item set bootstraps in MH-RM estimation Methods 1 and 3, item-LLM-level items are not included if they have no variance across all LLMs (i.e., all LLM responses are equal; for item j, $\sigma^2(x_j) = 0$).

## C. Structural Model Relationships

To determine the latent dimensional structure of the AI benchmark ecosystem, we employ a meta-analytic hypothesis-testing framework using Confirmatory Factor Analysis (CFA). This approach allows us to formally specify and compare a series of competing theoretical models about how different capabilities relate to one another. Given the vast number of items, we use a random-item subsampling procedure to ensure our findings are robust and not artifacts of a specific item selection.

*Table 4.* Per-benchmark item and model counts after dropping missing values

| Benchmark | N(items/tasks) | N(models tested) | Avg. item score | SD item score |
|---|---|---|---|---|
| BBH | 5758 | 4239 | 0.491 | 0.500 |
| GPQA | 1192 | 4237 | 0.301 | 0.459 |
| IF-Eval | 531 | 4240 | 0.418 | 0.493 |
| MMLU-Pro | 11864 | 4240 | 0.340 | 0.474 |
| MuSR | 712 | 4239 | 0.428 | 0.495 |
| Open LLM Math Level 5 | 686 | 4239 | 0.124 | 0.329 |

## C.1. The Confirmatory Factor Analysis Model

CFA is a measurement model within the broader SEM framework that tests hypotheses about the relationships between observed variables (item responses) and unobserved latent variables (factors or constructs). The general CFA model links a $p \times 1$ vector of observed item responses $\mathbf{y}_i$ for a given LLM $i$ to an $m \times 1$ vector of its latent factor scores $\boldsymbol{\eta}_i$ via (1):

$$\mathbf{y}_i = \boldsymbol{\nu} + \boldsymbol{\Lambda}\boldsymbol{\eta}_i + \boldsymbol{\epsilon}_i$$

where $\boldsymbol{\nu}$ is a vector of item intercepts, $\boldsymbol{\Lambda}$ is a $p \times m$ matrix of factor loadings specifying the relationship between items and factors, and $\boldsymbol{\epsilon}_i$ is a $p \times 1$ vector of item-specific residuals. The model-implied covariance matrix of the observed variables, $\boldsymbol{\Sigma}(\boldsymbol{\theta})$, is the target of prediction as shown in Sec. 2.1.1:

$$\boldsymbol{\Sigma}(\boldsymbol{\theta}) = \boldsymbol{\Lambda}\boldsymbol{\Psi}\boldsymbol{\Lambda}^{\top} + \boldsymbol{\Theta} \tag{6}$$

Here, $\boldsymbol{\Psi} = \mathrm{Cov}(\boldsymbol{\eta})$ is the $m \times m$ covariance matrix of the latent factors, and $\boldsymbol{\Theta} = \mathrm{Cov}(\boldsymbol{\epsilon})$ is the $p \times p$ (typically diagonal) covariance matrix of the residuals.[10] The parameters $\boldsymbol{\theta} = \{\boldsymbol{\Lambda}, \boldsymbol{\Psi}, \boldsymbol{\Theta}\}$ are estimated to minimize the discrepancy between $\boldsymbol{\Sigma}(\boldsymbol{\theta})$ and the observed sample covariance matrix $\mathbf{S}$.

For our binary item response data, we use two estimation algorithms to increase the robustness of the findings: one parameterization from the Factor Analysis tradition and another from the Item Response Theory tradition. In the dichotomous cases shown in this study, these parameterization were proved to have mathematical equivalence by (Takane & De Leeuw, 1987). For the former, we use the tetrachoric correlation matrix as $\mathbf{S}$ and the Diagonally Weighted Least Squares (DWLS) estimator (Rosseel, 2012). For the latter, we use a Metropolis-Hastings Robbins-Monro algorithm based on full information likelihood (Chalmers, 2012).

**Separating "structural signal" from flexibility with statistical comparisons and randomized permutation controls**
More flexible structural equation models can fit better even when benchmark assignment contains no information. In addition to the tests described next, we resolve the statistical question of these models by analyzing traditional covariance prediction measures, likelihood information criteria, out-of-sample prediction (which in CFA and structural equation modeling is unusual due to the statistical objective), and out-of-sample latent score reliabilities. A bootstrap iteration-wise average percent rank overview of these is in Table 6. These far exceed typical reporting requirements, and we report all of these to demonstrate that, while imperfect and prone to overfit, the bifactor model is the most statistically justified latent structure. To quantify further the extent to which "benchmark structure" drives fit improvements, we add a *within-replication permutation* condition. For each bootstrap replication $b$, we additionally construct a randomized mapping $\pi^{(b)} : \tilde{\mathcal{J}}^{(b)} \to \{1, \ldots, K\}$ that preserves the per-benchmark counts $\{r_k\}$ but shuffles which items are labeled as belonging to which benchmark. We then refit each model under the permuted mapping, with randomization acting as a "treatment". Let $T(\cdot)$ be any fit statistic (e.g., scaled RMSEA). For a given structure $s$ and replication $b$, define $\Delta T_s^{(b)} = T(s \mid \pi^{(b)}) - T(s \mid \mathrm{id})$, where id denotes the true benchmark assignment. Large positive $\Delta T$ indicates that the true assignment carries structural information exploited by $s$ beyond generic flexibility.

## C.2. Detailed Overview of Competing Structural Models

We test six increasingly complex models, depicted in Figure 2 and detailed in Table 1. Each represents a distinct hypothesis about the organization of LLM abilities (Table 1), spanning dominant hypotheses implicit in current benchmark practice.

---

[10]To support readers navigating other literature, for the remainder of this appendix, we will use the traditional notation: $\boldsymbol{\Theta} = \mathrm{Cov}(\boldsymbol{\epsilon})$.

Each model is a constraint set on $(\boldsymbol{\Lambda}, \boldsymbol{\Phi}, \boldsymbol{\Theta})$; importantly, models differ not only in parameter count but in whether benchmark identity is treated as a substantive construct, a residual grouping, or an artifact after partialing out $g$. Critically, these models represent different strategies for partitioning measurement variance into signal (latent factors) and noise (error terms $\boldsymbol{\Theta}$). Below are conceptual descriptions of the six structural models evaluated in this study (see Table 1 for formal specifications). Each model represents a distinct hypothesis about the nature of signal and noise in the benchmark ecosystem.

### C.2.1. MODEL 1: MULTI-UNIDIMENSIONAL INDEPENDENT BENCHMARK FACTORS MODEL

This model reflects the naive assumption of many leaderboards. It hypothesizes that each benchmark measures a completely distinct latent ability, and these abilities are uncorrelated. All cross-benchmark correlation is assumed to be zero. This model (see (a) Figure 2) assumes that each benchmark $k$ measures a distinct latent factor $f_k$ that is uncorrelated with all other benchmark factors. It serves as a baseline for assessing the presence of inter-benchmark correlations.

$$y_{ikj} = \lambda_{kj} f_{ik} + \epsilon_{ikj}, \quad \text{with} \quad \text{Cov}(f_{ik}, f_{ik'}) = 0 \text{ for } k \neq k' \tag{7}$$

This represents a modular view of AI capabilities, where proficiency in one domain (e.g., math) is entirely independent of proficiency in another (e.g., instruction following).

### C.2.2. MODEL 2: UNIDIMENSIONAL, SINGLE GENERAL FACTOR MODEL

The most parsimonious model. It posits that all systematic shared variance among all items across all benchmarks is explained by a single, monolithic latent ability. Any remaining variance is treated as item-specific measurement error (noise). This model (see (b) Figure 2) tests the hypothesis that a single latent factor, $g$, accounts for all non-error covariance among all items across all benchmarks. It represents the most parsimonious explanation of performance.

$$y_{ikj} = \lambda_{kj} g_i + \epsilon_{ikj} \tag{8}$$

where $\lambda_{kj} \in \mathbb{R}$ is the loading of item $j$ from benchmark $k$ on the general factor $g_i \sim \mathcal{N}(0, 1)$, and $\epsilon_{ikj} \sim \mathcal{N}(0, \theta_{kj})$ is the item-specific residual variance. This model posits that all benchmarks are, in effect, measuring the same underlying construct.

### C.2.3. MODEL 3: HIERARCHICAL SECOND-ORDER FACTOR MODEL

This model introduces a more structured explanation for the correlations among factors. It proposes that a higher-order general factor ($g$) gives rise to the first-order benchmark factors. Here, $g$ does not influence items directly but acts through the specific benchmark abilities. This model (see (c) Figure 2) proposes a hierarchical structure where a higher-order general factor, $g$, explains the correlations among the first-order benchmark factors from Model 3.

$$y_{ikj} = \lambda_{kj} f_{ik} + \epsilon_{ikj} \tag{9}$$
$$f_{ik} = \gamma_k g_i + \delta_{ik} \tag{10}$$

Here, the first-order factor $f_{ik}$ is regressed on the second-order factor $g_i$, with $\gamma_k$ being the loading and $\delta_{ik}$ the first-order factor disturbance. This structure implies that the general factor's influence on items is indirect, mediated entirely through the benchmark-specific factors.

### C.2.4. MODEL 4: MULTIDIMENSIONAL CORRELATED BENCHMARK FACTORS MODEL

A more realistic version of the independent factors model. It posits distinct latent abilities for each benchmark but allows these abilities to be freely correlated. For example, it can model the idea that math ability and reasoning ability are separate but related constructs. This is a standard CFA model (see (d) Figure 2) that relaxes the strict independence assumption of Model 2. Each item loads onto its designated benchmark factor, but the benchmark factors are allowed to covary.

$$y_{ikj} = \lambda_{kj} f_{ik} + \epsilon_{ikj}, \quad \text{with} \quad \text{Cov}(\mathbf{f}_i) = \boldsymbol{\Psi} \tag{11}$$

where $\mathbf{f}_i$ is the vector of latent factor scores for LLM $i$, and $\boldsymbol{\Psi}$ is a symmetric correlation matrix with 1s on the diagonal. The off-diagonal elements $\phi_{kk'}$ quantify the degree of relationship between distinct benchmark abilities.

### C.2.5. MODEL 5: THE BIFACTOR MODEL

A fundamentally different structure. It posits that every item's response is influenced by two independent sources of signal: a general factor ($g$) that affects all items, and a benchmark-specific factor that affects only items within that benchmark. The key assumption is that $g$ and the specific factors are uncorrelated (orthogonal). This model is ideal for testing if benchmark-specific variance is merely a residual grouping effect after accounting for general ability. The bifactor model (see (e) Figure 2, Code I.1) (Dunn & McCray, 2020) provides an alternative explanation for shared variance. It posits that each item's variance is simultaneously explained by a general factor $g$ (common to all items) and a benchmark-specific factor $s_k$ (common only to items within that benchmark). The factors are constrained to be orthogonal.

$$y_{ikj} = \lambda_{kj}^{(g)} g_i + \lambda_{kj}^{(s)} s_{ik} + \epsilon_{ikj} \tag{12}$$

where $\text{Cov}(g_i, s_{ik}) = 0$ and $\text{Cov}(s_{ik}, s_{ik'}) = 0$ for $k \neq k'$. This model cleanly partitions item variance into a general component and a specific component, allowing for a direct assessment of how much unique information each benchmark provides after accounting for general ability. While it is not likely that the bifactor model truly represents human latent factor organization (as capabilities are not likely orthogonal), this structure is not yet tested on the latent capability organization for AI models. The code for the model specifications for the bifactor model, as represented in Methods 1 and 3, are found in Appendix I.

### C.2.6. MODEL 6: MULTIDIMENSIONAL/CORRELATED BIFACTOR MODEL

A relaxation of the orthogonal bifactor model. It maintains the direct influence of $g$ on all items but allows the specific factors to be correlated with each other. This tests whether there is systematic residual covariance among benchmarks even after partialing out the main general factor.

This model (see (f) Figure 2) is a relaxation of the bifactor model, allowing the specific factors $s_k$ to correlate with each other while remaining orthogonal to the general factor $g$.

$$y_{ikj} = \lambda_{kj}^{(g)} g_i + \lambda_{kj}^{(s)} s_{ik} + \epsilon_{ikj}, \quad \text{with} \quad \text{Cov}(g_i, \mathbf{s}_i) = \mathbf{0} \text{ and } \text{Cov}(\mathbf{s}_i) = \mathbf{\Psi} \tag{13}$$

where $\mathbf{\Psi}$ is the correlation matrix of the specific factors. This model is useful for testing whether any residual covariance between benchmark domains remains after partialing out a broad general factor, which may be attributable to shared methods or sub-domain content.

In actuality, the more complicated model may be the best representation. If bootstrapped item set sizes were increased (e.g., 500 items per benchmark sampled), the more complex and expressive model would likely fit better. However, this does not mean that this is a more likely "structural truth". This would be a statistical artifact of a high-noise multidimensional system being explained by a higher capacity model. A key idea in comparing latent structural models is harmonizing the fit with explainable parsimony.

### C.3. Traditional Model Evaluation and Comprehensive Comparison Framework

We evaluate and compare the six models using a combination of global fit indices, formal hypothesis tests, and parameter-level diagnostics to build a convergent body of evidence.

### C.3.1. TRADITIONAL MEASURES OF FIT

In CFA, traditional measures of fit assess how well a model's implied covariance matrix $\mathbf{\Sigma}(\boldsymbol{\theta})$ reproduces the observed sample covariance matrix $\mathbf{S}$, we use a standard set of fit indices in addition to our more robust metrics and methods.

- **Comparative Fit Index (CFI)** and **Tucker-Lewis Index (TLI)**: These incremental fit indices measure the improvement in fit of a target model over a null (independence) model. Values $\geq .95$ are typically considered to indicate good fit (Medsker et al., 1994).
- **Standardized Root Mean Square Residual (SRMR)**: This index represents the average standardized difference between the observed and predicted correlations. Values $\leq .08$ are considered indicative of good fit.
- **Root Mean Square Error of Approximation (RMSEA)**: This index estimates the model misfit per degree of freedom, with smaller values indicating better fit. Values $\leq .06$ have traditionally been used to classify a close fit to the data (Medsker et al., 1994; Kenny et al., 2015).

**Weaknesses of Traditional Measures**    Traditional measures of fit are expansive, but share weakness. For example, RMSEA, while arguably one of the best options for fit, is sensitive to factors such as sample size, number of indicators, and data characteristics (Kenny et al., 2015; Groskurth et al., 2024). Fixed cutoff values like 0.06 derived from simulation studies may not be universally applicable across diverse real-world settings (McNeish & Wolf, 2023). With $n \approx 4000$, $p \geq 30$, as in the present study, RMSEA variants can become sensitive to trivially small covariance residuals. Traditional cutoffs were calibrated on $n \in [200, 500]$ with $p < 30$ (Remark D.2). At our scale, minor misspecification can inflate RMSEA while leaving substantive conclusions intact.

Thankfully, our model choice does not rest on covariance RMSEA; we establish practical and statistical fit with our robustness checks, such as out-of-sample AUC and MAE (Tables 5 and 6), Lagrangian Multiplier tests (§D), permutation controls, §F, etc. Table 6 shows bifactor-family models dominate across traditional criteria (e.g., CFI, SRMR, BIC, log-likelihood) in within-bootstrap percent-rank comparisons estimated with DWLS. Tab 8 confirms that the bifactor significantly improves over every simpler structure in 100% of bootstraps via LRTs.

*Table 5.* **Latent Structure Measurement Comparison Overview:** Iteration-wise Median value across Bootstraps by Metric

| | Fit Indices / Covariance Fit | | | | Test Accuracy | | Latent Factor Test Set Reliability Estimates | | | | | | |
|---|---|---|---|---|---|---|---|---|---|---|---|---|---|
| Structure | RMSEA | CFI | TLI | SRMR | AUC | MAE | G | IF-EVAL | Math Lvl. 5 | MMLU-Pro | BBH | GPQA | MUSR |
| indepfact | 0.09 | 0.31 | 0.28 | 0.22 | 0.90 | 0.25 | | *0.84* | **0.83** | 0.69 | 0.62 | 0.29 | 0.32 |
| gfact | 0.06 | 0.70 | 0.69 | 0.14 | 0.87 | 0.28 | **0.89** | | | | | | |
| hier2ord | 0.05 | 0.76 | 0.75 | 0.13 | 0.89 | 0.25 | *0.88* | *0.84* | 0.77 | 0.59 | 0.48 | **0.30** | *0.29* |
| corrfact | 0.05 | 0.76 | 0.75 | 0.13 | 0.55 | 0.25 | | **0.85** | **0.83** | **0.72** | **0.65** | 0.24 | 0.19 |
| bifact | *0.05* | *0.80* | *0.78* | *0.12* | **0.91** | **0.24** | 0.85 | 0.68 | 0.62 | 0.29 | 0.25 | *0.29* | *0.29* |
| corrbifact | **0.04** | **0.86** | **0.84** | **0.10** | 0.58 | 0.24 | 0.82 | 0.70 | 0.63 | 0.40 | 0.33 | 0.23 | 0.18 |

**Fit Statistics for Hypothesized Latent Structures: aggregated by median across bootstraps**. All metrics were ordered so higher percent ranks correspond with more desirable structural fit. RMSEA, CFI, and TLI all use their scaled and standardized versions which better support comparisons in bootstraps. The RMSEA values differ from those in Table 1 to be comparable with the other traditional metrics (e.g., TLI, SRMR), because these were only estimated with DWLS.

*Table 6.* **Latent Structure Measurement Comparison Overview:** Average Iteration-wise Percent Rank across Bootstraps by Metric

| | Fit Indices / Covariance Fit | | | | Information Criteria | | | Test Accuracy | | Latent Factor Test Set Reliability Estimates | | | | | | |
|---|---|---|---|---|---|---|---|---|---|---|---|---|---|---|---|---|
| Structure | RMSEA | CFI | TLI | SRMR | AIC | BIC | log Lik. | AUC | MAE | G | IF-EVAL | Math Lvl. 5 | MMLU-Pro | BBH | GPQA | MUSR |
| indepfact | 0.7 | 0.7 | 0.7 | 0.7 | 8.8 | 8.8 | 8.8 | 76.5 | 41.6 | | *74.0* | **80.2** | **74.3** | *66.8* | *56.5* | **60.2** |
| gfact | 25.6 | 25.7 | 25.9 | 24.6 | 10.9 | 10.9 | 10.9 | 38.2 | 0.0 | **86.0** | | | | | | |
| hier2ord | 50.4 | 45.0 | 50.3 | 40.3 | 41.5 | 41.5 | 41.5 | 59.5 | 38.7 | 54.9 | 51.3 | 53.0 | 56.8 | 52.4 | 35.3 | 37.9 |
| corrfact | 53.9 | 55.7 | 54.1 | 54.0 | 56.8 | 56.8 | 56.8 | 19.1 | 37.9 | | **83.8** | *78.6* | 69.6 | **76.3** | *54.9* | *55.3* |
| bifact | 75.7 | 75.7 | 75.6 | 75.8 | *79.4* | *79.4* | *79.4* | **94.5** | 86.5 | 40.5 | 18.9 | 14.2 | 24.0 | 21.4 | 50.7 | 48.3 |
| corrbifact | **99.1** | **98.8** | **98.8** | **99.2** | **100.0** | **100.0** | **100.0** | 44.9 | **93.1** | 20.5 | 22.4 | 25.1 | 28.1 | 34.1 | 46.4 | 43.3 |

**Fit Statistics for Hypothesized Latent Structures: aggregated across bootstraps by within bootstrap percent rank**. All metrics were ordered so higher percent ranks correspond with more desirable structural fit. RMSEA, CFI, and TLI all use their scaled and standardized versions which better support comparisons in bootstraps.

# D. Mathematical Set-up for Testing Local Dependence

## D.1. Mathematical Foundations for Score Tests and Local Dependence

### D.1.1. MODIFICATION INDICES AS LAGRANGE-MULTIPLIER DIAGNOSTICS FOR LOCAL DEPENDENCE

We formalize the common structural equation modeling "modification index" (MI) as a score/Lagrange-multiplier test and show how it quantifies (i) *violations of local dependence* (unmodeled residual dependence between items) and, more generally, (ii) *subject-dependent error* (unmodeled heterogeneity that induces residual correlations after conditioning on the latent variables).

**Setup (latent-response CFA).**    Let $\mathbf{y}_i \in \{0, 1\}^p$ be item responses for model/subject $i \in \{1, \ldots, n\}$ based on Eq. (1), generated via an underlying continuous response $\mathbf{y}_i^\star \in \mathbb{R}^p$:

$$\mathbf{y}_i^\star = \boldsymbol{\nu} + \boldsymbol{\Lambda}\boldsymbol{\eta}_i + \boldsymbol{\epsilon}_i, \ \boldsymbol{\eta}_i \sim \mathcal{N}(\mathbf{0}, \boldsymbol{\Phi}), \ \boldsymbol{\epsilon}_i \sim \mathcal{N}(\mathbf{0}, \boldsymbol{\Theta}),$$

with $y_{ij} = \mathbb{1}[y_{ij}^\star > \tau_j]$. In the standard CFA identification, $\boldsymbol{\Theta}$ is constrained to be diagonal. The implied covariance, defined in § 2.1.1, of $\mathbf{y}^\star$ is $\boldsymbol{\Sigma}(\boldsymbol{\theta}) = \boldsymbol{\Lambda\Phi\Lambda}^\top + \boldsymbol{\Theta}$, $\boldsymbol{\theta}$ collects all free parameters.

Let $\mathbf{S}$ be an estimate of the population correlation/covariance of $\mathbf{y}^\star$ (e.g., tetrachoric correlation matrix). Estimation proceeds by minimizing a discrepancy $F(\boldsymbol{\Sigma}(\boldsymbol{\theta}), \mathbf{S})$ (ML, WLSMV, etc.), equivalently maximizing a log-likelihood $\ell(\boldsymbol{\theta})$.

**Local dependence as a constrained parameter.** Pick a candidate residual covariance between items $(a, b)$, denoted

$$\psi \equiv \mathrm{Cov}(\epsilon_a, \epsilon_b) = \Theta_{ab}.$$

The standard model imposes the constraint $H_0 : \psi = 0$. Define the augmented parameter vector $\tilde{\boldsymbol{\theta}} = (\boldsymbol{\theta}, \psi)$ and the constrained optimum

$$\hat{\boldsymbol{\theta}} \in \arg\max_{\boldsymbol{\theta}} \ell(\boldsymbol{\theta}, \psi = 0).$$

### D.1.2. MI AS A LAGRANGE-MULTIPLIER / SCORE TEST

**Theorem D.1** (Modification index as an LM statistic). *Assume $\ell(\tilde{\boldsymbol{\theta}})$ is twice continuously differentiable in a neighborhood of the true parameter and that standard regularity conditions for score tests hold (listed below). Let the score for $\psi$ at the constrained optimum be*

$$U_\psi(\hat{\boldsymbol{\theta}}) \equiv \left. \frac{\partial\ell(\boldsymbol{\theta}, \psi)}{\partial\psi} \right|_{(\boldsymbol{\theta}, \psi) = (\hat{\boldsymbol{\theta}}, 0)}.$$

*Let $\mathcal{I}(\tilde{\boldsymbol{\theta}})$ be the expected Fisher information for $\tilde{\boldsymbol{\theta}}$ and write its block decomposition*

$$\mathcal{I} = \begin{bmatrix} \mathcal{I}_{\theta\theta} & \mathcal{I}_{\theta\psi} \\ \mathcal{I}_{\psi\theta} & \mathcal{I}_{\psi\psi} \end{bmatrix}.$$

*Define the* partial *information (Schur complement)*

$$\mathcal{I}_{\psi\psi\cdot\theta} \equiv \mathcal{I}_{\psi\psi} - \mathcal{I}_{\psi\theta}\mathcal{I}_{\theta\theta}^{-1}\mathcal{I}_{\theta\psi}.$$

*Then the Lagrange-multiplier (score) statistic for testing $H_0 : \psi = 0$ is*

$$\mathrm{MI}(\psi) = U_\psi(\hat{\boldsymbol{\theta}})^\top \mathcal{I}_{\psi\psi\cdot\theta}(\hat{\boldsymbol{\theta}})^{-1} U_\psi(\hat{\boldsymbol{\theta}}) \xrightarrow{d} \chi_1^2.$$

*Moreover, $\mathrm{MI}(\psi)$ is locally (asymptotically) most powerful among tests of size $\alpha$ for alternatives $\psi = O(n^{-1/2})$ (Buse, 1973; Engel, 1984).*

*Proof sketch.* The KKT conditions for constrained maximization yield $\hat{\lambda} = -U_\psi(\hat{\boldsymbol{\theta}})$: the Lagrange multiplier equals (minus) the score for the constrained parameter. Normalizing by the Schur complement $\mathcal{I}_{\psi\psi\cdot\theta}$ yields the stated quadratic form. Under $H_0$ and regularity, $n^{-1/2}U_\psi(\hat{\boldsymbol{\theta}}) \Rightarrow \mathcal{N}(0, \mathcal{I}_{\psi\psi\cdot\theta})$, so the quadratic form converges to $\chi_1^2$. Local asymptotic optimality follows from standard LAN arguments (Buse, 1973). $\square$

**Statistical conditions (typical SEM/ML assumptions).** A sufficient set of conditions for Theorem D.1 (and its robust analogs) includes:

1. **Identification:** the constrained model is locally identified at the true parameter; $\mathcal{I}_{\theta\theta}$ is nonsingular.
2. **Interior point:** $\psi = 0$ lies in the interior of the parameter space for the relevant parametrization (or use boundary-corrected tests if not).
3. **Regularity/smoothness:** $\ell$ is twice continuously differentiable in a neighborhood; interchange of differentiation and integration is valid.
4. **Asymptotics:** $n \to \infty$ with $p$ fixed (or $p$ increasing slowly under additional conditions); $\mathbf{S}$ is consistent for the population covariance of $\mathbf{y}^\star$.

*Remark* D.2 (Finite-sample performance). Simulation studies show that MI statistics maintain approximately correct Type I error rates for $n \geq 200$ and moderate model complexity ($p \leq 50$), with scaled corrections essential for categorical data (Saris et al., 2009; Kenny et al., 2015; McNeish & Wolf, 2023; Groskurth et al., 2024). In our bootstrap analysis, each replication uses $n \approx 3{,}000$ LLM submissions, well above the threshold for asymptotic validity.

D.1.3. MI QUANTIFIES VIOLATIONS OF LOCAL DEPENDENCE

We now connect the MI for $\psi = \Theta_{ab}$ to the *local dependence* principle.

**Definition D.3** (Local independence). Items are locally independent given the latent variables $\boldsymbol{\eta}$ if

$$p(\mathbf{y} \mid \boldsymbol{\eta}) = \prod_{j=1}^{p} p(y_j \mid \boldsymbol{\eta}).$$

In the Gaussian latent-response CFA above, local independence is equivalent to $\boldsymbol{\Theta}$ being diagonal.

**Proposition D.4** (Residual covariance as a measure of local dependence). *Under the latent-response CFA, if $\Theta_{ab} \neq 0$ then $y_a \not\perp\!\!\!\perp y_b \mid \boldsymbol{\eta}$ (local dependence fails). Conversely, if $\boldsymbol{\Theta}$ is diagonal, then $y_a \perp\!\!\!\perp y_b \mid \boldsymbol{\eta}$ for all $a \neq b$.*

*Proof.* For multivariate Gaussians, conditional independence holds iff the corresponding covariance is zero. Thus $\Theta_{ab} \neq 0$ implies dependence between $y_a^\star$ and $y_b^\star$ given $\boldsymbol{\eta}$, which transfers to the binary items. $\square$

**Interpretation of MI.** In this setting, $\mathrm{MI}(\Theta_{ab})$ is precisely a test for whether the data demand a nonzero residual dependence between items $a$ and $b$ *after* accounting for the latent structure in $\boldsymbol{\Lambda}\boldsymbol{\Phi}\boldsymbol{\Lambda}^\top$. The larger the MI, the stronger the evidence that the current factor structure under-explains (or over-explains) the empirical association between $(a, b)$. Geometrically, it measures the instantaneous rate of improvement in log-likelihood if constraint $(a, b)$ were relaxed by an infinitesimal amount. In information-geometric terms, it quantifies the projection of the unconstrained score onto the constraint manifold.

D.1.4. MI DETECTS SUBJECT-DEPENDENT ERROR AS INDUCED RESIDUAL DEPENDENCE

Local dependence is not only "item redundancy." It can also arise because the error is *subject-dependent*: different subjects/models induce different item difficulties or different effective loadings, producing residual covariance when the model incorrectly assumes homogeneity.

**A concrete heterogeneity model.** Let $z_i \in \mathbb{R}$ be an unmodeled subject-specific variable (e.g., a latent "format adherence" trait, prompting style, or evaluator-specific artifact) independent of $\boldsymbol{\eta}_i$, and suppose the true latent response model is

$$\mathbf{y}_i^\star = \boldsymbol{\nu} + \boldsymbol{\Lambda}\boldsymbol{\eta}_i + \mathbf{b}\, z_i + \boldsymbol{\xi}_i,$$

$$z_i \sim \mathcal{N}(0, \sigma_z^2), \; \boldsymbol{\xi}_i \sim \mathcal{N}(\mathbf{0}, \boldsymbol{\Theta}_0) \text{ diagonal.}$$

Then the *marginal* covariance of $\mathbf{y}^\star$ is

$$\mathrm{Cov}(\mathbf{y}^\star) = \boldsymbol{\Lambda}\boldsymbol{\Phi}\boldsymbol{\Lambda}^\top + \underbrace{\mathbf{b}\mathbf{b}^\top \sigma_z^2}_{\text{induced residual dependence}} + \boldsymbol{\Theta}_0.$$

If we fit the misspecified CFA that omits $z_i$ and forces residuals diagonal, the term $\mathbf{b}\mathbf{b}^\top \sigma_z^2$ cannot be represented by $\boldsymbol{\Theta}$ and will appear as systematic misfit. In particular,

$$\Theta_{ab}^{\text{true, marginal}} = b_a b_b \sigma_z^2 \neq 0$$

whenever $b_a b_b \neq 0$, inducing residual covariance among the affected items.

**Corollary D.5** (MI detects unmodeled subject heterogeneity through residual covariances). *Under the heterogeneity model above, for any pair $(a, b)$ with $b_a b_b \sigma_z^2 \neq 0$, the score $U_{\Theta_{ab}}(\hat{\boldsymbol{\theta}})$ is generically nonzero in the diagonal-residual CFA, and $\mathrm{MI}(\Theta_{ab})$ diverges at rate $O_p(n)$, yielding asymptotic power 1 to detect the induced local dependence.*

*Proof.* The fitted model cannot match the population covariance entry $b_a b_b \sigma_z^2$ when forcing $\Theta_{ab} = 0$, producing a nonzero population score. By M-estimation theory, MI grows at rate $O_p(n)$, yielding consistency. $\square$

*Remark* D.6. Corollary D.5 is the key bridge to "subject-dependent error" in benchmark ecosystems: if some models systematically exploit (or fail on) a shared artifact (formatting, instruction parsing, choice biases), that unmodeled trait acts like $z_i$ and induces a low-rank residual covariance pattern among the affected items. MIs will concentrate on item pairs within that pattern—even if the original factor structure is otherwise correct—flagging a *measurement failure mode* rather than merely "redundant questions."

D.1.5. EFFECT-SIZE DIRECTION VIA EXPECTED PARAMETER CHANGE (EPC)

Beyond hypothesis testing, practitioners want an approximate *magnitude and sign* of the violation. Under a second-order approximation, freeing $\psi$ yields

$$\widehat{\Delta\psi} \approx \mathrm{EPC}(\psi) = \mathcal{I}_{\psi\psi\cdot\theta}(\hat{\boldsymbol{\theta}})^{-1} U_\psi(\hat{\boldsymbol{\theta}}).$$

In particular:

- $\mathrm{EPC}(\Theta_{ab}) > 0$ indicates the model underpredicts the residual association of items $(a, b)$ after conditioning on the factors (shared method/content, induced heterogeneity); the items share more dependence than the latent factors $\boldsymbol{\eta}$ can explain.
- $\mathrm{EPC}(\Theta_{ab}) < 0$ indicates overprediction (the factor structure implies stronger association than observed), often signaling cross-loading structure, suppressor effects, or miscoded item polarity.

The **Standardized EPC (SEPC)** by implied residual variances places this on a correlation scale, making it interpretable as the estimated residual correlation, as used in our redundancy analyses.

D.1.6. SUMMARY: WHAT MI CERTIFIES AND WHAT IT DOES NOT

- **What MI is:** a Lagrange-multiplier/score statistic measuring the (curvature-normalized) gradient pressure to relax a constraint. For $\Theta_{ab} = 0$, it is a direct diagnostic for conditional dependence remaining after the latent structure.
- **What MI detects in specific:** (i) local dependence from item overlap effects, and (ii) residual dependence induced by unmodeled subject heterogeneity or other misspecification that manifests as correlated errors.
- **What MI detects across bootstraps:** (i) propensities in local dependence, and (ii) trends in residual dependence induced by unmodeled subject heterogeneity or other misspecification that manifests as correlated errors.
- **What MI does not certify:** a causal explanation. A large MI says "this constraint is wrong for the dependence structure," not "this pair is semantically redundant." Pairwise disambiguation can require substantive inspection or modeling the alternative mechanism (e.g., adding a method factor versus correlating residuals, bootstrapping for generalization).

This formal view justifies our use of MI/SEPC aggregated over bootstrap replications: it yields a statistically principled, computationally efficient, and interpretable map of *where* the benchmark ecosystem violates local independence, and whether these violations are stable enough to be treated as structural (rather than idiosyncratic) properties of the item pool.

# E. Robustness via Meta-Analytic Bootstrapping

## E.1. Bootstrapped Methods 1 and 3 and Aggregations of Statistics

Fitting models to the thousands of available items is computationally intractable and risks overfitting to item-specific idiosyncrasies. To ensure our conclusions are stable and generalizable, we implement a meta-analytic bootstrapping procedure. For each of $B = 400$ replications, we:

1. Randomly sample a small, computationally manageable set of items from each of the 6 benchmarks.
2. Fit all six structural models to the tetrachoric correlation matrix of this item subset.
3. Store key model fit indices (e.g., scaled RMSEA, CFI), parameter estimates ($\boldsymbol{\Lambda}$, $\boldsymbol{\Psi}$), and local-fit diagnostics (modification indices).

This process yields distributions of fit indices and parameters, allowing us to perform a meta-analysis. We assess the central tendency and stability of each model's performance, effectively integrating out the noise from any single item sample. To formally compare models, we fit a mixed-effects regression to the stacked results from all bootstrap replications, predicting fit indices from model structure while accounting for between-replication variance.

**Bootstrap-by-item design (computational tractability and robustness)** Let benchmarks be indexed by $k \in \{1, \dots, K\}$, with item sets $\mathcal{J}_k$ and $|\mathcal{J}_k| = p_k$. In bootstrap replication $b \in \{1, \dots, B\}$ we sample (without replacement) a subset $\tilde{\mathcal{J}}_k^{(b)} \subset \mathcal{J}_k$ of size $r_k \ll p_k$ for each benchmark and pool to $\tilde{\mathcal{J}}^{(b)} = \cup_k \tilde{\mathcal{J}}_k^{(b)}$ with $\tilde{p} = \sum_k r_k$. We then fit all candidate models to the same sampled item pool. This design targets a population quantity: the expected comparative fit of each structural hypothesis under the distribution of items induced by the benchmark ecosystem. It also provides a principled stability check: a structural conclusion is credible only if it persists across many item-resampled realizations.

E.1.1. MOTIVATION FOR META-AGGREGATIONS

Each bootstrap replication $b \in \{1, \ldots, B\}$ samples a different item subset $\tilde{\mathcal{J}}^{(b)} = \cup_k \tilde{\mathcal{J}}_k^{(b)}$, with $|\tilde{\mathcal{J}}_k^{(b)}| = r_k \ll p_k$ items per benchmark. The MH-RM algorithm (Cai, 2010a) then returns, for each estimand $\xi$ of interest, a point estimate $\hat{\xi}^{(b)}$ together with a standard error $\mathrm{SE}^{(b)}(\hat{\xi})$ derived from the observed information matrix at convergence. Two properties of this design make naïve averaging ($\bar{\xi} = B^{-1} \sum_b \hat{\xi}^{(b)}$) statistically inefficient:

1. **Heterogeneous precision.** Different item draws yield different effective test lengths, item discrimination profiles, and conditioning on the latent trait. A bootstrap that happens to sample highly discriminating items produces a more precise estimate of $\beta_d$ than one dominated by low-discrimination items. Treating both equally wastes information.

2. **Occasional near-degeneracy.** Some item subsets may produce near-singular Fisher information for particular parameters (e.g., a specific factor receiving very few high-discrimination items), inflating $\mathrm{SE}^{(b)}$ by orders of magnitude. Naïve averaging allows these unstable estimates to contaminate the aggregate.

Inverse-variance weighting addresses both issues: it automatically downweights imprecise estimates in proportion to their squared standard error, yielding the minimum-variance linear unbiased combination under independence.

## E.2. Summary of the aggregation pipeline

The complete pipeline, applied for Methods 1 and 3, proceeds as follows:

1. **Sample.** Draw item subset $\tilde{\mathcal{J}}^{(b)}$ with $r_k$ items per benchmark $k$, without replacement.

2. **Fit.** Estimate the target model (CFA structures for Method 1 via both MH-RM and DWLS; bifactor IRT with/without latent regression for Method 3 via MH-RM), obtaining estimates $\hat{\xi}^{(b)}$ and standard errors $\mathrm{SE}^{(b)}$.

3. **Weight and Aggregate via Mixed Effect Meta-regression (Method 1).** Obtain statistical estimates based on (14).

4. **Weight (Method 3).** Compute $w^{(b)} = 1/(\mathrm{SE}^{(b)})^2$, applying fallback rules (18) for degenerate cases.

5. **Aggregate (Method 3).** Form IVW point estimates $\tilde{\xi}$ via (16) and standard errors via (17).

6. **Diagnose.** Compute between-bootstrap heterogeneity $\hat{\tau}^2$ and, for scaling slopes, the reliability $\mathcal{R}_d$ via (23).

This design ensures that final estimates reflect the most informative item draws, propagate both within- and between-bootstrap uncertainty, and provide built-in diagnostics for the stability of every reported quantity.

## E.3. Meta-analytic aggregation of bootstrapped estimates via mixed-effects regression

For more robustness when evaluating statistics and out-of-sample prediction for Method 1, we estimate these relationships using two different estimation techniques: the *full information* stochastic *expectation maximizing* Metropolis-Hastings Robbins-Monro (MH-RM) algorithm (Cai, 2010b; Chalmers, 2015) and the robust *limited information likelihood maximizing* Diagonal Weighted Least Squares (Rosseel, 2012) (See Appendix I for further estimation details). Each fitted model yields robust fit indices (we use scaled/robust variants because item responses are non-Gaussian). Let $t_{b,s,r}$ denote a fit statistic from replication $b$, structure $s$, and condition $r \in \{0, 1\}$ (true vs. randomized assignment). We summarize across replications using a mixed-effects meta-analytic regression:

$$t_{b,s,r} = \alpha_s + \beta_s \, r + u_b + v_b \, r + \varepsilon_{b,s,r}, \tag{14}$$

where $u_b$ captures bootstrap-to-bootstrap heterogeneity (item-sample difficulty) and $v_b$ allows the randomization penalty to vary by sampled item pool. This regression yields (i) an estimate $\alpha_s$ of the expected fit of structure $s$ under correct item assignment, (ii) an estimate $\beta_s$ of how much of that fit is explainable by structure rather than flexibility, and (iii) uncertainty that correctly propagates item-sampling variability.

## E.4. Inverse-error meta-analytic aggregation of bootstrapped estimates

This subsection describes the statistical machinery that converts the raw output of $B$ bootstrap replications—each producing point estimates with associated standard errors from the MH-RM algorithm—into the final aggregated quantities reported

*Table 7.* Estimated Overall Predictive Effects by latent Structure based on meta-analytic regression

| | (AUC) | | (AUC) | | (MAE) | | (MAE) | |
|---|---|---|---|---|---|---|---|---|
| | Est. | S.E. | Est. | S.E. | Est. | S.E. | Est. | S.E. |
| bifact Fig. 2.e, §C.2.5 | **0.905***** | 0.010 | **0.954***** | 0.011 | **0.244***** | 0.007 | **0.208***** | 0.007 |
| corrbifact Fig. 2.f, §C.2.6 | 0.718*** | 0.008 | 0.771*** | 0.009 | 0.357*** | 0.005 | 0.321*** | 0.006 |
| corrfact Fig. 2.d, §C.2.4 | 0.677*** | 0.008 | 0.731*** | 0.009 | 0.385*** | 0.005 | 0.349*** | 0.006 |
| gfact Fig. 2.b, §C.2.2 | 0.868*** | 0.006 | 0.915*** | 0.008 | 0.279*** | 0.004 | 0.248*** | 0.005 |
| hier2ordFig. 2.c, §C.2.3 | 0.885*** | 0.010 | 0.938*** | 0.011 | 0.259*** | 0.006 | 0.223*** | 0.007 |
| indepfact Fig. 2.a, §C.2.1 | 0.900*** | 0.008 | 0.953*** | 0.010 | 0.250*** | 0.005 | 0.215*** | 0.006 |
| SD (Intercept bootstrap) | 0.039 | | 0.000 | | 0.027 | | 0.000 | |
| SD (Observations) | 0.103 | | 0.102 | | 0.065 | | 0.066 | |
| SD (num. items in estimation) | | | 0.006 | | | | 0.004 | |
| Num.Obs. | 1082 | | 1082 | | 1082 | | 1082 | |
| R2 Marg. | 0.406 | | 0.440 | | 0.368 | | 0.402 | |
| R2 Cond. | 0.482 | | | | 0.462 | | | |

$+ \; p < 0.1, \; * \; p < 0.05, \; ** \; p < 0.01, \; *** \; p < 0.001$

throughout this study. The procedure applies uniformly to fixed-effect coefficients (scaling slopes $\hat{\beta}_d^{(b)}$), latent ability scores (posterior BLUPs $\hat{\theta}_{id}^{(b)}$), and variance components ($\hat{\sigma}_S^{2\,(b)}$), though we develop it here primarily for the fixed effects and latent scores of Method 3.

### E.4.1. GENERAL FRAMEWORK FOR INVERSE ERROR AGGREGATION

Let $\xi$ denote a generic scalar estimand (a single element of $\mathbf{\Gamma}$, a latent score $\theta_{id}$, or a variance component). For each bootstrap $b$, the MH-RM algorithm returns $\hat{\xi}^{(b)}$ with $\mathrm{SE}^{(b)} \equiv \mathrm{SE}(\hat{\xi}^{(b)})$. The posterior means of the latent trait BLUPs and the asymptotic standard errors from the observed information matrix are direct outputs of MH-RM's stochastic EM convergence diagnostics (Cai, 2010b; Chalmers, 2015).

**Inverse-variance weights.** Define bootstrap-specific weights

$$w^{(b)} \; = \; \frac{1}{\left(\mathrm{SE}^{(b)}\right)^2}, \tag{15}$$

so that $w^{(b)}$ is large when the estimate from replication $b$ is precise and small when it is noisy. In the sequel, we write $W = \sum_{b=1}^{B} w^{(b)}$ for the total weight.

**Aggregated point estimate.** The inverse-variance weighted (IVW) mean is

$$\tilde{\xi} \; = \; \frac{\sum_{b=1}^{B} w^{(b)} \hat{\xi}^{(b)}}{\sum_{b=1}^{B} w^{(b)}} \; = \; \frac{1}{W} \sum_{b=1}^{B} w^{(b)} \hat{\xi}^{(b)}. \tag{16}$$

**Standard error of the aggregate.** Under the assumption that bootstrap replications are conditionally independent given the data (which holds by construction, as each replication draws items independently), the variance of $\tilde{\xi}$ is

$$\mathrm{Var}(\tilde{\xi}) \; = \; \frac{1}{W} \; = \; \frac{1}{\sum_{b=1}^{B} w^{(b)}}, \qquad \mathrm{SE}(\tilde{\xi}) \; = \; \frac{1}{\sqrt{W}}. \tag{17}$$

**Proposition E.1** (Optimality of inverse-variance weighting). *Among all linear combinations $\hat{\xi}_{\mathbf{a}} = \sum_b a_b \hat{\xi}^{(b)}$ with $\sum_b a_b = 1$, the inverse-variance weighted mean $\tilde{\xi}$ uniquely minimizes $\mathrm{Var}(\hat{\xi}_{\mathbf{a}})$ when $\hat{\xi}^{(1)}, \dots, \hat{\xi}^{(B)}$ are uncorrelated with known variances $(\mathrm{SE}^{(b)})^2$.*

*Proof.* By the method of Lagrange multipliers, minimize $\sum_b a_b^2 (\mathrm{SE}^{(b)})^2$ subject to $\sum_b a_b = 1$. The stationary conditions yield $a_b = w^{(b)}/W$, recovering (16). The minimized variance is $1/W$. □

### E.4.2. HANDLING EDGE CASES

Two practical complications arise in our setting and must be addressed to ensure robustness.

**Missing or degenerate standard errors.**  Occasionally, the MH-RM algorithm may fail to converge for a particular parameter in a given bootstrap, or may return $\mathrm{SE}^{(b)} = 0$ (exact boundary solution) or $\mathrm{SE}^{(b)} = \infty$ (non-identified parameter under that item draw). We handle these as follows:

$$w^{(b)} = \begin{cases} 1/\left(\mathrm{SE}^{(b)}\right)^2 & \text{if } 0 < \mathrm{SE}^{(b)} < \infty, \\ \dfrac{1}{2} \min_{b' : \, \mathrm{SE}^{(b')} \text{ valid}} w^{(b')} & \text{if } \mathrm{SE}^{(b)} \text{ is missing or degenerate.} \end{cases} \tag{18}$$

That is, degenerate replications receive half the weight of the least precise valid replication. This ensures they contribute minimally without being discarded entirely, preserving the unbiasedness of the mean while bounding their influence.

**Infinite or zero weights.**  If $\mathrm{SE}^{(b)} = 0$ exactly (perfect information), we cap the weight at a large but finite value; if $\mathrm{SE}^{(b)}$ is undefined (e.g., non-convergence), the replication is assigned the fallback weight above. In our experiments, fewer than 0.5% of replication–parameter pairs required fallback treatment.

### E.4.3. APPLICATION TO FIXED-EFFECT COEFFICIENTS

For the scaling slope on latent dimension $d \in \{0, 1, \ldots, K\}$, the MH-RM algorithm returns in each bootstrap $b$ a coefficient $\hat{\beta}_d^{(b)} \equiv (\hat{\mathbf{B}}^{(b)})_{d,\,x}$ and its standard error $\mathrm{SE}_d^{(b)}$ from the observed information matrix. The aggregated scaling slope and its uncertainty are

$$\tilde{\beta}_d = \frac{\sum_b w_d^{(b)} \hat{\beta}_d^{(b)}}{\sum_b w_d^{(b)}}, \qquad w_d^{(b)} = \frac{1}{\left(\mathrm{SE}_d^{(b)}\right)^2}, \qquad \mathrm{SE}(\tilde{\beta}_d) = \frac{1}{\sqrt{\sum_b w_d^{(b)}}}. \tag{19}$$

This procedure is applied independently for each latent dimension, yielding the aggregated scaling vector $\tilde{\boldsymbol{\beta}} = (\tilde{\beta}_0, \tilde{\beta}_1, \ldots, \tilde{\beta}_K)$ reported in Table 3 and Figures 3–4.

*Remark* E.2 (Connection to fixed-effects meta-analysis). Equation (19) is the standard fixed-effect meta-analytic estimator. In our context, the "studies" are bootstrap replications rather than independent experiments, but the statistical logic is identical: each replication provides an estimate of the *same* population quantity $\beta_d$ (the scaling slope under the item distribution induced by the benchmark ecosystem), and precision varies across replications due to the random item draw. The fixed-effect assumption—that all replications target the same $\beta_d$—is justified because item subsets are drawn from the same benchmark population and models are held fixed across replications.

### E.4.4. APPLICATION TO LATENT ABILITY SCORES

Each MH-RM fit produces posterior means (BLUPs) $\hat{\theta}_{id}^{(b)}$ and associated posterior standard deviations $\mathrm{SE}^{(b)}(\hat{\theta}_{id})$ for every model $i$ and latent dimension $d$. The aggregated latent score for model $i$ on dimension $d$ is

$$\tilde{\theta}_{id} = \frac{\sum_b w_{id}^{(b)} \hat{\theta}_{id}^{(b)}}{\sum_b w_{id}^{(b)}}, \qquad w_{id}^{(b)} = \frac{1}{\left(\mathrm{SE}^{(b)}(\hat{\theta}_{id})\right)^2}. \tag{20}$$

Note that the weights here are *model-and-dimension-specific*: a model that responds to many high-discrimination items in bootstrap $b$ will have a more precisely estimated $\hat{\theta}_{id}^{(b)}$ (smaller posterior SD) and thus receive higher weight for that replication.

### E.4.5. PERCENTILE RANKS FROM AGGREGATED SCORES.

The leaderboard comparisons in §4.5 use percentile ranks derived from the aggregated general-factor scores $\{\tilde{\theta}_{i0}\}_{i=1}^N$ and from the aggregated latent-regression-adjusted scores. Because both are IVW aggregates, they reflect the full bootstrap distribution of item sampling uncertainty while upweighting the most informative item draws. Percent ranking counts the total number of values less than $\theta_{gi}$, and divides it by the number of observations minus 1: $\frac{1}{n-1} \sum_x \mathbb{1}[x < \theta_{gi}]$.

E.4.6. DECOMPOSING BOOTSTRAP VARIABILITY: WITHIN VERSUS BETWEEN

The total variability of $\hat{\xi}^{(b)}$ across bootstraps reflects two distinct sources: **Within-bootstrap estimation uncertainty** $(\text{SE}^{(b)})^2$: the posterior or asymptotic variance of $\hat{\xi}^{(b)}$ conditional on item set $\tilde{\mathcal{J}}^{(b)}$. This is what the MH-RM information matrix measures. **Between-bootstrap item-sampling variability** $\tau_\xi^2$: the variance of the true population estimand across item draws, reflecting the sensitivity of the conclusion to the particular items selected. The observed variance of the estimates across bootstraps conflates both:

$$\text{Var}_b\big(\hat{\xi}^{(b)}\big) \;=\; \tau_\xi^2 \;+\; \mathbb{E}_b\big[(\text{SE}^{(b)})^2\big]. \tag{21}$$

The first term measures genuine item-sampling heterogeneity (how much the estimand depends on which items are drawn); the second measures average estimation noise. Inverse-variance weighting eliminates the second source from the aggregated point estimate, but both contribute to the uncertainty around any single bootstrap's result.

**Proposition E.3** (IVW aggregation removes within-bootstrap noise from the point estimate). *Under the decomposition in (21), the variance of the IVW aggregate $\tilde{\xi}$ satisfies $\text{Var}(\tilde{\xi}) \;\leq\; \frac{1}{B}\text{Var}_b\big(\hat{\xi}^{(b)}\big)$, with equality only when all weights are identical. When weights vary, the IVW estimate achieves strictly lower variance than the unweighted mean, with the improvement proportional to the heterogeneity of $\{(\text{SE}^{(b)})^2\}_b$.*

*Proof.* By Proposition E.1, $\text{Var}(\tilde{\xi}) \;=\; 1/W$. Under identical weights $w^{(b)} = \bar{w}$, we have $W = B\bar{w}$ and $\text{Var}(\tilde{\xi}) = 1/(B\bar{w}) = \text{Var}_b(\hat{\xi}^{(b)})/B$. For heterogeneous weights, the Cauchy-Schwarz inequality applied to $\sum_b a_b^2(\text{SE}^{(b)})^2$ with optimal $a_b = w^{(b)}/W$ yields strict improvement whenever the $(\text{SE}^{(b)})^2$ are not all equal. $\square$

E.4.7. SLOPE RELIABILITY FROM THE IVW FRAMEWORK

A natural by-product of the IVW aggregation is the *slope reliability index* for each latent dimension $d$. From the bootstrapped estimates $\{\hat{\beta}_d^{(b)}\}_b$, we compute the weighted between-bootstrap variance

$$\hat{\tau}_d^2 \;=\; \frac{\sum_b w_d^{(b)}\big(\hat{\beta}_d^{(b)} - \tilde{\beta}_d\big)^2}{\sum_b w_d^{(b)}} \;-\; \frac{1}{\bar{w}_d}, \tag{22}$$

where $\bar{w}_d = W_d/B$ is the average weight, and the subtracted term is a bias correction removing the expected contribution of within-bootstrap noise (analogous to the DerSimonian-Laird estimator in meta-analysis). The slope reliability index is then

$$\mathcal{R}_d \;=\; \frac{\tilde{\beta}_d^2}{\tilde{\beta}_d^2 + \hat{\tau}_d^2}. \tag{23}$$

High $\mathcal{R}_d$ (close to 1) indicates that the scaling slope for latent dimension $d$ is stable across item subsets and contributor clusters: the estimated law is a property of the ecosystem, not of a particular item draw. Low $\mathcal{R}_d$ signals that the slope is sensitive to item composition, and any single estimate should be interpreted cautiously. Table 3 reports $\mathcal{R}_d$ for each latent dimension.

# F. Method 1 Fit Statistics

**Localized Diagnostics with Modification Indices**   Global fit indices (e.g., CFI, RMSEA) assess overall model-data congruence but do not identify specific sources of misfit or redundancy. For this, we use modification indices (MIs). An MI estimates the expected decrease in the model's $\chi^2$ statistic if a single fixed parameter (e.g., a cross-loading or a residual covariance fixed at zero) were freely estimated (Saris et al., 2009). This provides a powerful, localized diagnostic for detecting systematic measurement noise that could inflates apparent reliability without increasing construct coverage. More importantly, in the bootstrap context, these serve as generalized indicators of where benchmarks are over and underfitting their respective constructs: this is the extent to which items are measuring each other rather than the latent constructs intended by their respective benchmark.

**Proposition F.1** (Modification Indices as Score Tests). *Let $L(\boldsymbol{\theta})$ be the log-likelihood function for the model parameters $\boldsymbol{\theta}$. Consider a model with a constraint $c(\boldsymbol{\theta}) = 0$. The modification index for relaxing this constraint is the score statistic (or Lagrange Multiplier test) for the null hypothesis $H_0 : c(\boldsymbol{\theta}) = 0$. It is calculated as $s(\hat{\boldsymbol{\theta}}_r)^T I(\hat{\boldsymbol{\theta}}_r)^{-1} s(\hat{\boldsymbol{\theta}}_r)$, where $\hat{\boldsymbol{\theta}}_r$ is the maximum likelihood estimate under the constraint, $s(\cdot)$ is the score vector $(\partial L/\partial c)$, and $I(\cdot)$ is the Fisher information matrix for the parameter c.*

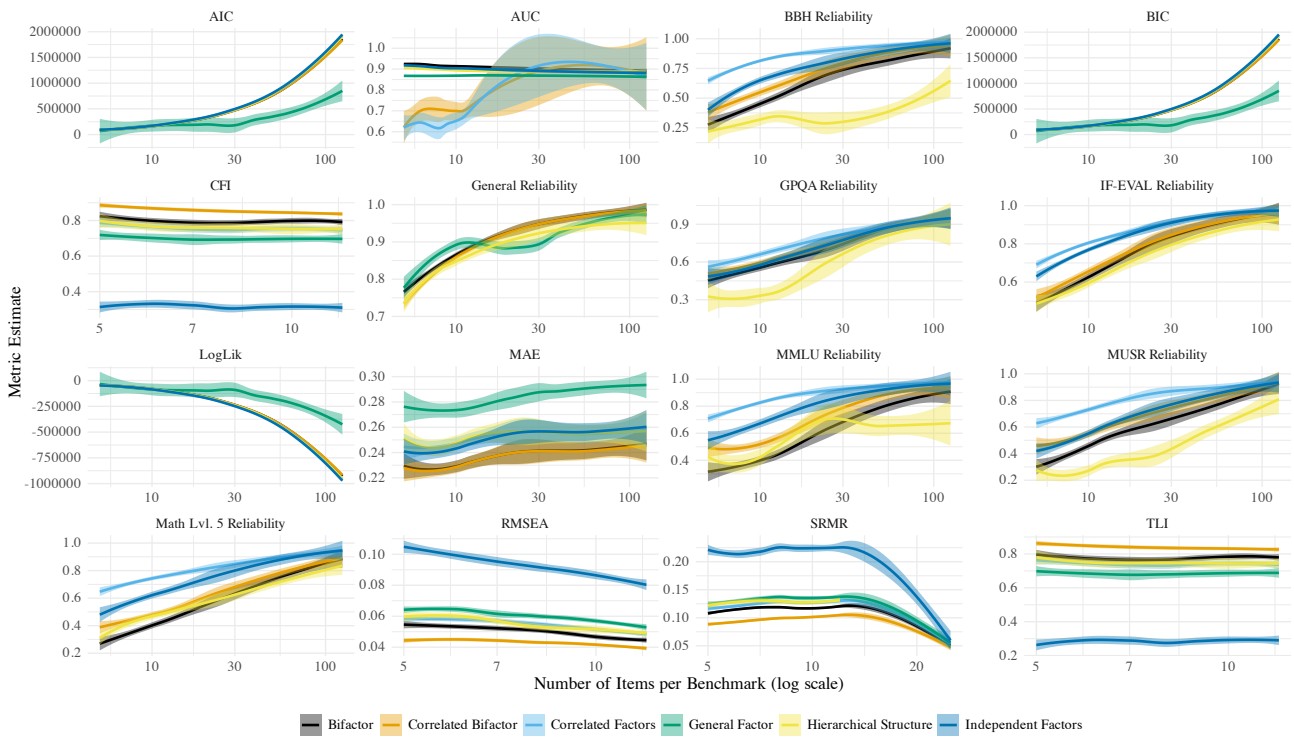

*Figure 5.* Sensitivity of Fit Estimates to Bootstrap sample size, meta-aggregated across both estimation methods.

*Table 8.* Likelihood Ratio Tests of Statistical Significance across Bootstraps

| Simpler Structure $\rightarrow$ | Proportion Significant LRT ($\alpha = 0.05$) | | | | |
| --- | --- | --- | --- | --- | --- |
| Complex Structure $\downarrow$ | Independent | General | Hierarchical | Correlated | Bifactor |
| General Factor | 0.58[†] | – | – | – | – |
| Independent Factors | – | 0.42[†] | – | – | – |
| Hierarchical Structure | 1.00 | 1.00[†] | – | – | – |
| Correlated Factors | 1.00 | 1.00[†] | 0.91[†] | – | – |
| Bifactor | 1.00 | 1.00 | 1.00[†] | 1.00[†] | – |
| Correlated Bifactor | 1.00 | 1.00 | 1.00[†] | 1.00 | 1.00 |

Values represent the proportion of total bootstraps where likelihood ratio tests indicated that the more complex model was statistically significantly a better fit. † indicates pseudo-LRT for non-nested models using evidence ratio and combined information criteria indices.

*Table 9.* Traditional Fit Statistics under Meta-analytic bootstrap aggregation from DWLS Estimator

| term | RMSEA fit / Est. | RMSEA fit / S.E. | CFI fit / Est. | CFI fit / S.E. | TLI fit / Est. | TLI fit / S.E. | SRMR fit / Est. | SRMR fit / S.E. |
|---|---|---|---|---|---|---|---|---|
| structbifact | 0.049*** | 0.001 | 0.843*** | 0.008 | 0.829*** | 0.008 | 0.113*** | 0.002 |
| structcorrbifact | 0.043*** | 0.001 | 0.886*** | 0.007 | 0.874*** | 0.008 | 0.096*** | 0.002 |
| structcorrfact | 0.054*** | 0.001 | 0.804*** | 0.007 | 0.793*** | 0.008 | 0.123*** | 0.002 |
| structgfact | 0.059*** | 0.001 | 0.760*** | 0.007 | 0.750*** | 0.008 | 0.131*** | 0.002 |
| structhier2ord | 0.054*** | 0.001 | 0.803*** | 0.010 | 0.794*** | 0.011 | 0.123*** | 0.002 |
| structindepfact | 0.090*** | 0.001 | 0.344*** | 0.007 | 0.315*** | 0.008 | 0.215*** | 0.002 |
| structbifact × randTRUE | 0.009*** | 0.001 | -0.046*** | 0.005 | -0.051*** | 0.005 | 0.012*** | 0.001 |
| structcorrbifact × randTRUE | 0.006*** | 0.001 | -0.017*** | 0.004 | -0.019*** | 0.005 | 0.008*** | 0.001 |
| structcorrfact × randTRUE | 0.005*** | 0.001 | -0.025*** | 0.004 | -0.027*** | 0.005 | 0.008*** | 0.001 |
| structhier2ord × randTRUE | 0.005* | 0.003 | -0.025* | 0.011 | -0.026* | 0.012 | 0.007** | 0.003 |
| structindepfact × randTRUE | 0.015*** | 0.001 | -0.231*** | 0.004 | -0.242*** | 0.005 | 0.022*** | 0.001 |
| SD (intercept item batch) | 0.007 | | 0.094 | | 0.098 | | 0.019 | |
| SD (rand item batch) | 0.001 | | 0.011 | | 0.010 | | 0.002 | |
| Cor (intercept ∼ rand item batch) | -1.000 | | 1.000 | | 1.000 | | 1.000 | |
| SD (Observations) | 0.006 | | 0.046 | | 0.049 | | 0.011 | |
| Num.Obs. | 2062 | | 2341 | | 2341 | | 2341 | |
| R2 Marg. | 0.905 | | 0.966 | | 0.965 | | 0.944 | |
| AIC | -14225.2 | | -6610.5 | | -6333.2 | | -13329.5 | |
| BIC | -14140.8 | | -6524.1 | | -6246.8 | | -13243.1 | |
| RMSE | 0.01 | | 0.04 | | 0.05 | | 0.01 | |

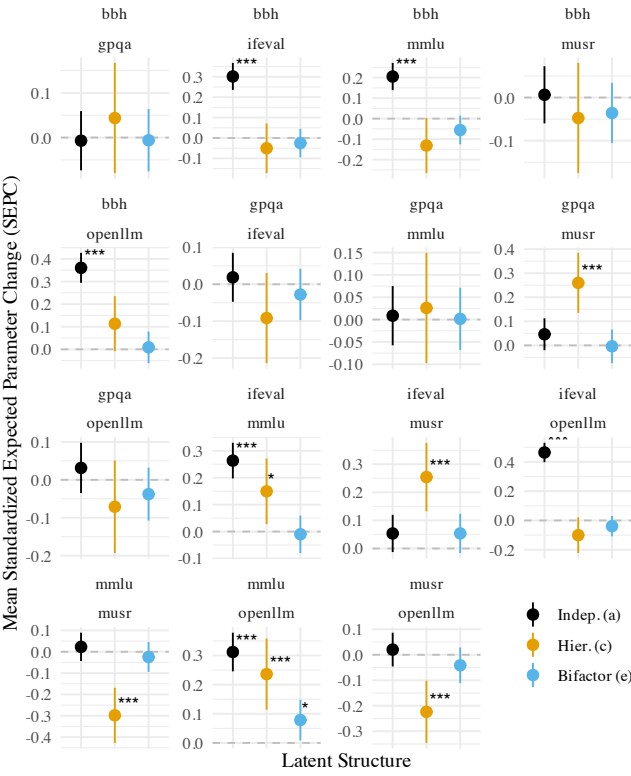

Benchmark Relationship Misfit by Latent Structure

*Figure 6.* Estimated mean misfit of relationships between benchmark constructs by latent structure using Langrangian Multiplier test standardized expected parameter changes over bootstraps. Higher SEPC values indicate the model underestimates the inter-benchmark relationships, and lower values indicate overestimation. Labels above each plot indicate which two benchmarks and stars indicate levels of significance: '***'< 0.001; '**'< 0.01; '*'< 0.05.

## F.1. Permutation Controls

Additionally, the traditional fit results from the permutation controls from the DWLS estimation are found in Table 9.

Benchmark Relationship Misfit by Latent Structure

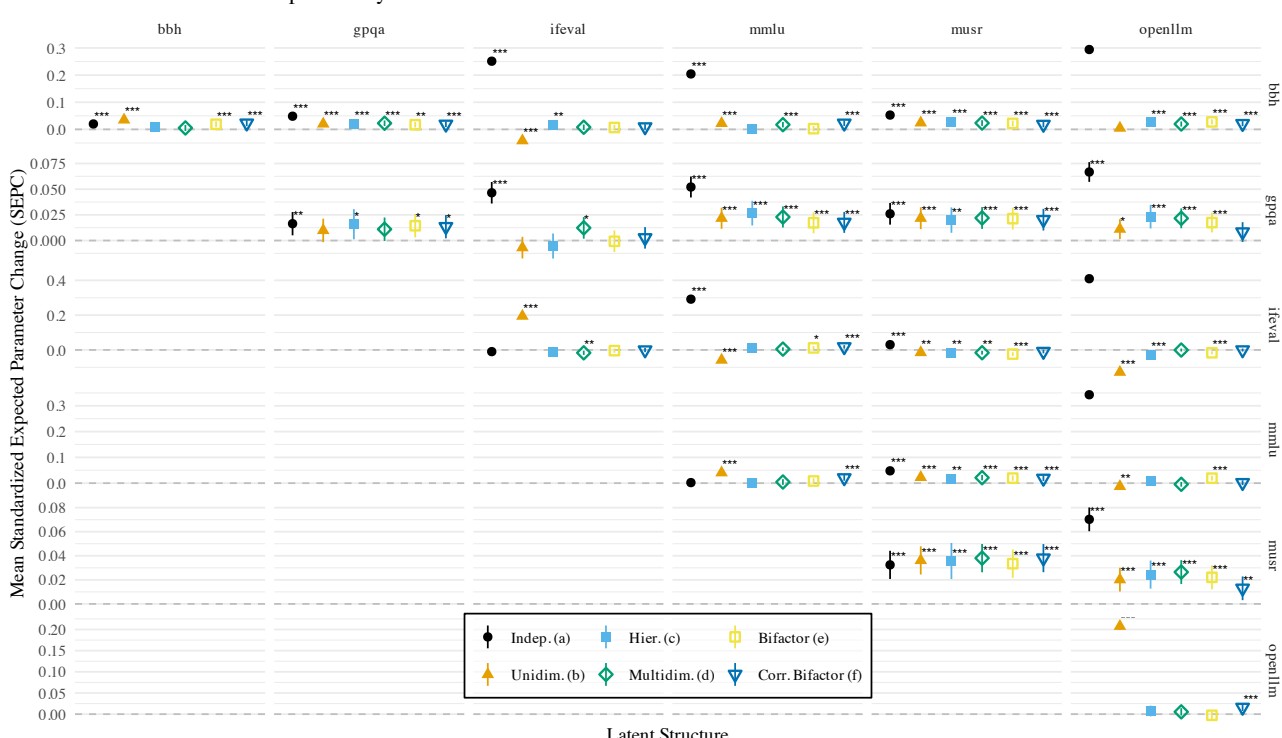

*Figure 7.* Estimated mean misfit of relationships between benchmark constructs by latent structure using Langrangian Multiplier test standardized expected parameter changes after freeing interitem residual variances over bootstraps. Higher SEPC values indicate the hypothesized latent structure underestimates the inter-benchmark relationships, and lower values indicate overestimation. Labels above each plot indicate which two benchmarks and stars indicate levels of significance: '***'< 0.001; '**'< 0.01; '*'< 0.05.

# G. Variance Decomposition

## G.1. Relationships in Variance Decomposition

### G.1.1. FROM VARIANCE TO RELIABILITY

Variance components are the building blocks for estimating reliability. Generalizability Theory operationalizes variance components and has found increasing use in LLM evaluations (Brennan, 2003; Hardy & Kim, 2026; Hardy, 2025b). The Generalizability Coefficient (G-coefficient) provides a strong measure of reliability for a given measurement context, defined as the ratio of "true" variance to expected observed variance:

$$G = \frac{\sigma_{\text{true}}^2}{\sigma_{\text{true}}^2 + \sigma_{\text{error}}^2} \tag{24}$$

The definitions of "true" and "error" variance depend on the decision being made. For example, if we wish to assess the reliability of ranking models by their average score across our $n_B = 6$ benchmarks, the "true" variance is the variance associated with the model's stable characteristics (e.g., $\sigma_A^2, \sigma_C^2, \sigma_D^2$ and their non-benchmark interactions). The "error" variance includes all components that interact with the Benchmark facet, as these cause a model's rank to change from one benchmark to another. For a relative decision (ranking), the G-coefficient for a model's average score across $n_B$ benchmarks is:

$$G_{\text{relative}} = \frac{\sigma_{\text{Universe Score}}^2}{\sigma_{\text{Universe Score}}^2 + \sigma_{\text{Relative Error}}^2} \tag{25}$$

where $\sigma_{\text{Universe Score}}^2$ is the sum of variance components for the object of measurement (e.g., model facets A, C, D) and $\sigma_{\text{Relative Error}}^2$ is the sum of all interaction variances involving the benchmark facet (e.g., $\sigma_{AB}^2/n_B, \sigma_{BC}^2/n_B, \dots$) plus the residual error. This framework allows us to precisely estimate the trustworthiness of leaderboard rankings.

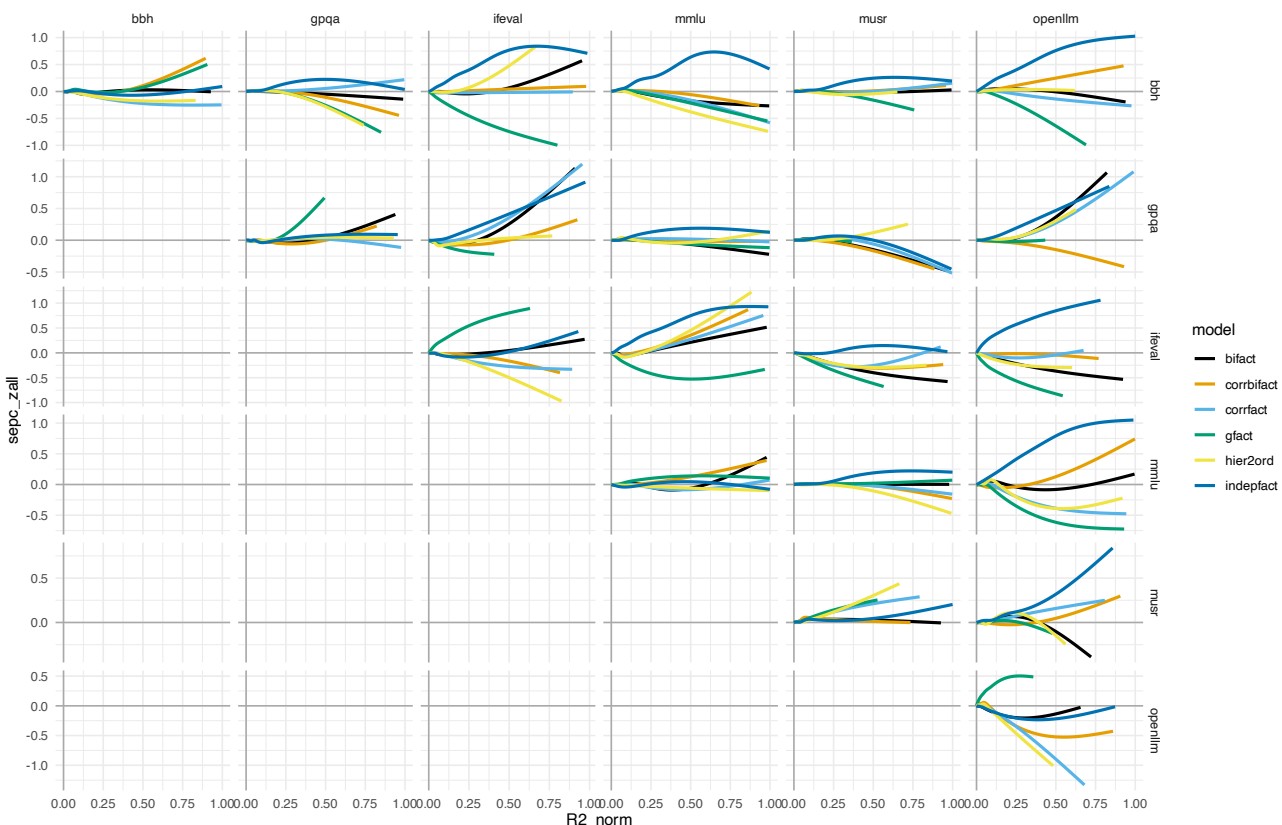

*Figure 8.* Average Effect Size Changes to models based on single-degree Lagrangian Multiplier tests of residual variance. X axis represents the amount of variance explained implied by the score tests; more positive values indicate that the test is capturing more variation. Y axis represents the actual effect sizes; more positive and more negative values mean that the latent structure is underestimating and overestimating, respectively, the relationships. The columns and rows indicate from which benchmarks the residual variance is being freed.

- **Architecture facet (A):** groups by base architecture and model generation (e.g., `architecture:generation`). This captures systematic differences between underlying model families and releases.

- **Benchmark facet (B):** groups by benchmark identity bench to capture persistent differences in difficulty and scaling across tasks.

- **Contributor facet (C):** groups by the Hugging Face account and submission-related availability metadata. Operationally, we use `author:removed:not_avail`, where `removed` indicates whether the base model in the lineage is removed and `not_avail` indicates whether the evaluated model remains available on the hub. This facet measures stable differences attributable to the submission source and associated behaviors (without asserting causal interpretation).

- **Deployment facet (D):** groups by tuning approaches/ deployment type `type:chat_template:mo_e: merged:precision`. This captures differences between base vs. chat models, use of chat templates, mixture-of-experts, model merging, and precision/quantization.

### G.2. Reliability of scaling effects and rank ordering under a four-facet crossed mixed model

To study the *reliability of scaling laws* in benchmark ecosystems, it is not enough to report a fixed-effect slope $\hat{\beta}$ for model size. We must also quantify (i) how much of cross-model rank ordering is *stably* attributable to size, (ii) how strongly the size–score relationship varies across the ecosystem facets (benchmarks, contributors, deployments, architectures), and (iii) how precisely we can generalize the estimated slope to new benchmarks and ecosystem conditions.

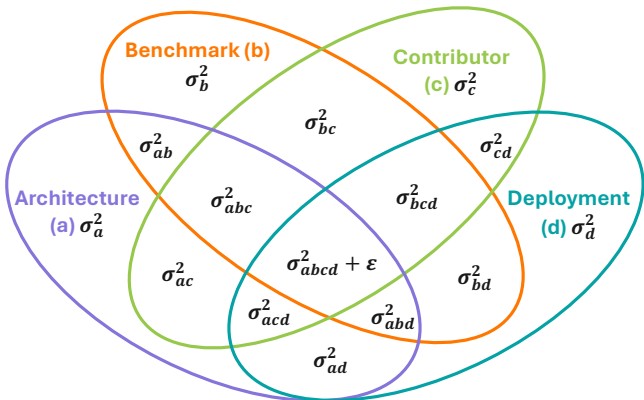

*Figure 9.* Variance Decomposition diagram for four fully crossed facets of variation

**Model.** Let $y_i$ denote a per-benchmark score observation with associated log-size covariate $x_i = \log_{10}(\#\text{params}_i)$ and facet indices

$$a(i) \in \mathcal{A}, \quad b(i) \in \mathcal{B}, \quad c(i) \in \mathcal{C}, \quad d(i) \in \mathcal{D}$$

for Architecture (A), Benchmark (B), Contributor (C), and Deployment (D). We fit a fully crossed random-intercept/random-slope LMM:

$$y_i = \mu + \beta x_i + \sum_{S \in \mathcal{S}} \left( u_{g_S(i)}^S + v_{g_S(i)}^S x_i \right) + \varepsilon_i, \tag{26}$$

where $\mathcal{S}$ is the set of random-effect terms (main effects and interactions) included by the measurement design (e.g., $A, B, C, D, AB, AC, \ldots, ABCD$), and $g_S(i)$ maps observation $i$ to the level of term $S$ (e.g., $g_{AB}(i) = (a(i), b(i))$). We assume $u_g^S \sim \mathcal{N}(0, \sigma_{S,0}^2)$, $v_g^S \sim \mathcal{N}(0, \sigma_{S,1}^2)$, $\varepsilon_i \sim \mathcal{N}(0, \sigma_\varepsilon^2)$, and, to reduce overfitting in a high-dimensional crossed design, we use the independence parameterization $\text{Cov}(u_g^S, v_g^S) = 0 \; \forall S, g$. unconditional variance in $y$ is only valid after clearly defining whether the variance is (i) unconditional over $x$ and facets, (ii) conditional on the realized $x$'s in-sample, and (iii) whether benchmark main effects are included (which otherwise dominate the total variance budget). Below we define inferentially precise quantities aligned with "reliability of scaling laws" and "reliability of size-induced ranking".

### G.3. Variance Decomposition Robustness Checks for Sensitivity to Categorizations

**Challenge: Sensitivity to categorization.** Variance decompositions can be sensitive to groupings used. **Solution 1:** *Multi-granularity robustness checks.* We re-estimate main-effects variance shares at three levels of categorical resolution (Table 10) to verify that the observed ordering of $B \gg C > A > D$ is not an artifact of a particular coding scheme. **Solution 2:** *Bayesian estimation.* We complement Restricted ML with a Bayesian MCMC framework (4 chains, weak priors) to obtain full posterior distributions over variance shares, providing 95% HDI intervals for each component (Table 13).

G.3.1. GRANULARITY TESTS

We conducted robustness analyses across multiple levels of categorical granularity using Bayesian estimation with 95% credible intervals. Table 10 reports the percentage of variance attributable to each facet under main-effects-only models.

*Table 10.* **Categorization Robustness checks:** Variance-share estimates (%) across alternative granularity specifications. Numbers in parentheses indicate the number of levels within each facet.

| Facet | Finest | Fine (Table 2) | Coarsest | Coarsest Groups |
|---|---|---|---|---|
| (A) Architecture | 11.0 (1120) | 10.1 (119) | 9.2 (54) | LLM "family" |
| (B) Benchmark | 46.4 (6) | 47.2 (6) | 46.8 (6) | Benchmarks |
| (C) Contribution | 15.8 (1163) | 13.8 (878) | 13.8 (714) | HF Account |
| (D) Deployment | 2.3 (74) | 3.8 (61) | 3.5 (7) | HF model "type" |

*Table 11.* Variance Component Estimates, Covariate Model (3)

| Variation Source | Var. ($\sigma^2$) | $p_S$ (%) | PSI$_S$ |
|---|---|---|---|
| A (Architecture) | 18.73 | 6.34 | 0.13 |
| B (Benchmark) | 136.30 | 46.13 | 0.30 |
| C (Contributor) | 23.09 | 7.82 | 0.17 |
| D (Deployment) | 3.26 | 1.10 | 0.03 |
| C×D | 3.26 | 1.11 | 0.01 |
| A×C | 12.20 | 4.14 | 0.06 |
| B×A | 10.97 | 3.71 | 0.05 |
| A×D | 0.91 | 0.31 | 0.00 |
| B×D | 13.55 | 4.58 | 0.02 |
| B×C | 7.87 | 2.66 | 0.03 |
| B×C×D | 5.43 | 1.84 | 0.05 |
| B×A×C | 11.12 | 3.77 | 0.07 |
| B×A×D | 1.72 | 0.58 | 0.01 |
| C×A×D | 13.79 | 4.66 | 0.07 |
| $\epsilon_{\text{resid}}$ (+ A×B×C×D) | 33.24 | 11.25 | |

Variance component estimates $\sigma^2$ and $p_S$ represent the total variance associated with both slope and intercept. The base model (2) can be found in Table 12 and Bayesian posterior draws of $p_S$ can be found in Table 13.

The primary finding—that contributor variance exceeds architecture variance—is robust across all granularity specifications considered. More broadly, the ordering of variance components remains stable across analyses, with benchmark effects dominating contributor effects, followed by architecture and deployment effects: $B \gg C > A > D$. This stability suggests that the substantive conclusions of the analysis are not artifacts of any particular categorization scheme.

## G.4. Reliability of the fixed scaling slope: generalizability of $\beta$

The fixed effect $\beta$ summarizes the average size–score relationship across the ecosystem. Its reliability is naturally expressed by an inferential precision measure for $\hat{\beta}$ under the fitted mixed model.

### G.4.1. Restricted Maximum Likelihood and Bayesian Estimations

For the base model, we report its Restricted Maximum Likelihood estimation in Table 12 and Bayesian estimation in Table 13

**A reliability coefficient for slope $\beta$.** For comparability across datasets, we report a dimensionless reliability-like coefficient for the scaling slope:

$$R_\beta \;\equiv\; \frac{\hat{\beta}^2}{\hat{\beta}^2 + \widehat{\text{Var}}(\hat{\beta})} \in [0, 1].$$

This is the fraction of (squared) signal retained after accounting for estimation noise, analogous to a shrinkage factor. High $R_\beta$ indicates that the estimated scaling law is stable under the current measurement design; low $R_\beta$ indicates that the fixed scaling effect is not reliably estimable given crossed ecosystem variability.

## G.5. Reliability of rank ordering attributable to size

Scaling laws are often used to justify *rank ordering* models by predicted performance from size alone. We therefore define reliability at the level of the *size-induced predictor* and its stability relative to ecosystem noise.

Let the size-only linear predictor be

$$\hat{y}_i^{(\text{size})} \;=\; \hat{\mu} + \hat{\beta}x_i.$$

*Table 12.* Variance Component Estimates (No Covariate Model).

| Source of Variation | Var. ($\sigma^2$) | % of Total |
|---|---|---|
| *Main Effects* | | |
| (B) Benchmark | 150.0 | 43.0% |
| (C) Contributor | 29.91 | 8.6% |
| (A) Architecture | 14.5 | 4.17% |
| (D) Deployment Type | 6.64 | 1.91% |
| *Two-Way Interactions* | | |
| B × C | 12.7 | 3.66% |
| A × C | 12.3 | 3.5% |
| B × A | 10.21 | 2.94% |
| B × D | 17.2 | 4.9% |
| C × D | 2.64 | 0.76% |
| A × D | 0.77 | 0.22% |
| *Higher-Order Interactions* | | |
| Three-Way Interactions | 38.9 | 11.2% |
| $\epsilon_{\text{resid}}$ (+ A×B×C×D) | 52.0 | 15.0% |

*Table 13.* Bayesian Estimation posterior draws for percentage of variance ($p_S$) from base variance decomposition model (2)

| Facet | mean | median | SD | MAD | q5 | q95 | $\hat{R}$ | MAP | HDI$_{low}$ | HDI$_{high}$ |
|---|---|---|---|---|---|---|---|---|---|---|
| A | 0.05 | 0.04 | 0.02 | 0.02 | 0.02 | 0.09 | 1.01 | 0.04 | 0.02 | 0.10 |
| A×C | 0.03 | 0.03 | 0.01 | 0.01 | 0.01 | 0.05 | 1.02 | 0.02 | 0.01 | 0.06 |
| A×D | 0.00 | 0.00 | 0.00 | 0.00 | 0.00 | 0.01 | 1.02 | 0.00 | 0.00 | 0.01 |
| C | 0.08 | 0.08 | 0.03 | 0.03 | 0.04 | 0.13 | 1.01 | 0.08 | 0.03 | 0.13 |
| C×A×D | 0.08 | 0.08 | 0.02 | 0.03 | 0.04 | 0.12 | 1.01 | 0.09 | 0.03 | 0.12 |
| C×D | 0.01 | 0.00 | 0.01 | 0.00 | 0.00 | 0.02 | 1.15 | 0.00 | 0.00 | 0.02 |
| B | 0.50 | 0.49 | 0.15 | 0.17 | 0.27 | 0.75 | 1.01 | 0.46 | 0.25 | 0.79 |
| B×A | 0.03 | 0.03 | 0.01 | 0.01 | 0.01 | 0.04 | 1.01 | 0.03 | 0.01 | 0.04 |
| B×A×C | 0.02 | 0.02 | 0.01 | 0.01 | 0.01 | 0.03 | 1.01 | 0.02 | 0.01 | 0.03 |
| B×A×D | 0.00 | 0.00 | 0.00 | 0.00 | 0.00 | 0.01 | 1.01 | 0.00 | 0.00 | 0.01 |
| B×C | 0.03 | 0.03 | 0.01 | 0.01 | 0.02 | 0.05 | 1.00 | 0.03 | 0.01 | 0.05 |
| B×C×D | 0.01 | 0.01 | 0.00 | 0.00 | 0.00 | 0.01 | 1.01 | 0.01 | 0.00 | 0.01 |
| B×D | 0.04 | 0.04 | 0.01 | 0.02 | 0.02 | 0.07 | 1.01 | 0.05 | 0.02 | 0.07 |
| D | 0.02 | 0.02 | 0.01 | 0.01 | 0.01 | 0.05 | 1.00 | 0.01 | 0.00 | 0.06 |
| sigma | 0.14 | 0.14 | 0.04 | 0.05 | 0.07 | 0.20 | 1.01 | 0.15 | 0.06 | 0.21 |

To directly estimate sensitivity of variance shares to categorizations, we re-estimate Table 12 base model using a MCMC Bayesian framework with 4 chains and weak priors. The table values represent proportions of variance explained by component, $p_S$, calculated directly from posterior draws.

A natural population-level reliability of size-driven ranking is the fraction of marginal variance in $y$ explained by $\hat{y}^{(\text{size})}$:

$$\Omega_x \equiv \frac{\text{Var}(\beta x)}{\text{Var}(y)} = \frac{\beta^2 \, \text{Var}(x)}{\beta^2 \text{Var}(x) + \text{Var}\left(\sum_S u^S_{g_S}\right) + \text{Var}(\varepsilon)}, \tag{27}$$

where $\text{Var}(y)$ is the unconditional marginal variance under (26). In our model, $\text{Var}\left(\sum_S u^S_{g_S}\right) \approx \sum_S \sigma^2_{S,0}$ (for a randomly drawn observation ), and $\text{Var}(\varepsilon) = \sigma^2_\varepsilon$. We estimate (27) by plug-in using $\hat{\beta}$ and variance-component estimates. We interpret $\Omega_x$ as: *how much of overall score variability (and thus potential rank ordering) is attributable to size alone.* For this dataset, $\Omega_x = 0.135$, indicating that global rankings cannot be reliably without considering the benchmark's influence.

**Benchmark-independent rank ordering.** Because the benchmark main effect often dominates total variance, $\Omega_x$ can be misleadingly small even when size strongly explains *cross-benchmark* rank order. We therefore also report the benchmark-demeaned reliability:

$$\Omega_x^{(-B)} \equiv \frac{\beta^2 \operatorname{Var}(x)}{\beta^2 \operatorname{Var}(x) + \sum_{S \neq B} \sigma_{S,0}^2 + \sigma_\varepsilon^2}, \tag{28}$$

i.e., the size-explainable share of variance after removing the benchmark difficulty component (and optionally other nuisance main effects depending on the decision context). This aligns with the practical question: *if we compare models across benchmarks, how much of the benchmark-independent ranking is size-driven?* In this case, $\Omega_x^{(-B)} = 0.26$.

### G.6. Decomposing variance to quantify the reliability of scaling laws

A central claim in modern AI is the existence of so-called "scaling laws", where performance improves predictably with model size. In our framework, this corresponds to the fixed effect of log-parameters, $\beta$, in a random-slopes model. However, an estimate of $\beta$ is only meaningful if the scaling relationship is reasonably stable across the ecosystem. A large average effect can mask extreme heterogeneity, where scaling benefits are strong for some benchmarks or architectures but weak or non-existent for others. Our goal is therefore not only to estimate the average scaling effect, but to precisely quantify its *reliability* and diagnose the sources of its instability.

To this end, we extend (2) to a full random-slopes model. Let $y_i$ be the score for an observation with associated log-size $x_i = \log_{10}(\# \text{ parameters})$. The score is modeled in Eq. (3):

$$s_i = (\mu + \beta x_i) + \sum_S \left( u_{g_S}^S + v_{g_S}^S x_i \right) + \varepsilon_i,$$

where the sum is over all random-effect terms $S$ (main effects and interactions), and $g_S$ indexes the level of term $S$ for observation $i$. For each term $S$, we estimate a variance component for the random intercepts, $u_{g_S}^S \sim \mathcal{N}(0, \sigma_{S,0}^2)$, and a variance component for the random slopes, $v_{g_S}^S \sim \mathcal{N}(0, \sigma_{S,1}^2)$. To ensure model stability and avoid overfitting in the fully crossed design, we constrain the random intercepts and slopes within each term to be uncorrelated.

This model allows us to decompose the total variance of scores into components related to baseline performance (intercepts) and components related to performance scaling (slopes).

**Proposition G.1** (Exact variance decomposition under a random-slopes model)**.** *Assume the random effects $u^S, v^S$ are independent across facets $S$ and independent of the covariate $x$. The total marginal variance of the outcome $y$ can be decomposed exactly as:*

$$\operatorname{Var}(y) = \underbrace{\sum_S \sigma_{S,0}^2}_{\text{Intercept Variance}} + \underbrace{\beta^2 \operatorname{Var}(x)}_{\text{Fixed Slope Variance}} + \underbrace{\left( \operatorname{Var}(x) + \operatorname{E}[x]^2 \right) \sum_S \sigma_{S,1}^2}_{\text{Random Slope Variance}} + \underbrace{\sigma_\varepsilon^2}_{\text{Residual Variance}}. \tag{29}$$

*Proof.* By the law of total variance, $\operatorname{Var}(y) = \operatorname{E}[\operatorname{Var}(y|x)] + \operatorname{Var}(\operatorname{E}[y|x])$. The conditional expectation is $\operatorname{E}[y_i|x_i] = \mu + \beta x_i$, so its variance is $\operatorname{Var}(\operatorname{E}[y|x]) = \beta^2 \operatorname{Var}(x)$. The conditional variance is $\operatorname{Var}(y_i|x_i) = \operatorname{Var}\left( \sum_S (u_{g_S}^S + v_{g_S}^S x_i) + \varepsilon_i \right) = \sum_S \sigma_{S,0}^2 + x_i^2 \sum_S \sigma_{S,1}^2 + \sigma_\varepsilon^2$. Taking the expectation over $x$ yields $\operatorname{E}[\operatorname{Var}(y|x)] = \sum_S \sigma_{S,0}^2 + \operatorname{E}[x^2] \sum_S \sigma_{S,1}^2 + \sigma_\varepsilon^2$. Since $\operatorname{E}[x^2] = \operatorname{Var}(x) + \operatorname{E}[x]^2$, the result follows. $\square$

Building on this decomposition, we introduce metrics to provide an inferentially precise assessment of the scaling law's reliability.

**1. Slope Signal-to-Noise Ratio (SNR$_\beta$).** This novel metric directly quantifies the reliability of the fixed-effect scaling law. It measures the strength of the average scaling trend ($\beta$) relative to its instability across the ecosystem's facets.

**Definition G.2** (Slope Signal-to-Noise Ratio)**.** The SNR of the fixed-effect slope $\beta$ is the ratio of the squared magnitude of the fixed effect to the total variance of the random slopes:

$$\operatorname{SNR}_\beta = \frac{\beta^2}{\sum_S \sigma_{S,1}^2}. \tag{30}$$

A high $\mathrm{SNR}_\beta$ ($\gg 1$) implies a robust, universal scaling law: the average effect $\beta$ dominates the context-specific variations. A low $\mathrm{SNR}_\beta$ ($\approx 1$ or less) indicates that the average scaling law is misleading, as the actual slope experienced by a model is highly dependent on context (i.e., which benchmark, which architecture family, etc.).

**Result**    We find that $\mathrm{SNR}_\beta = 1.12$, suggesting that an average scaling law would be misleading, and that scaling is context-dependent.

**2. Proportion of Slope Instability (PSI$_S$).**    While $\mathrm{SNR}_\beta$ quantifies the **total** instability of the scaling law, PSI$_S$ diagnoses its sources by partitioning the total random slope variance. For each facet or interaction term $S$, we define:

$$\mathrm{PSI}_S = \frac{\sigma_{S,1}^2}{\sum_{S'} \sigma_{S',1}^2}. \tag{31}$$

PSI$_S$ is the proportion of the total instability in the scaling relationship that is attributable to facet $S$. For example, a large PSI$_\mathrm{Benchmark}$ would imply that a primary reason the scaling law is unreliable is that different benchmarks respond to increases in model size in vastly different ways. This metric provides a clear, quantitative guide to where efforts to create more uniform evaluation standards should be focused.

Together, these metrics transform the variance decomposition from a descriptive exercise into a powerful diagnostic tool for assessing the structural reliability of one of the most fundamental claims in large-scale machine learning. They allow us to move beyond asking "do models scale?" to answering the more critical questions: "how reliably do they scale, and what features of the ecosystem make that scaling relationship untrustworthy?"

### G.7. Heterogeneity of scaling: where scaling laws break

A single $\beta$ is only meaningful if slope heterogeneity is small relative to the mean slope. Our crossed random slopes make this explicit.

**Global slope heterogeneity index.**    Let the effective slope for observation $i$ be $\beta_i^\mathrm{eff} = \beta + \sum_S v_{g_S(i)}^S$. We summarize heterogeneity by the variance of effective slopes:

$$\sigma_{\beta,\mathrm{eff}}^2 \equiv \mathrm{Var}(\beta_i^\mathrm{eff}) \approx \sum_S \sigma_{S,1}^2, \tag{32}$$

and report the coefficient of variation (dimensionless) $\mathrm{CV}_\beta \equiv \frac{\sqrt{\sigma_{\beta,\mathrm{eff}}^2}}{|\beta|}$. Large $\mathrm{CV}_\beta$ indicates that scaling is not a single law but a distribution of laws varying by benchmark and ecosystem facets. In the case of the present study, $\mathrm{CV}_\beta = 0.944$, meaning the standard deviation of the slope is 94.4% of the estimate, suggesting high imprecision for a global scaling law across all the data without appropriate controls.

**Facet-specific slope reliability.**    For each facet term $S$, we calculate an intra-class style share of slope variance: $H_S \equiv \frac{\sigma_{S,1}^2}{\sum_T \sigma_{T,1}^2}$, identifying which ecosystem components (benchmarks vs. contributors vs. deployments vs. architectures and their interactions) most distort scaling behavior. This answers: *where does the scaling law change?*

### G.8. A corrected "size-linked variance" decomposition

If one wishes to quantify how much of marginal variance is attributable to *all size-related terms* (fixed slope plus random slopes), define the size-linked component conditional on the distribution of $x$:

$$\mathrm{Var}_\mathrm{size}(y) \equiv \mathrm{Var}\Big(\beta x + \sum_S v_{g_S}^S x\Big).$$

The corresponding *size-linked share of marginal variance* is then

$$\Omega_\mathrm{size\text{-}all} \equiv \frac{\beta^2 \mathrm{Var}(x) + \big(\mathrm{Var}(x) + \mathrm{E}[x]^2\big) \sum_S \sigma_{S,1}^2}{\beta^2 \mathrm{Var}(x) + (\mathrm{Var}(x) + \mathrm{E}[x]^2) \sum_S \sigma_{S,1}^2 + \sum_S \sigma_{S,0}^2 + \sigma_\varepsilon^2}. \tag{33}$$

We emphasize that $\Omega_\mathrm{size\text{-}all}$ answers a different question than $\Omega_x$: it includes *heterogeneous scaling* effects (random slopes) and therefore reflects both generalizable scaling and ecosystem-specific slope distortions.

**Result**  For the present study, we calculate (33) to be $\Omega_{\text{size-all}} = 0.402$, suggesting that about 40% of all observed variation is linked to model size, as reported in the introduction.

**Reporting summary.**  To make scaling-law reliability interpretable, we report:

1. $\hat{\beta}$ with model-based and robust SE, and $R_\beta$.
2. $\Omega_x$ and $\Omega_x^{(-B)}$ (size-driven rank-order potential overall and benchmark-independent).
3. $\sigma^2_{\beta,\text{eff}}$ and $\text{CV}_\beta$ (how much scaling varies across the ecosystem).

These metrics separate three phenomena that are conflated in typical leaderboard analyses: (i) whether scale predicts performance on average, (ii) whether scale predicts *rank ordering* robustly across benchmarks, and (iii) whether "scaling laws" are stable or conditional on ecosystem facets.

*Table 14.* Variance Component Decomposition

| Facet | Effect | Main | Free Corrs. |
|-------|--------|------|-------------|
| A | $\sigma^2_{S,0}$ | 4.67 | 2.10 |
| A | log_nbpars | 14.06 | 14.08 |
| AxC | $\sigma^2_{S,0}$ | 5.60 | 23.52 |
| AxC | log_nbpars | 6.61 | 24.47 |
| AxD | $\sigma^2_{S,0}$ | 0.51 | 2.86 |
| AxD | log_nbpars | 0.40 | 3.35 |
| B | $\sigma^2_{S,0}$ | 104.21 | 105.35 |
| B | log_nbpars | 32.09 | 33.00 |
| BxA | $\sigma^2_{S,0}$ | 6.15 | 5.56 |
| BxA | log_nbpars | 4.83 | 4.53 |
| BxAxC | $\sigma^2_{S,0}$ | 3.92 | 0.00 |
| BxAxC | log_nbpars | 7.20 | 10.34 |
| BxAxD | $\sigma^2_{S,0}$ | 0.82 | 0.48 |
| BxAxD | log_nbpars | 0.90 | 0.52 |
| BxC | $\sigma^2_{S,0}$ | 4.40 | 10.36 |
| BxC | log_nbpars | 3.47 | 8.39 |
| BxCxD | $\sigma^2_{S,0}$ | 0.50 | 0.63 |
| BxCxD | log_nbpars | 4.93 | 2.92 |
| BxD | $\sigma^2_{S,0}$ | 11.65 | 8.02 |
| BxD | log_nbpars | 1.90 | 1.34 |
| C | $\sigma^2_{S,0}$ | 5.35 | 10.75 |
| C | log_nbpars | 17.75 | 38.53 |
| CxAxD | $\sigma^2_{S,0}$ | 6.36 | 22.98 |
| CxAxD | log_nbpars | 7.42 | 19.06 |
| CxD | $\sigma^2_{S,0}$ | 1.88 | 0.98 |
| CxD | log_nbpars | 1.38 | 1.03 |
| D | $\sigma^2_{S,0}$ | 0.00 | 1.86 |
| D | log_nbpars | 3.26 | 0.60 |
| Residual | | 33.24 | 32.41 |

Note: Variance components are reported as raw estimated random effect variances (Eq. (3)).

# H. Beyond Sum Scores: Modeling Latent Structure and its Covariates

## H.1. Hierarchical Mixed-Effects Latent Regression

### H.1.1. MODEL SPECIFICATION

To simultaneously model the latent dimensional structure, the hierarchical dependencies in the data, and the influence of external covariates, we employ a Hierarchical Mixed-Effects Latent Regression Model. This model extends the bifactor CFA structure by treating the latent traits themselves as outcomes in a mixed-effects regression. This allows us to parse the variance in each latent ability into components attributable to fixed effects (like model size), random effects (like contributor-specific variance), and residual unexplained variance. The model is composed of two interconnected components: a measurement model (Eq. (4)) that links observed item responses to latent traits, and a structural model (Eq. (5)) that explains the variance in those latent traits.

### H.1.2. BIFACTOR IRT MEASUREMENT WITH MIXED-EFFECTS LATENT REGRESSION

To support readers who may be more comfortable with more explicit notation, we redefine the relationships of (4) and (5). Let $y_{ij} \in \{0, 1\}$ denote the response of model submission $i \in \{1, \ldots, N\}$ to item $j \in \{1, \ldots, p\}$. Items are partitioned into benchmarks $k(j) \in \{1, \ldots, K\}$. We use a bifactor measurement model with one general factor and $K$ benchmark-specific factors. Let the latent ability vector be

$$\boldsymbol{\theta}_i = \begin{bmatrix} \theta_{i0} \\ \theta_{i1} \\ \vdots \\ \theta_{iK} \end{bmatrix} \in \mathbb{R}^{K+1}, \ \theta_{i0} = \text{general ability}, \ \theta_{ik} = \text{benchmark-specific ability}. \tag{34}$$

For item $j$, define discrimination parameters $(a_{j0}, a_{j,k(j)})$ and difficulty $b_j$. Under a 2PL bifactor IRT model in (4),

$$\Pr(y_{ij} = 1 \mid \boldsymbol{\theta}_i) = \sigma\big(a_{j0}\theta_{i0} + a_{j,k(j)}\theta_{i,k(j)} - b_j\big),$$

where $\sigma(t) = (1 + e^{-t})^{-1}$. The key extension is a *mixed-effects regression* for each latent dimension that accounts for ecosystem clustering and covariates. Let $x_i = \log_{10}(\#\text{params})$ and let $\mathbf{w}_i$ collect deployment covariates (e.g., type, chat_template, mo_e). Let $c(i)$ denote the contributor/provenance label used in Method 2. We can reformulate the model latent abilities in (5) as vectors

$$\boldsymbol{\theta}_i = \mathbf{B}\mathbf{z}_i + \mathbf{U}\mathbf{u}_{c(i)} + \boldsymbol{\varepsilon}_i, \qquad \mathbf{z}_i = \begin{bmatrix} 1 & x_i & \mathbf{w}_i^\top \end{bmatrix}^\top, \tag{35}$$

where $\mathbf{B} \in \mathbb{R}^{(K+1)\times(2+d)}$ are fixed effects (including the scaling slope for each latent dimension), $\mathbf{u}_c \sim \mathcal{N}(\mathbf{0}, \boldsymbol{\Sigma}_u)$ are contributor-level random effects, and $\boldsymbol{\varepsilon}_i \sim \mathcal{N}(\mathbf{0}, \boldsymbol{\Sigma}_\varepsilon)$ captures remaining between-submission latent variation. (Additional random effects for architecture or deployment can be incorporated analogously; in our main specification we include contributor as the dominant non-i.i.d. source, consistent with Method 2.)

### H.1.3. ESTIMATION AND EVALUATION PROTOCOL

We estimate (4)–(5)/(35) with the Metropolis-Hastings Robbins-Monro (MH-RM) algorithm (Cai, 2010a;b; Chalmers, 2015), implemented in mirt (Chalmers, 2012). We fit (i) a baseline bifactor IRT model (measurement only) and (ii) the latent mixed regression bifactor (measurement + covariates + random effect slopes). To ensure conclusions are not driven by a particular subset of items, we repeat estimation across $B = 500$ item bootstraps (sampling a fixed number of items per benchmark as in Method 1).

Within each bootstrap, we extract the fixed-effect coefficients ($\hat{\boldsymbol{\Gamma}}$) the latent ability scores ($\hat{\boldsymbol{\Theta}}$), and their standard errors for each model under both the regression and non-regression specifications, and evaluate:

1. **Model comparability:** likelihood-based comparison between (i) and (ii) to verify that adding covariates/random effects improves fit without distorting the bifactor measurement structure.
2. **Noise-controlled scaling slopes:** estimates $\{\hat{\beta}_d\}_{d=0}^K$ and their uncertainty, aggregated across bootstraps via inverse-variance weighting.
3. **Provenance variance:** $\boldsymbol{\Sigma}_u$ as the contribution of contributor/provenance to latent variability after controlling for size and deployment.

4. **Stability diagnostics:** bootstrap distributions of $\hat{\beta}_d$ and (when applicable) $\hat{\mathcal{R}}_d$.

### H.1.4. RELIABILITY OF SCALING EFFECTS UNDER PSEUDO-REPLICATION

The object of interest is the scaling law slope on a latent dimension $d \in \{0, \dots, K\}$, $\beta_d \equiv (\mathbf{B})_{d,\,x}$, and its reliability. As with all aggregations in this study of estimated parameters, inverse squared-standard error weighted means provide the precision-weighted average: $w_i = 1/\operatorname{SE}(\beta_{di})^2$. We quantify slope stability with a *random-slope extension*:

$$\theta_{id} = \alpha_d + \beta_d x_i + \mathbf{w}_i^\top \boldsymbol{\gamma}_d + u_{c(i),d} + v_{c(i),d} x_i + \varepsilon_{id}, \tag{36}$$

with $\operatorname{Var}(v_{\cdot,d}) = \tau_d^2$, the weighted variance of the dimension. We then report a *slope reliability index* in Table 3:

$$\mathcal{R}_d \equiv \frac{\beta_d^2}{\beta_d^2 + \tau_d^2}, \tag{37}$$

### H.2. Results and discussion (Method 3): noise-controlled scaling laws differ by latent ability

#### H.2.1. QUANTIFYING PROVENANCE: TRAINING SIGNAL VERSUS MEASUREMENT NOISE

A key advantage of the mixed-effects model is its ability to quantify the variance attributable to unobserved, systematic differences between contributors. Even after controlling for item effects, model size, and deployment type, in (4)-(5), the inverse-error weighted median random effect for `contributor` still accounts for 3.0% of the variance in latent ability scores.

This "provenance variance" is not merely measurement noise. It represents a significant signal about the efficacy of unobserved training methods, data mixtures, and resource allocation associated with different model providers. From the perspective of evaluating a base architecture (e.g., assessing `Llama-3-70B`'s inherent capabilities), this contributor-specific "secret sauce" is a confounding factor. However, from the perspective of advancing the field, this variance is a critical indicator of where the most effective (and often proprietary) training innovations lie. Our model successfully isolates this signal, allowing for a clearer interpretation of both architectural capabilities and the impact of training methodology.

#### H.2.2. SIGNIFICANCE: SCALING IS NOT A SINGLE LAW BUT A VECTOR OVER LATENT ABILITIES

Method 3 reframes "does performance scale with size?" into a dimension-specific statement: $\boldsymbol{\beta} = (\beta_0, \beta_1, \dots, \beta_K)$ is a *scaling vector* over latent abilities, not a scalar property of a benchmark average. The observed pattern—strong scaling for $g$, heterogeneous and sometimes negligible scaling for benchmark-specific factors, and persistent provenance variance—explains why leaderboard-based scaling curves can be simultaneously predictive (for aggregate scores) and misleading (for capability-specific claims).

Practically, this implies that benchmark ecosystems should report, at minimum, (i) a general-factor scaling curve, (ii) benchmark-specific residual scaling curves after removing $g$, and (iii) a provenance-adjusted uncertainty estimate (e.g., via $N_{\text{eff}}$ or random-slope reliability $\mathcal{R}_d$). Without these, apparent "laws" may reflect measurement structure and ecosystem composition rather than transferable properties of model scaling.

# I. Code Listings

The bifactor model definitions are included here as well as methods 2 and 3 from the paper. All other code containing presented statistical modeling and estimations can be found in the online repository.[11]

## I.1. Representative CFA Model: Bifactor Structure

The bifactor model is our recommended structure. Below is the `lavaan` syntax generator. Other model specifications (unidimensional, correlated factors, hierarchical) follow similar patterns and are available in our code repository.

*Listing 1.* `lavaan` Bifactor Model Syntax

```
generate_bifact_syntax <- function(item_map) {
```

---

[11]https://github.com/hardy-education/ai_bench_cfa

```
  benchmarks <- unique(item_map$benchmark)
  items <- item_map$item_full

  # General factor loads on all items
  general_spec <- paste0("general =~ ", paste(items, collapse = " + "))

  # Specific factors for each benchmark
  specific_specs <- map_chr(benchmarks, function(b) {
    items_b <- item_map %>% filter(benchmark == b) %>% pull(item_full)
    paste0(b, " =~ ", paste(items_b, collapse = " + "))
  })

  # Orthogonalize: general uncorrelated with specifics
  orthog_general <- map_chr(benchmarks, function(b) {
    paste0("general ~~ 0*", b)
  })

  # Orthogonalize specifics with each other
  orthog_specific <- combn(benchmarks, 2, function(pair) {
    paste0(pair[1], " ~~ 0*", pair[2])
  }, simplify = FALSE) %>% unlist()

  paste0("# Bifactor Model\n",
    general_spec, "\n",
    paste(specific_specs, collapse = "\n"), "\n",
    paste(orthog_general, collapse = "\n"), "\n",
    paste(orthog_specific, collapse = "\n"), "\n")
}
```

### I.2. Bifactor via MH-RM and Latent Regression

Below is the `mirt` syntax generator for the bifactor. Other model specifications (unidimensional, correlated factors, hierarchical) follow similar patterns and are available in our code repository. Included in the code listing here is the MH-RM estimation.

*Listing 2.* `mirt` Bifactor Model and Bifactor Latent Mixed Effect Regression Syntax and Estimation

```
#' @param benches Vector of benchmark names
#' @param n_items Number of items per benchmark
#' @return Character string for factor specification
generate_bifactor_string <- function(benches, n_items) {
  item_nums <- seq_len(n_items)
  bench_nums <- seq_along(benches)

  factor_specs <- sapply(bench_nums, function(x) {
    min_item <- min(item_nums) + (x - 1) * length(item_nums)
    max_item <- max(item_nums) + (x - 1) * length(item_nums)
    glue::glue("{benches[x]} = {min_item}-{max_item}")
  })
  general_str <- glue::glue("G = 1-{length(item_nums)}")
  paste(general_str, factor_specs, collapse = "\n")
}
dat = sdf %>% select(-any_of(names(metadf))) # response matrix
covdat = sdf %>% select(any_of(names(metadf))) # covariates
# estimate bifactor model and latent regression bifactor
bifact = dat |>
    mirt::mixedmirt(model = generate_bifactor_string(b,ni),
    itemtype = "2PL", SE = T, fixed = ~ 0 + items,
    accelerate = "squarem", GenRandomPars = TRUE,
    technical = list(NCYCLES = 5000))
bifact_regression = dat |>
    mirt::mixedmirt(model = generate_bifactor_string(b,ni),
    covdata = covdat, itemtype = "2PL", SE = T,
```

```
      fixed = ~ 0 + items, random = ~ 1|contrib,
      lr.fixed = ~ type + chat_template + mo_e + log_nbpars,
      accelerate = "squarem", GenRandomPars = TRUE,
      technical = list(NCYCLES = 5000))
}
```

## I.3. Variance Decomposition Model

The G-theory variance decomposition uses a fully-crossed random-effects model with random slopes for model size.

*Listing 3.* Generalizability Study: Full Random-Slopes Model

```
lmer(formula = val ~ 1
    + log_nbpars
    + (log_nbpars|bench) # (B) Benchmark
    + (log_nbpars|architecture:generation) # (A) Architecture
    + (log_nbpars|author:removed:not_avail) # (C) Contributor
    + (log_nbpars|type:chat_template:mo_e:merged:precision) # (D) Deployment and Tuning
    + (log_nbpars|bench:architecture:generation) # AxB
    + (log_nbpars|bench:author:removed:not_avail) # BxC
    + (log_nbpars|bench:type:chat_template:mo_e:merged:precision) # BxD
    + (log_nbpars|architecture:generation:author:removed:not_avail) # AxC
    + (log_nbpars|architecture:generation:type:chat_template:mo_e:merged:precision) # AxD
    + (log_nbpars|author:removed:not_avail: type:chat_template:mo_e:merged:precision) # CxD
    + (log_nbpars|bench:architecture:generation: type:chat_template:mo_e:merged:precision)
        ↪ # AxBxD
    + (log_nbpars|bench:author:removed:not_avail: type:chat_template:mo_e:merged:precision)
        ↪  # BxCxD
    + (log_nbpars|architecture:generation:author:removed:not_avail: type:chat_template:mo_e
        ↪ :merged:precision) # AxCxD
    + (log_nbpars|bench:architecture:generation:author:removed:not_avail), # AxBxC
    data = df)
```

For the reduced model discussed in the main body of the paper, replacing the single |with || specifies uncorrelated random intercepts and slopes. Each facet (A, B, C, D) and their interactions contribute both intercept variance (baseline performance) and slope variance (scaling relationship heterogeneity).

# J. Extended Results and Discussion

## J.1. Quantifying Noise Source Magnitudes

### J.1.1. WITHOUT SIZE CONTROL, SIGNAL IS BURIED IN CONFOUNDED NOISE

Introducing model size (log_nbpars) as a covariate dramatically clarifies the sources of variation, as shown in Table 14. The residual variance drops by 36% (from 52.0 to 33.24 in the full model summary), indicating that model size explains a substantial portion of previously unexplained performance differences—without this control, architectural signal is confounded with scale.

We calculate (33) to be $\Omega_{\text{size-all}} = 0.402$, suggesting that about 40% of all observed variation is linked to model size. This confirms that, without explicit size control, aggregate rankings primarily reflect scale rather than architectural innovation; "size is nearly all you need" when noise dominates.

Moreover, the influence of other facets is reshaped by their relationship to model size. The Contributor facet (C) has the largest slope variance ($\sigma_{C,0}^2 = 17.75$) among all non-benchmark facets, indicating that the performance gap between different contributors widens significantly as model size increases. In contrast, the Architecture facet (A) has a larger intercept variance ($\sigma_{A,\text{intcpt}}^2 = 4.67$) than slope variance ($\sigma_{A,\text{slope}}^2 = 14.06$), though both are substantial. This suggests that while some architectures have a higher performance baseline independent of size, the primary differentiator remains how effectively they scale. The main effect of model size is strongly positive and significant, confirming that, on average, larger models perform better.

## J.2. Actionable Recommendations

### J.2.1. TOOLS FOR MOVING THE FIELD FORWARD

Our framework provides concrete tools for noise control. We frame these constructively as methods practitioners can adopt to improve measurement quality:

**For Leaderboard Operators**

- **Decompose and report variance sources:** Use G-theory variance decomposition (Table 12 provides a template) to show users what proportion of ranking variance comes from benchmarks, contributors, architecture, and scale.
- **Provide reliability context:** Display confidence intervals or G-coefficients alongside ranks to indicate trustworthiness. Reliability is conditional on use case (within- vs. across-contributor, size-controlled vs. uncontrolled).

**For Scaling Law Analysis**

- **Report SNR$_\beta$ or $\mathcal{R}_\beta$ alongside $\beta$:** A large average effect $\beta$ with low SNR$_\beta$ ($\approx 1$) indicates scaling is context-dependent—the relationship is unreliable across ecosystem facets.
- **Use PSI to diagnose instability:** Our analysis reveals Benchmark PSI (30%) as the largest source of scaling law unreliability. Different benchmarks respond differently to size increases—acknowledge this heterogeneity.

The key insight: *reliability is not binary but quantifiable and improvable*. Our metrics (e.g., SNR$_\beta$, PSI) provide numerical targets for noise control efforts.

## K. Consolidated Notation

Tables 15 and 16 provide a comprehensive reference for all mathematical notation used throughout the paper and appendices.

*Table 15.* Comprehensive notation reference. Symbols are grouped by the model layer in which they primarily appear.

| Symbol | Domain | Description |
|---|---|---|
| *Observed data* | | |
| $y_{ij}, y_{ikj}$ | $\{0, 1\}$ | Binary response of LLM $i$ to item $j$ (in benchmark $k$) |
| $\mathbf{y}_i$ | $\{0, 1\}^p$ | Full response vector for LLM $i$ across $p$ items |
| $y_{ij}^{\star}$ | $\mathbb{R}$ | Underlying continuous latent response (probit/logit link) |
| $s_i$ | $\mathbb{R}$ | Manifest (observed) benchmark score for submission $i$ |
| $\mathbf{S}$ | $\mathbb{R}^{p \times p}$ | Observed sample covariance/correlation matrix |
| $x_i$ | $\mathbb{R}$ | Log-size covariate: $\log_{10}(\#\text{parameters}_i)$ |
| $N$ | $\mathbb{N}$ | Number of LLM submissions |
| $p$ | $\mathbb{N}$ | Total number of items across all benchmarks |
| $K$ | $\mathbb{N}$ | Number of benchmarks (= number of specific factors) |
| $B$ | $\mathbb{N}$ | Number of bootstrap replications |
| $r_k$ | $\mathbb{N}$ | Number of items sampled per benchmark $k$ per bootstrap |
| *CFA / SEM measurement model (Method 1)* | | |
| $\boldsymbol{\eta}_i$ | $\mathbb{R}^m$ | Latent factor score vector for LLM $i$ |
| $\boldsymbol{\nu}$ | $\mathbb{R}^p$ | Vector of item intercepts |
| $\boldsymbol{\Lambda}$ | $\mathbb{R}^{p \times m}$ | Factor loading matrix (items $\rightarrow$ factors) |
| $\lambda_{kj}$ | $\mathbb{R}$ | Loading of item $j$ on factor $k$ |
| $\boldsymbol{\Phi}$ | $\mathbb{R}^{m \times m}$ | Factor covariance matrix; $\boldsymbol{\Phi} = \text{Cov}(\boldsymbol{\eta})$ |
| $\boldsymbol{\Theta}$ | $\mathbb{R}^{p \times p}$ | Residual (unique) covariance matrix; typically diagonal |
| $\Theta_{ab}$ | $\mathbb{R}$ | Residual covariance between items $a$ and $b$ |
| $\boldsymbol{\Sigma}(\boldsymbol{\theta})$ | $\mathbb{R}^{p \times p}$ | Model-implied covariance: $\boldsymbol{\Lambda}\boldsymbol{\Phi}\boldsymbol{\Lambda}^{\top} + \boldsymbol{\Theta}$ |
| $\boldsymbol{\theta}$ | — | Collected free parameters: $\{\boldsymbol{\Lambda}, \boldsymbol{\Phi}, \boldsymbol{\Theta}\}$ |
| $\boldsymbol{\Psi}$ | $\mathbb{R}^{m \times m}$ | Correlation matrix of latent factors (in correlated-factor models) |
| $F(\cdot, \cdot)$ | $\mathbb{R}_{\geq 0}$ | Discrepancy function between $\boldsymbol{\Sigma}(\boldsymbol{\theta})$ and $\mathbf{S}$ |
| $s \in \mathcal{S}$ | — | Candidate CFA structure (one of six) |
| *Bifactor-specific notation* | | |
| $g_i$ | $\mathbb{R}$ | General factor score for LLM $i$ |
| $f_{ik}, s_{ik}$ | $\mathbb{R}$ | Specific (benchmark) factor score; $k$-th factor for LLM $i$ |
| $\lambda_{kj}^{(g)}$ | $\mathbb{R}$ | Loading of item $j$ on the general factor $g$ |
| $\lambda_{kj}^{(s)}$ | $\mathbb{R}$ | Loading of item $j$ on specific factor $s_k$ |
| $\mathbf{f}_i$ | $\mathbb{R}^K$ | Vector of specific factor scores for LLM $i$ |
| *Hierarchical / second-order model* | | |
| $\gamma_k$ | $\mathbb{R}$ | Loading of first-order factor $f_k$ on second-order factor $g$ |
| $\delta_{ik}$ | $\mathbb{R}$ | First-order factor disturbance |
| $\boldsymbol{\Delta}$ | $\mathbb{R}^{K \times K}$ | Disturbance covariance matrix |
| *Local dependence diagnostics* | | |
| $\psi$ | $\mathbb{R}$ | Candidate constrained parameter (residual covariance $\Theta_{ab}$) |
| $U_\psi(\hat{\boldsymbol{\theta}})$ | $\mathbb{R}$ | Score (gradient) for $\psi$ at constrained optimum |
| $\mathcal{I}$ | $\mathbb{R}^{q \times q}$ | Expected Fisher information matrix |
| $\mathcal{I}_{\psi\psi \cdot \theta}$ | $\mathbb{R}$ | Schur complement (partial information for $\psi$) |
| $\text{MI}(\psi)$ | $\mathbb{R}_{\geq 0}$ | Modification index (LM test statistic); $\text{MI}(\psi) \xrightarrow{d} \chi_1^2$ |
| $\text{EPC}(\psi)$ | $\mathbb{R}$ | Expected Parameter Change upon freeing $\psi$ |
| SEPC | $[-1, 1]$ | Standardized EPC (on correlation scale) |
| *Meta-analytic bootstrap aggregation* | | |
| $t_{b,s,r}$ | $\mathbb{R}$ | Fit statistic under condition $r$ (true vs. permuted) |
| $\alpha_s$ | $\mathbb{R}$ | Intercept: expected fit of structure $s$ under true assignment |
| $\beta_s$ | $\mathbb{R}$ | Permutation penalty: fit loss from randomized assignment |
| $\Delta T_s^{(b)}$ | $\mathbb{R}$ | $T(s \mid \pi^{(b)}) - T(s \mid \text{id})$; structural signal beyond flexibility |
| $\pi^{(b)}$ | — | Random item-to-benchmark permutation in bootstrap $b$ |
| $u_b, v_b$ | $\mathbb{R}$ | Random intercept/slope for bootstrap $b$ |

*Table 16.* Comprehensive notation reference. Symbols are grouped by the model layer in which they primarily appear.

| Symbol | Domain | Description |
|---|---|---|
| *G-theory variance decomposition (Method 2)* | | |
| $\mu$ | $\mathbb{R}$ | Grand mean score |
| $a(i), b(i), c(i), d(i)$ | — | Facet levels: Architecture, Benchmark, Contributor, Deployment |
| $\mathcal{S}$ | — | Set of random-effect terms (main effects + interactions) |
| $u_{g_S}^S$ | $\mathbb{R}$ | Random intercept for level $g_S$ of term $S$ |
| $v_{g_S}^S$ | $\mathbb{R}$ | Random slope for level $g_S$ of term $S$ |
| $\sigma_{S,0}^2$ | $\mathbb{R}_{\geq 0}$ | Intercept variance component for term $S$ |
| $\sigma_{S,1}^2$ | $\mathbb{R}_{\geq 0}$ | Slope variance component for term $S$ |
| $\sigma_\varepsilon^2$ | $\mathbb{R}_{\geq 0}$ | Residual variance |
| $p_S$ | $[0, 1]$ | Variance share attributable to term $S$ |
| $\beta$ | $\mathbb{R}$ | Fixed effect of log-size on score |
| $\mathrm{SNR}_\beta$ | $\mathbb{R}_{\geq 0}$ | Slope signal-to-noise ratio: $\beta^2 / \sum_S \sigma_{S,1}^2$ |
| $\mathrm{PSI}_S$ | $[0, 1]$ | Proportion of slope instability from term $S$ |
| $R_\beta$ | $[0, 1]$ | Reliability coefficient for the fixed slope $\beta$ |
| $\mathrm{CV}_\beta$ | $\mathbb{R}_{\geq 0}$ | Coefficient of variation of effective slopes |
| *Latent mixed-effects regression (Method 3)* | | |
| $\boldsymbol{\theta}_i$ | $\mathbb{R}^{K+1}$ | Latent ability vector: $(\theta_{i0}, \theta_{i1}, \ldots, \theta_{iK})^\top$ |
| $\theta_{i0}$ | $\mathbb{R}$ | General ability for LLM $i$ |
| $\theta_{ik}$ | $\mathbb{R}$ | Benchmark-specific ability for LLM $i$, benchmark $k$ |
| $a_{j0}$ | $\mathbb{R}_{>0}$ | Item $j$ discrimination on the general factor |
| $a_{j,k(j)}$ | $\mathbb{R}_{>0}$ | Item $j$ discrimination on its benchmark-specific factor |
| $b_j$ | $\mathbb{R}$ | Item $j$ difficulty |
| $\boldsymbol{\Theta}$ | $\mathbb{R}^{N \times (K+1)}$ | Matrix of all latent traits |
| $\mathbf{V}$ | $\mathbb{R}^{N \times F}$ | Fixed-effect design matrix |
| $\boldsymbol{\Gamma}$ | $\mathbb{R}^{F \times (K+1)}$ | Fixed-effect coefficient matrix |
| $\beta_k$ | $\mathbb{R}$ | Effect of log-size on latent dimension $k$: $\Gamma_{x,k}$ |
| $\boldsymbol{\beta}$ | $\mathbb{R}^{K+1}$ | Scaling vector: $(\beta_0, \beta_1, \ldots, \beta_K)$ |
| $\mathbf{W}$ | $\mathbb{R}^{N \times R}$ | Random-effect design matrix (contributor grouping) |
| $\boldsymbol{\zeta}$ | $\mathbb{R}^{R \times (K+1)}$ | Contributor-level random effects |
| $\boldsymbol{\Sigma}_\zeta$ | — | Covariance of $\mathrm{vec}(\boldsymbol{\zeta})$; block-diagonal |
| $\boldsymbol{\Sigma}_u$ | — | Provenance variance (contributor random-effect covariance) |
| $\boldsymbol{\Sigma}_E$ | $\mathbb{R}^{(K+1) \times (K+1)}$ | Residual latent covariance |
| $\mathbf{E}$ | $\mathbb{R}^{N \times (K+1)}$ | Residual latent variation |
| $\mathbf{B}$ | $\mathbb{R}^{(K+1) \times (2+d)}$ | Fixed-effect matrix in IRT parameterization |
| $\mathbf{z}_i$ | $\mathbb{R}^{2+d}$ | Covariate vector: $[1, \ x_i, \ \mathbf{w}_i^\top]^\top$ |
| $\mathbf{w}_i$ | $\mathbb{R}^d$ | Deployment covariate vector for LLM $i$ |
| $\mathbf{u}_{c(i)}$ | $\mathbb{R}^{K+1}$ | Contributor random-effect vector for LLM $i$ |
| $\mathcal{R}_d$ | $[0, 1]$ | Slope reliability index: $\beta_d^2 / (\beta_d^2 + \tau_d^2)$ |
| $\tau_d^2$ | $\mathbb{R}_{\geq 0}$ | Random-slope variance for latent dimension $d$ |
| *Fit indices and model comparison* | | |
| RMSEA | $\mathbb{R}_{\geq 0}$ | Root Mean Square Error of Approximation |
| CFI | $[0, 1]$ | Comparative Fit Index |
| TLI | $[0, 1]$ | Tucker-Lewis Index |
| SRMR | $\mathbb{R}_{\geq 0}$ | Standardized Root Mean Square Residual |
| AIC, BIC | $\mathbb{R}$ | Akaike / Bayesian Information Criteria |
| AUC | $[0, 1]$ | Area Under the ROC Curve (out-of-sample prediction) |
| MAE | $\mathbb{R}_{\geq 0}$ | Mean Absolute Error (out-of-sample prediction) |
| LRT | $\mathbb{R}_{\geq 0}$ | Likelihood Ratio Test statistic |
| *General conventions* | | |
| $\tau_j$ | $\mathbb{R}$ | Item threshold (probit parameterization) |
| $\mathcal{P}^*(\cdot)$ | — | Non-empty Power Set |

