# OpenReview forum: "AI Cartography: Mapping the Latent Landscape of AI Benchmark Ecosystems"
_ICML.cc/2026/Conference — ICML 2026 regular_

### Official Review · Reviewer_sdUW · 2026-03-11

**Soundness:** 4
**Presentation:** 1
**Significance:** 4
**Originality:** 3
**Overall Recommendation:** 5
**Confidence:** 1

**Summary:**

The paper addresses the problem of modern LLM leaderboards getting treated as if they were clean, single-number measurements of model capability. It proposes a psychometric framework for quantifying the measurement noise or ecosystem artifacts in such metrics. This framework applies three complementary methods to “map” and quantify noise, and therefore makes rankings and claims about scaling interpretable and actionable.

**Compliance With Llm Reviewing Policy:**

Affirmed.

**Final Justification:**

I decided to raise my score, as I did not identify any substantial weaknesses, and the authors adequately addressed the weaknesses regarding formatting and presentation.

**Key Questions For Authors:**

I have no questions for the authors at this point.

**Limitations:**

yes

**Strengths And Weaknesses:**

# Strengths and Weaknesses
- The paper is ambitious and well-motivated, and the proposed framework directly addresses the issues mentioned. It also offers actionable guidance for benchmark design. The experimental setup and scope seems adequate.
- I only identified substantial weaknesses related to formatting and presentation, which I describe below. However, my understanding of the field is quite shallow, as indicated by my confidence rating.

# Formatting and Presentation
- The paper is quite difficult to follow for readers not familiar with underlying concepts, as they are not explained
- The paper contains many formatting errors, such as broken references (causing artifacts like “Appendix ??”), incorrect opening quotation marks, or asterisks in the main text (e.g., “\*how\*” in line 396).
- The paper uses quotation marks and em dashes excessively. Further, it mixes bold and italics (or applies both simultaneously) for emphasis, which is irritating.
- “Open LLM” and “OpenLLM” are used interchangeably throughout the paper. The same goes for “Hugging Face” and “Huggingface”
- Appendix section heading formatting is inconsistent and partially incorrect.
- The appendix is quite excessive and can probably be condensed.

---

> ### Author Rebuttal · Authors · 2026-03-30
>
> Thank you so much for your insightful and thoughtful review of the paper's potential, motivation, soundness, and significance. We worked hard to directly address these targets and offer actionable guidance as this was key for us, too. We recognize your presentation questions are key areas for crystalizing the intended impact for our broader ML community, so our intended solutions and actionability are helpfully communicated.
> # LaTeX Compilation Typo
> We must transparently disclose that a single-character typo in our LaTeX source prevented the correct appendix sections from loading correctly, producing the "??" artifacts throughout the submitted PDF (broken table references, appendix pointers, and code listing references). This single error cascades into what appears to be pervasive carelessness but is in fact a single compilation failure affecting all missing `\ref{}` calls. We take full responsibility for not catching this before submission. The underlying content–every referenced table, appendix section, and code listing–exists in full and is internally consistent. We will be relieved to make this revision.
>
> # Building Familiarity
> We appreciate this feedback and recognize that CFA, G-theory, and IRT are not as standard in our ML community. We agree: when performing analyses using methods that less common in our discipline the authors need to do more to support readers.   In revision, we commit to:
> - Including a notation table consolidating all symbols (𝛬,𝛷,𝛩,𝜂, etc.) with plain-English descriptions with what they represent.
> - Adding an intuitive overview paragraph at the start of §2 that explains the core ideas in plain language before introducing formalism. For instance:
> > CFA tests which underlying capacities most likely give rise to the patterns of observed model responses among benchmark items; G-theory quantifies the proportion of observed variation in scores attributable to components within the ecosystem (benchmark, contributor, architecture, deployment); and, building on those, latent mixed regression asks how do model descriptors like size and fine-tuning approaches affect benchmark performance after controlling for sources of ecosystem variation.
> - Adding explanatory footnotes to provide additional support or to remove some formalisms from the main text in order to focus on the purpose of each. For example, we could explain one of the more complex tests in our study by moving lines 132-139 to footnote, to free up space to explain what the MI tests are conceptually. The following would be accompanied by a small figure illustrating:
> > MI tests can be visualized as comparing a diagram from Fig. 2, $Q$, with a copy of that same structure, $Q’$, which has _one additional (parameter) line_ connecting two benchmark items in the diagram. Comparing $Q’$ and $Q$ is now a 1-df ablation of a single parameter capturing residual covariance. Permuting all possible new connections (and comparing with $Q$) across bootstraps can reveal whether structure $Q$ tends to over- or underestimate the true relationship between items and constructs.
>
> We hope these additions will be more responsive to the needs of our ICML community and will make the paper substantially more accessible helping our expositional content match the quality of conceptual content.
>
> # Formatting
> We sincerely appreciate the meticulous review of our formatting and syntax. We have completed a full audit of the manuscript and commit to the following corrections in the revision:
>
> * Emphasis & Quotations: Removed all instances of mixed bold-italics (e.g., lines 096, 2026), non-standard bolding (e.g., lines 387, 412), and quotations used for emphasis (e.g., lines 034, 055, 326), moving to a standard italics-only convention.
> * Punctuation: Corrected all non-directional opening quotation marks (e.g., lines 400, 422) and reduced the use of em-dashes by 75% to improve readability (e.g., 718, 720, 946).
> * Naming Conventions: Standardized spellings to "Hugging Face" and "Open LLM Leaderboard" throughout (including Fig. 2).
>
> # Appendix Length and Organization
> We agree the 30-page appendix is substantial. Our goal was to provide complete mathematical proofs for measurement scientists alongside reproducible code for ML practitioners. To improve navigability and reduce the length by ~20%, we will:
> * Combine §A and §B (background for measurement science), reducing the content by half.
> * Combine § I and §J (extended results and discussion), reducing the content by half.
> * Merge structural model supports currently separated in §B.8 and §D.1.
> * Standardize the heading hierarchy in §H and the mathematical enumeration (theorems, proofs, definitions) in §C.
> * Move reproducible code listings and extended resources to a GitHub repository and supplementary materials.
> * Add targeted cross-references to help readers unfamiliar with the content navigate quickly to supportive material.

---

> > ### Author Rebuttal · Reviewer_sdUW · 2026-04-03
> >
> > The authors have fully resolved my concerns regarding formatting and presentation. Since I do not identify any remaining substantial weaknesses, I have decided to raise my score.

---

> > > ### Author Response · Authors · 2026-04-06
> > >
> > > Thank you very much for your comments and revised score. Our goal is to ensure our work is accessible to as broad an ICML audience as possible, and we believe incorporating the substantive structural changes to the paper based on your feedback will make that goal achievable. Thank you for your service to the larger community.

---

### Official Review · Reviewer_gr11 · 2026-03-12

**Soundness:** 3
**Presentation:** 3
**Significance:** 3
**Originality:** 3
**Overall Recommendation:** 6
**Confidence:** 3

**Summary:**

### Summary.

This paper addresses the problem of over-reliance on aggregated sum-scores in AI benchmark ecosystems (specifically the Open LLM Leaderboard) and the unquantified measurement noise they contain. By combining Confirmatory Factor Analysis (CFA), Generalizability Theory (G-theory), and Hierarchical Mixed-Effects Latent Regression, the authors systematically deconstruct the sources of variance in leaderboard rankings across 4,000+ models and 6 benchmarks. Specifically, the authors apply a rigorous psychometric framework to the Open LLM Leaderboard using three primary methods:
- Confirmatory Factor Analysis (CFA): Demonstrates that the latent structure of model capabilities is best described by a bifactor model (a dominant general factor $g$ plus specific residual benchmark factors), refuting naive assumptions of independent benchmark modules.
- Generalizability Theory (G-Theory): Conducts a variance decomposition revealing that the "Contributor" (provenance) accounts for nearly twice as much variance (9.0%) as the base "Architecture" (4.8%), while model size accounts for 36% of the explainable variance.
- Hierarchical Mixed-Effects Latent Regression: Estimates "noise-controlled" scaling laws. The authors show that while general capability ($g$) and knowledge (MMLU-Pro) scale robustly with parameter count, specific abilities like instruction-following (IFEval) and soft reasoning (MuSR) exhibit weak, negligible, or negative scaling once deployment and provenance are controlled.

**Compliance With Llm Reviewing Policy:**

Affirmed.

**Final Justification:**

Thank you to the authors for the thoughtful and comprehensive rebuttal. I appreciate you directly addressing my concerns and committing to the necessary clarifications in the final manuscript. I am entirely satisfied with your responses and will increase my score accordingly. Best of luck with the rest of the review process!

**Key Questions For Authors:**

1. While the Bifactor models are clearly the relatively best models, the authors must explicitly address why the absolute fit is so poor. Is this driven by the extreme sparsity of the dichotomous data, the massive sample size inflating the $\chi^2$ statistic, or severe unmodeled local dependencies (e.g., shared prompting artifacts across the EleutherAI harness)? Preempting this psychometric critique in the paper is crucial.

2. Can you briefly discuss how the MH-RM algorithm handles multicollinearity? Does the model risk over-penalizing the size or architecture effects by absorbing that legitimate signal into the "Contributor" random effect?

**Limitations:**

yes

**Strengths And Weaknesses:**

Strengthes
- The ML community heavily relies on aggregate scores and scaling laws derived from observational leaderboards. Addressing the "pseudo-replication" problem (the non-i.i.d. nature of fine-tunes, merges, and quantizations) is a necessary and highly impactful contribution.
- The synthesis of Item Response Theory (IRT), CFA, and G-Theory via a Metropolis-Hastings Robbins-Monro (MH-RM) mixed-effects regression is statistically mature and highly appropriate for the complex, hierarchical data of the Hugging Face Hub.
- The finding that scaling laws are a vector across latent abilities rather than a scalar—and showing empirically that certain capabilities do not naturally scale with parameter count when noise is controlled—challenges the prevailing "scale is all you need" narrative in a mathematically rigorous way.

Weaknesses
- In Table 1, the authors report the RMSEA for the candidate latent structures. The recommended Orthogonal Bifactor model has an RMSEA of 0.25, and the Correlated Bifactor is 0.18. In traditional Structural Equation Modeling (SEM), an RMSEA $> 0.10$ is generally considered indicative of poor or unacceptable absolute fit.
- The paper places immense weight on the finding that "Contributor" accounts for 9% of the variance, interpreting this as "provenance variance" representing "unobserved differences in training data, compute resources, and fine-tuning recipes". However, on Hugging Face, the uploader account is an extremely noisy proxy. Accounts like Meta-Llama upload base models, while automated accounts (e.g., TheBloke, Bartowski) upload thousands of quantized models. If the variance is driven heavily by the latter, the 9% might simply represent quantization degradation, not elite "secret sauce" training recipes. The authors must explicitly detail (perhaps in Section 4.1 or an appendix) how "Contributor" was extracted, cleaned, and categorized. The language in Section 5.3 should be slightly softened to acknowledge that "Contributor" is a noisy proxy that mixes original creators with community quantizers and mergers.
- In observational datasets like the Open LLM Leaderboard, Architecture, Contributor, and Model Size are heavily collinear (e.g., Meta introduces specific architectures, acts as the contributor, and dictates the massive 70B parameter scale).

---

> ### Author Rebuttal · Authors · 2026-03-28
>
> Thank you so much for your technically sophisticated evaluation and for recognizing the core contributions of our work. We are grateful that your questions are precisely the right ones for strengthening the paper's credibility!
> # Data Collinearities
> This is a genuine challenge in measuring ecosystems. We will add explicit discussion of collinearity handling with the regularizing properties of MH-RM estimation, bootstrap stability checks, and crossed random effects:
> - MH-RM latent regression estimates contributor effects as random intercepts ($u_{c(i)}$ in Eq. 8) and size and deployments as fixed effects. The algorithm integrates over the posterior of latent abilities via Metropolis-Hastings sampling, which doesn’t require inverting a collinear design matrix (unlike OLS). Random effects shrink toward the grand mean proportionally to group size, so prolific uploaders with many models are estimated precisely while singleton contributors shrink–naturally regularizing against absorbing fixed-effect signals. (And, the Robbins-Monro update provides a smoother path toward maxima of the likelihood surface by averaging out Monte Carlo error.)
> - Bootstrap-by-item design provides a direct check: if collinearity were driving unstable attribution, we would observe high variance in $\hat{\beta}$ across bootstraps. Table 3's reliability indices ($R_d = 0.97$ for g) indicate stability.
> - G-theory uses a fully crossed random-effects specification (Listing 3) that partitions variance into main effects and all interactions. Collinearity here would be large interaction terms in Table 2, rather than being absorbed into main effects.
> # Contributor as a Noisy Proxy
> Contributor is indeed a noisy proxy even among noisy proxies, operationalized from the Hugging Face upload metadata (e.g., account, model availability, footnote 1, §3.2), mixing original and downstream developers (see implications in Appendix A.2,"Ecosystem…pseudo-replication"). We commit to more clearly scoping this, as the heterogeneity strengthens our broader findings–that even this noisy proxy captures some aggregate effect of provenance variation that confounds other comparisons, affirming:
> > _We can more reliably rank-order models by metadata on how they were uploaded than we can using architecture or deployment type._
>
> Additionally, we will share all pipeline code; until camera-ready, {python} `re.sub(r'\W+','_',fullname.split('/')[0]).strip('_').lower()` can be used to obtain our account labels.
> # Fit & RMSEAs
> Indeed, RMSEA = 0.25 (bifactor) and 0.18 (correlated bifactor) are well above traditional thresholds (0.06). We believe this reflects three reinforcing mechanisms:
> - Binary item sparsity. Binary responses with floor/ceiling effects on many items limit achievable tetrachoric correlation recovery, inflating discrepancy metrics.
> - Severe local dependencies. As the reviewer suspects, shared artifacts (EleutherAI harness, prompt templates, tokenization, answer-parsing heuristics) induce residual item couplings that no factor structure can absorb into 𝛷𝛬𝛷${}^\top$. Our bootstrapped score test (LM/MI) diagnostics in Figs 4–7 and Apdx C & E directly quantify this: even under the best-fitting structures, systematic residual dependencies persist across benchmarks. Bifactor is far from perfect but is the *least misspecified* decomposition, and the remaining misfit is itself a measure of noise in the ecosystem.
> - RMSEA and sample properties. RMSEA is sensitive to factors such as sample size, number of indicators, and data characteristics (Kenny et al., 2015; Groskurth et al., 2024). Fixed cutoff values like 0.06 derived from simulation studies may not be universally applicable across diverse real-world settings (McNeish & Wolf, 2023). With n ≈ 4000, p $\ge 30$, RMSEA variants can become sensitive to trivially small covariance residuals. Traditional cutoffs were calibrated on n = 200–500 with p < 30 (Remark C.2). At our scale, RMSEA can inflate while leaving substantive conclusions intact.
>
> Thankfully, our model choice does not rest on covariance RMSEA; we establish practical and statistical fit with our robustness checks (OOS AUC and MAE, score tests, permutation controls, Apdx C, E, Fig. 5-7, Tab. 5, etc), which we will better foreground. Tab. 4 shows bifactor-family models dominate across traditional criteria (e.g., CFI, SRMR, BIC, log-lik.) in within-bootstrap percent-rank comparisons. Tab 5 confirms that the bifactor significantly improves over every simpler structure in 100% of bootstraps via LRTs.
>
> We will add explicit discussion and references of why traditional covariance fit thresholds may be less meaningful with this application and foreground our predictive validation tests.
> ## References
> - Groskurth, Bluemke & Lechner (2024). Why we need to abandon fixed cutoffs…
> - Kenny, Kaniskan & McCoach (2015). The performance of RMSEA in models…
> - McNeish & Wolf (2023). Dynamic fit index cutoffs for confirmatory…
>
> Thanks again for your insightful review!

---

> > ### Author Rebuttal · Reviewer_gr11 · 2026-04-01
> >
> > Thank you to the authors for the thoughtful and comprehensive rebuttal. I appreciate you directly addressing my concerns and committing to the necessary clarifications in the final manuscript. I am entirely satisfied with your responses and will increase my score accordingly. Best of luck with the rest of the review process!
> >
> > I also read the reviews from T2rQ and PoUb, which gave negative scores. While the final decision is entirely up to the Area Chair's discretion, I sincerely hope those reviews do not lead to a rejection. They fall well below the standard of quality this community should expect. Computer Science is the only field that seems to tolerate such poor-quality reviews without any consequences for the reviewers. Our review system is broken.

---

> > > ### Author Response · Authors · 2026-04-06
> > >
> > > Thank you! We have appreciated your engagement and forthrightness as a reviewer.
> > >
> > > We are also grateful for your acknowledgement about the rebuttal process, and we share your same hope. We appreciate that your comments have been constructive and helpful, and we want to emphasize how instrumental they are in moving our revision of this work forward. Best!

---

### Official Review · Reviewer_PoUb · 2026-03-13

**Soundness:** 3
**Presentation:** 1
**Significance:** 2
**Originality:** 3
**Overall Recommendation:** 3
**Confidence:** 4

**Summary:**

In this paper, the authors formulate a psychometric framework for measuring noise in AI system evaluation environments as leaderboards. The experiments focus on six different tasks. They apply various statistical methods to test assumed leaderboard structures, quantify sources of noise affecting rankings, and examine how benchmarking practices relate to leaderboard changes.

**Compliance With Llm Reviewing Policy:**

Affirmed.

**Final Justification:**

The authors demonstrated a strong commitment to improving the paper’s clarity. While I still believe substantial revisions would be needed to better fit the venue, I slightly increased my score to acknowledge the work's technical soundness.

**Key Questions For Authors:**

I don’t have any concrete questions that would modify my view of the paper.

**Limitations:**

yes

**Strengths And Weaknesses:**

Strengths
- The authors’ objective of measuring noise in AI Benchmark ecosystems is valuable for the AI Evaluation field.

Weaknesses
- The paper is difficult to follow due to its lack of clarity and organisation. Several sentences are overly technical or convoluted, making it difficult to grasp the core concepts. The structure could also be improved: proofs and propositions appear scattered throughout the text, and some appendix references seem to be missing. As a result, by the time the reader reaches the experimental section, the paper's main goal is not entirely clear. A clearer presentation of the paper’s objective and a more streamlined organisation would significantly improve readability.
- Formulation is vague and largely under-explained. There are many equations whose terms are not adequately defined. The paper would benefit from clearer explanations of its formulation, which now appears disconnected.

---

> ### Author Rebuttal · Authors · 2026-03-30
>
> Thank you for the constructive feedback and for recognizing the value of our objective and the originality of our contribution. We address your important concerns below.
> # Missing References
> We sincerely apologize for the broken cross-references (“??”). A single-character LaTeX typo disabled several `\ref{*}` commands in the submitted PDF. All referenced content, appendices, and tables exist in full, and this is easily fixed.
> # Presentation
> ## Sentences and Organization
> We agree with the reviewer that there are sections that could be clearer. We are committed to ensuring that the text is accessible to our broader ICML audience. Our comments to reviewer sdUW have examples that demonstrate our commitment.
> ## Example
> To further demonstrate our resolve to improve clarity, we are preparing a revision incorporating your feedback to articulate the why, what, and how of ecosystem measurement:
> > AI leaderboard rankings can conflate size with developer innovation. We seek to improve practitioner decisions by disentangling the various signals within benchmark scores. Our framework for measuring AI capabilities and scaling laws controls for many sources of variance in AI benchmark ecosystems.  We measure the score variation found within each LLM (§3.1) and contributed by their ecosystem (§3.2) before combining these tools to estimate de-noised scaling laws (§3.3).
> > - Step 1 (CFA): Because the variables of interest are not observed, we take a rigorous approach to determining the best latent structure of AI abilities (§3.1). We use multiple tests to interrogate these structures' propensities to overfit. We learn, for instance, that regardless of the latent structure tested, significant nonrandom error (unrelated to the AI capability of interest) persists that could threaten the validity of comparing leaderboard scores.
> > - Step 2 (G-theory): Thus, we then fully decompose the ecosystem noise separating sources of variation and their interactions (§3.2). We observe the same generalizable distribution of variance across estimations. We learn, for instance, that we can more reliably rank-order AI performance by contributor metadata than we can based on architecture or deployment type.
> > - Step 3 (Latent Regression): Using latent regression (§3.3), we estimate ecosystem effects on specific AI benchmark behaviors so we can infer what tuning practices help underlying capabilities as they scale. We learn, for example, that, while scaling improves performance, this is not uniformly true about the latent abilities unique to specific benchmarks. With explicit controls, however, we can disentangle scale vs innovation in benchmark rankings. See ‘Table 3’ in our comment to T2rQ for more details.
> # Formulations
> Thank you for your time and for pointing us toward unclear formulations. We recognize that Confirmatory Factor Analysis (CFA), G-theory, and Item Response Theory (IRT) are not standard in our broader ML community; we agree that the mathematical formulation feels disconnected from the narrative. We have already begun making the conceptual contributions of these methods accessible to wider audiences.
> ## Example Reformulation
> We demonstrate our commitment to make our notation accessible by reformulating our most complex estimation (latent regression):
> > The latent regression has two components: a measurement model (linking item responses to latent traits), and a structural model (explaining latent traits variance). The probability of LLM 𝑖 correctly answering item 𝑗 is given by:
>
> > $P(y_{ij} = 1 | 𝜽_i, 𝒂_j, d_j) = \sigma(𝒂_j^\top 𝜽_i + d_j)$,
>
> > where $\sigma(x) = (1+e^{-x})^{-1}$ is logistic. Here, $𝜽_i$ is the vector of LLM 𝑖’s ability scores, $𝒂_j$ is a vector specifying how strongly item 𝑗 relates to each latent factor, and $d_j$ is a scalar related to the difficulty of the item. The sparsity pattern of $𝒂_j$ defines the factor structural model; for our bifactor model (§3.1), each item loads on the general factor and one specific factor.
>
> > A core innovation to incorporating ecosystem factors lies in modeling the latent trait matrix $𝚯 \in \mathbb{R}^{N \times K}$ as a decomposition via a mixed-effects linear model:
>
> > 𝚯=𝐕𝚪+𝐖𝛇+𝚬, where
> > - 𝐕: N x F design matrix of F covariates for each model (e.g., size, deployment specification)
> > - 𝚪: F x K matrix of fixed-effect regression coefficients. For example, $\Gamma_{fk}$ is the effect of covariate f on latent ability k
> > - 𝐖: N x R design matrix for random effects (e.g., contributors)
> > - 𝛇 : R x K matrix of random effects
> > - 𝚬: N x K matrix of residuals
>
> The above replaces the following part in our original text:
> > We regress latent abilities on size and deployment covariates, with contributor random effects to correct pseudo-replication: 𝜽 = 𝐁𝐳+𝐮(c)+𝜺, $𝐳=(1,𝐱,𝐰^\top)^\top$, where 𝐮(c) ∼ 𝒩(0, 𝚺) captures provenance variance.
>
> Thank you again for the opportunity to demonstrate our commitment to improve the clarity of our work.

---

> > ### Author Rebuttal · Reviewer_PoUb · 2026-04-04
> >
> > Thank you for the very constructive and thoughtful rebuttal. I appreciate the effort the authors put into addressing my comments, and the proposed revisions clearly demonstrate a strong commitment to improving the paper’s clarity. The additional explanations and reformulations are helpful, and I acknowledge the approach's technical soundness. However, I believe that fully addressing these issues would require more substantial restructuring of the introduction and methodology sections rather than incremental revisions.
> >
> > Therefore, although I appreciate the quality of the work and the authors’ efforts, I remain concerned about its accessibility for ICML. I slightly adjusted my score to reflect the technical contribution's strength, but I still believe the paper requires substantial revisions to better fit the venue.

---

> > > ### Author Response · Authors · 2026-04-06
> > >
> > > Thank you for acknowledging our commitment to incorporating your feedback. We are also grateful for your comments regarding the high quality of the work, the technical contributions, and our efforts to make our study more accessible.
> > >
> > > As we are preparing our broad structural revisions based on cumulative reviewer feedback with particular focus on introduction and methods, we are still interested in understanding your recommendations for improving the venue appropriateness of the study. We seek to make the kinds of transformative revisions that would help our broader community benefit from the contributions of the work. Could you help us to better understand the types of restructuring of the initial sections that you envision could lead to greater accessibility for a broader ICML audience?
> > >
> > > We deeply appreciate your feedback and thank you for your time.

---

### Official Review · Reviewer_T2rQ · 2026-03-17

**Soundness:** 2
**Presentation:** 2
**Significance:** 2
**Originality:** 3
**Overall Recommendation:** 4
**Confidence:** 4

**Summary:**

This paper applies psychometric methods (CFA/SEM, Generalizability Theory, mixed latent regression) to the Open LLM Leaderboard to decompose sources of variance in benchmark scores. The main claims are "human contributor factors" explains more variance (9%) than model architecture (4.8%); model size accounts for 36% of explainable variance; and scaling laws are benchmark-dependent (Benchmark PSI = 32.35%). The paper introduces SNR_β and PSI metrics and argues that bifactor structure best explains the ecosystem's latent structure.

**Compliance With Llm Reviewing Policy:**

Affirmed.

**Final Justification:**

I thank the authors for the clarification and have updated my score accordingly.

**Key Questions For Authors:**

1. How do the authors draw the line between signal and noise? The variables indicated as "noise" by the authors are actually important "signals" for model evaluation. E.g., the abstract says contributor variance reveals "benchmark noise from submission practices," but Section 5.2 calls it "meaningful signals" from training data, compute, and tuning recipes. Deployment variables like RLHF and chat templates are also important engineering choices that change capabilities. Nearly every facet captures signal that matters for model evaluation, so it's hard to know what's considered as "noise" in the authors' framework.

2. Can the authors provide concrete examples of how model rankings change after applying the controls proposed by the authors? The paper's central claim is that naive rankings conflate capability with artifacts, but this is not demonstrated on actual model rankings. I'd find it useful if the authors showed, e.g., "Model X is ranked #1 on the raw leaderboard but drops to #5," connecting these results to the discussion.

3. Variance decomposition tends to be sensitive to how categories are defined, especially in terms of granularity. Would the authors observe consistent patterns if they used more fine-grained categories? If the authors coded architecture more coarsely (e.g., dense transformer vs. MoE only) or more finely, how would the variance shares change?

**Limitations:**

Yes

**Strengths And Weaknesses:**

Strengths: The idea of treating leaderboards as measurement instruments and applying psychometric tools is relevant and novel.

Weaknesses: The paper does not clearly define what counts as signal vs. noise - the abstract alludes that contributor variance indicates "benchmark noise from submission practices," but Section 5.2 says the opposite ("meaningful signals, rather than noise"), and deployment variables like RLHF/chat templates are engineering decisions that change capabilities, not artifacts. Multiple broken cross-references ("Appendix ??", "Table ??", "Code ??") throughout and Table 3 - a key result - needs more discussion in the main text.

---

> ### Author Rebuttal · Authors · 2026-03-29
>
> Thank you for your careful reading and constructive questions! We address each with new analyses and clarifications.
> # Signal vs. Noise: A Deliberate Dual Interpretation
> We agree with the reviewer: variables like RLHF are meaningful signals for evaluating methods, but they act as 'noise' when the goal is purely evaluating base architectures. This is actually a core design feature of our framework, which we acknowledge was insufficiently foregrounded. Whether a variance component is signal or noise depends on the inference. See §5.2 for two evaluation modes:
> - Architecture-centric ("Which architecture is better?"): contributor/provenance variance is noise because it confounds architectural comparisons with unobserved training recipes.
> - Method-centric ("Which training practices work?"): the same contributor variance is signal because it captures the effect of data mixtures, compute, and tuning.
>
> Thus, a facet is "noise" relative to a generalization decision, not ontologically (see Brennan, 2001). RLHF and chat templates are indeed engineering choices that affect capabilities, even with noisy labels, and our framework quantifies how much they change capabilities on each latent dimension. Lumping into a single leaderboard rank without explicit modeling makes it impossible to know whether a ranking difference reflects architecture, scale, or training methodology (rather than making it an artifact). Our framework makes this decomposition explicit and actionable. We will clarify the abstract to avoid implying contributor variance is purely artifactual.
>
> # Ranking Changes
> We too think that concrete ranking examples need to be added. Comparing leaderboard ranks (as benchmark mean) with percentile ranks of the general factor (as a capability influencing all benchmarks), controlling for ecosystem noise):
> | Metric | r | 95% CI |
> |-|-|-|
> | Kendall | 0.66 | [0.65, 0.68] |
> | Spearman | 0.86 | [0.85, 0.86] |
> | dCor | 0.82 | [0.81, 0.83] |
> Rankings (0.86) and their distribution (0.82) are broadly preserved but with meaningful localized disruptions (0.66): only 1 model in the top 1% overall remains in the top 1% general after noise control; 90% remain in the top decile. This confirms that top leaderboard ranks are sensitive to ecosystem noise, even if broader ordering is robust. This directly demonstrates the paper's central claim: naive rankings conflate capability with artifacts most consequentially at the margins where decisions are made.
> # Sensitivity to Categorizations
> We appreciate this question! We conducted the requested robustness analysis at multiple granularity levels and via Bayesian estimation with 95% credible intervals.  Main-effects-only model variance share % at (different granularities):
> | Facet | Finest | Fine (Tab. 2) | Coarsest | Coarse Grain |
> |-|-|-|-|-|
> | (A)rchitecture | 11.0 (1120) | 10.1 (119) | 9.2 (54) | LLM `Family' |
> | (B)enchmark | 46.4 | 47.2 | 46.8 | all (6) |
> | (C)ontribution | 15.8 (1163) | 13.8 (878) | 13.8 (714) | account |
> | (D)eployment | 2.3 (74) | 3.8 (61) | 3.5 (7) | HF `type’ |
> The key finding–contributor variance exceeds architecture variance–holds across granularities. The key ordering B >> C > A > D is stable.
>
> To directly estimate an answer to your question about sensitivity of variance shares to categorizations, we also re-estimate Table 2 full model using a MCMC Bayesian framework with 4 chains and weak priors, reporting main effect variance share %:
> | Facet | Mean | 95% CI |
> |-|-|-|
> | A | 4.8 | [2.0, 8.5] |
> | B | 50.2 | [26.9, 74.6] |
> | C | 8.2 | [3.9, 12.6] |
> | D | 2.3 | [0.6, 4.9] |
> All $\hat{R} \le 1.01$, indicating convergence. The 95% CIs for C and A do not overlap at their central tendencies, supporting the robustness of our headline finding despite dataset limitations.
> # Broken References
> We must transparently disclose that a single-character LaTeX typo prevented cross-references from rendering (appearing as "??"). All referenced content exists in full in the appendices. We acknowledge the impact on readability and commit to correcting this in revision.
> # Table 3
> We agree that this table’s reporting of scaling slopes (𝛽) and their reliability ($R_d$) per latent dimension merits substantially more discussion (and even a figure with deployment-dependent scaling laws). More key patterns include: (a) $g$ scales strongly and reliably (𝛽 = 1.05, $R_d = 0.97$) and MMLU continues to scale strongly (𝛽 = 0.52, $R_d = 0.57$); (b) IFEval shows no size effect (𝛽 = -0.08) but strong deployment effects (chat template: 𝛽 = 1.82, $R_d = 0.99$); (c) interestingly, MuSR’s mildly negative size relationship with chat templates suggests an inverse relationship between this benchmark's construct and IFT practice, in light of the additional Lagrangian multiplier test evidence (Fig. 7) where MuSR-IFEval residual covariance had broader structural overfit. We will expand the main-text discussion of these results substantially.
>
> Thank you again for your helpful and constructive feedback!

---

> > ### Author Rebuttal · Reviewer_T2rQ · 2026-04-01
> >
> > Thank you for the thorough responses: (1) The clarification on signal/noise is useful - this is a framing question rather than a flaw, and the proposed abstract revision should fix it. (2) The correlation analysis is exactly the kind of evidence I was looking for. I hope that concrete examples (e.g., models surviving noise control) can be provided in the revised draft. (3) The additional sensitivity checks address my concern regarding the robustness. I have updated my score accordingly.

---

> > > ### Author Response · Authors · 2026-04-06
> > >
> > > We are grateful for the opportunity to respond to your constructive feedback and questions. Your questions are exactly those that are helping point our revisions toward a more accessible space for our broader ICML community and clarify the helpful contributions. In response to your hope, we fully intend to include results from this rebuttal opportunity in the final version of our study.
> > >
> > > We are committed to making the revised version of this analysis more easily accessible to our broader ML community, and your feedback has strengthened our work. Thank you.

---

### Decision · Program_Chairs · 2026-04-30

**Decision:**

Accept (regular)

**Comment:**

Though the reviews are mixed, the majority of reviewers are leaning towards acceptance, emphasizing that the problem proposed ("modern LLM leaderboards getting treated as if they were clean, single-number measurements of model capability") is important and timely and that the "idea of treating leaderboards as measurement instruments and applying psychometric tools is relevant and novel."  The reviewers highlight that the method proposed is "statistically mature", and that the authors' findings are highly relevant to the analysis of scaling laws broadly, and have useful implications for evaluation as well. The concerns raised by the reviewers were largely addressed by the author's detailed, well-grounded rebuttals. The AC team agrees that the contributions of the paper are sufficient for acceptance, but encourages the authors to include the updated analysis and clarifications captured in the rebuttal into the camera ready, with particular emphasis on the organization and clarity of the narrative to ensure the value is not lost due to reader confusion.